# Federated Accelerated Stochastic Gradient Descent

**Honglin Yuan**
Stanford University
yuanhl@stanford.edu

**Tengyu Ma**
Stanford University
tengyuma@stanford.edu

## Abstract

We propose Federated Accelerated Stochastic Gradient Descent (FEDAC), a principled acceleration of Federated Averaging (FEDAVG, also known as Local SGD) for distributed optimization. FEDAC is the first provable acceleration of FEDAVG that improves convergence speed and communication efficiency on various types of convex functions. For example, for strongly convex and smooth functions, when using $M$ workers, the previous state-of-the-art FEDAVG analysis can achieve a linear speedup in $M$ if given $\tilde{\mathcal{O}}(M)$ rounds of synchronization, whereas FEDAC only requires $\tilde{\mathcal{O}}(M^{\frac{1}{3}})$ rounds. Moreover, we prove stronger guarantees for FEDAC when the objectives are third-order smooth. Our technique is based on a potential-based perturbed iterate analysis, a novel stability analysis of generalized accelerated SGD, and a strategic tradeoff between acceleration and stability.

## 1 Introduction

Leveraging distributed computing resources and decentralized data is crucial, if not necessary, for large-scale machine learning applications. Communication is usually the major bottleneck for parallelization in both data-center settings and cross-device federated settings [Kairouz et al., 2019].

We study the distributed stochastic optimization $\min_{w \in \mathbb{R}^d} F(w) := \mathbb{E}_{\xi \sim \mathcal{D}} f(w; \xi)$ where $F$ is convex. We assume there are $M$ parallel workers and each worker can access $F$ at $w$ via oracle $\nabla f(w; \xi)$ for independent sample $\xi$ drawn from distribution $\mathcal{D}$. We assume synchronization (communication) among workers is allowed but limited to $R$ rounds. We denote $T$ as the parallel runtime.

One of the most common and well-studied algorithms for this setting is *Federated Averaging* (FEDAVG) [McMahan et al., 2017], also known as Local SGD or Parallel SGD [Mangasarian, 1995, Zinkevich et al., 2010, Coppola, 2014, Zhou and Cong, 2018] in the literature.[1] In FEDAVG, each worker runs a local thread of SGD [Robbins and Monro, 1951], and periodically synchronizes with other workers by collecting the averages and broadcast to all workers. The analysis of FEDAVG [Stich, 2019a, Stich and Karimireddy, 2019, Khaled et al., 2020, Woodworth et al., 2020] usually follows the perturbed iterate analysis framework [Mania et al., 2017] where the performance of FEDAVG is compared with the idealized version with infinite synchronization. The key idea is to control the stability of SGD so that the local iterates held by parallel workers do not differ much, even with infrequent synchronization.

We study the acceleration of FEDAVG and investigate whether it is possible to improve convergence speed and communication efficiency. The main challenge for introducing acceleration lies in the disaccord of acceleration and stability. Stability is essential for analyzing distributed algorithms such as FEDAVG, whereas momentum applied for acceleration may amplify the instability of the algorithm. Indeed, we show that standard Nesterov accelerated gradient descent algorithm [Nesterov, 2018] *may not be initial-value stable even for smooth and strongly convex functions*, in the sense that the initial

Table 1: **Summary of results on the synchronization rounds $R$ required for linear speedup in $M$.** All bounds hide multiplicative polylog factors and variables other than $M$ and $T$ for ease of presentation. Notation: $M$: number of workers; $T$: parallel runtime; "Asm." stands for Assumption.

| Asm. | Algorithm | Synchronization Required for Linear Speedup | | Reference |
| --- | --- | --- | --- | --- |
| | | Strongly Convex | General Convex | |
| A1 | FEDAVG | $T^{\frac{1}{2}}M^{\frac{1}{2}}$ | – | [Stich, 2019a] |
| | | $T^{\frac{1}{3}}M^{\frac{1}{3}}$ | – | [Haddadpour et al., 2019b] |
| | | $M$ | $T^{\frac{1}{2}}M^{\frac{3}{2}}$ | [Stich and Karimireddy, 2019] |
| | | $M$ | $T^{\frac{1}{2}}M^{\frac{3}{2}}$ | [Khaled et al., 2020] |
| | FEDAC | $\boldsymbol{M^{\frac{1}{3}}}$ | $\min\{\boldsymbol{T^{\frac{1}{4}}M^{\frac{3}{4}}, T^{\frac{1}{3}}M^{\frac{2}{3}}}\}$ | **Theorems 3.1, E.1 and E.2** |
| A2 | FEDAVG | $\max\{\boldsymbol{T^{-\frac{1}{2}}M^{\frac{1}{2}},1}\}$ | $T^{\frac{1}{2}}M^{\frac{3}{2}}$ | **Theorems 3.4 and E.4** |
| | FEDAC | $\max\{\boldsymbol{T^{-\frac{1}{6}}M^{\frac{1}{6}},1}\}$ | $\max\{\boldsymbol{T^{\frac{1}{4}}M^{\frac{1}{4}}, T^{\frac{1}{6}}M^{\frac{1}{2}}}\}$ | **Theorems 3.3 and E.3** |

infinitesimal difference may grow exponentially fast (see Theorem 4.2). This evidence necessitates a more scrutinized acceleration in distributed settings.

We propose a principled acceleration for FEDAVG, namely *Federated Accelerated Stochastic Gradient Descent* (FEDAC), which provably improves convergence rate and communication efficiency. Our result extends the results of Woodworth et al. [2020] on LOCAL-AC-SA for quadratic objectives to broader objectives. To the best of our knowledge, this is the **first provable acceleration** of FEDAVG (and its variants) for general or strongly convex objectives. FEDAC parallelizes a generalized version of Accelerated SGD [Ghadimi and Lan, 2012], while we carefully balance the acceleration-stability tradeoff to accommodate distributed settings. Under standard assumptions on smoothness, bounded variance, and strong convexity (see Assumption 1 for details), FEDAC converges at rate $\tilde{\mathcal{O}}(\frac{1}{MT} + \frac{1}{TR^3})$.[2] The bound will be dominated by $\tilde{\mathcal{O}}(\frac{1}{MT})$ for $R$ as low as $\tilde{\mathcal{O}}(M^{\frac{1}{3}})$, which implies the synchronization $R$ required for linear speedup in $M$ is $\tilde{\mathcal{O}}(M^{\frac{1}{3}})$.[3] In comparison, the state-of-the-art FEDAVG analysis Khaled et al. [2020] showed that FEDAVG converges at rate $\tilde{\mathcal{O}}(\frac{1}{MT} + \frac{1}{TR})$, which requires $\tilde{\mathcal{O}}(M)$ synchronization for linear speedup. For general convex objective, FEDAC converges at rate $\tilde{\mathcal{O}}(\frac{1}{\sqrt{MT}} + \frac{1}{T^{\frac{1}{3}}R^{\frac{2}{3}}})$, which outperforms both state-of-the-art FEDAVG $\tilde{\mathcal{O}}(\frac{1}{\sqrt{MT}} + \frac{1}{T^{\frac{1}{3}}R^{\frac{1}{3}}})$ by Woodworth et al. and Minibatch-SGD baseline $\Theta(\frac{1}{\sqrt{MT}} + \frac{1}{R})$ [Dekel et al., 2012].[4] We summarize communication bounds and convergence rates in Tables 1 and 2 (on the row marked A1).

Our results suggest an **intriguing synergy between acceleration and parallelization**. In the single-worker sequential setting, the convergence is usually dominated by the term related to stochasticity, which is in general not possible to be accelerated [Nemirovski and Yudin, 1983]. In distributed settings, the communication efficiency is dominated by the overhead caused by infrequent synchronization, which can be accelerated as we show in the convergence rates summary Table 2.

We establish **stronger guarantees for FEDAC when objectives are 3$^{\text{rd}}$-order-smooth**, or "close to be quadratic" intuitively (see Assumption 2 for details). For strongly convex objectives, FEDAC converges at rate $\tilde{\mathcal{O}}(\frac{1}{MT} + \frac{1}{T^2R^6})$ (see Theorem 3.3). We also prove the convergence rates of FEDAVG in this setting for comparison. We summarize our results in Tables 1 and 2 (on the row marked A2).

We empirically verify the efficiency of FEDAC in Section 5. Numerical results suggest a considerable improvement of FEDAC over all three baselines, namely FEDAVG, (distributed) Minibatch-SGD, and (distributed) Accelerated Minibatch-SGD [Dekel et al., 2012, Cotter et al., 2011], especially in the regime of highly infrequent communication and abundant workers.

Table 2: **Summary of results on convergence rates.** All bounds omit multiplicative polylog factors and additive exponential decaying term (for strongly convex objective) for ease of presentation. Notation: $D_0$: $\|w_0 - w^*\|$; $M$: number of workers; $T$: parallel runtime; $R$: synchronization; $\mu$: strong convexity; $L$: smoothness; $Q$: 3$^{rd}$-order-smoothness (if Assumption 2 is assumed).

| Assumption | Algorithm | Convergence Rate ($\mathbb{E}[F(\cdot)] - F^* \leq \cdots$) | Reference |
|---|---|---|---|
| A1($\mu > 0$) | FEDAVG | exp. decay $+\frac{\sigma^2}{\mu MT} + \frac{L\sigma^2}{\mu^2 TR}$ | [Woodworth et al., 2020] |
| | FEDAC | exp. decay $+\frac{\sigma^2}{\mu MT} + \min\left\{\frac{L\sigma^2}{\mu^2 TR^2}, \frac{L^2\sigma^2}{\mu^3 TR^3}\right\}$ | **Theorem 3.1** |
| A2($\mu > 0$) | FEDAVG | exp. decay $+\frac{\sigma^2}{\mu MT} + \frac{Q^2\sigma^4}{\mu^5 T^2 R^2}$ | **Theorem 3.4** |
| | FEDAC | exp. decay $+\frac{\sigma^2}{\mu MT} + \frac{Q^2\sigma^4}{\mu^5 T^2 R^6}$ | **Theorem 3.3** |
| A1($\mu = 0$) | FEDAVG | $\frac{LD_0^2}{T} + \frac{\sigma D_0}{\sqrt{MT}} + \frac{L^{\frac{1}{3}}\sigma^{\frac{2}{3}}D_0^{\frac{4}{3}}}{T^{\frac{1}{3}}R^{\frac{1}{3}}}$ | [Woodworth et al., 2020] |
| | FEDAC | $\frac{LD_0^2}{TR} + \frac{\sigma D_0}{\sqrt{MT}} + \min\left\{\frac{L^{\frac{1}{3}}\sigma^{\frac{2}{3}}D_0^{\frac{4}{3}}}{T^{\frac{1}{3}}R^{\frac{2}{3}}}, \frac{L^{\frac{1}{2}}\sigma^{\frac{1}{2}}D_0^{\frac{3}{2}}}{T^{\frac{1}{4}}R^{\frac{3}{4}}}\right\}$ | **Theorems E.1 and E.2** |
| A2($\mu = 0$) | FEDAVG | $\frac{LD_0^2}{T} + \frac{\sigma D_0}{\sqrt{MT}} + \frac{Q^{\frac{1}{3}}\sigma^{\frac{2}{3}}D_0^{\frac{5}{3}}}{T^{\frac{1}{3}}R^{\frac{1}{3}}}$ | **Theorem E.4** |
| | FEDAC | $\frac{LD_0^2}{TR} + \frac{\sigma D_0}{\sqrt{MT}} + \frac{L^{\frac{1}{3}}\sigma^{\frac{2}{3}}D_0^{\frac{4}{3}}}{M^{\frac{1}{3}}T^{\frac{1}{3}}R^{\frac{2}{3}}} + \frac{Q^{\frac{1}{3}}\sigma^{\frac{2}{3}}D_0^{\frac{5}{3}}}{T^{\frac{1}{3}}R}$ | **Theorem E.3** |

## 1.1 Related work

The analysis of FEDAVG (a.k.a. Local SGD) is an active area of research. Early research on FEDAVG mostly focused on the particular case of $R = 1$, also known as "one-shot averaging", where the iterates are only averaged once at the end of procedure [Mcdonald et al., 2009, Zinkevich et al., 2010, Zhang et al., 2013, Shamir and Srebro, 2014, Rosenblatt and Nadler, 2016]. The first convergence result on FEDAVG with general (more than one) synchronization for convex objectives was established by [Stich, 2019a] under the assumption of uniformly bounded gradients. [Stich and Karimireddy, 2019], Haddadpour et al. [2019b], Dieuleveut and Patel [2019], Khaled et al. [2020] relaxed this requirement and studied FEDAVG under assumptions similar to our Assumption 1. These works also attained better rates than [Stich, 2019a] through an improved stability analysis of SGD. However, recent work [Woodworth et al., 2020] showed that all the above bounds on FEDAVG are strictly dominated by minibatch SGD [Dekel et al., 2012] baseline. Woodworth et al. [2020] provided the first bound for FEDAVG that can improve over minibatch SGD for certain cases. This is to our knowledge the state-of-the-art bound for FEDAVG and its variants. Our FEDAC uniformly dominates this bound on FEDAVG.

The specialty of quadratic objectives for better communication efficiency has been studied in an array of contexts [Zhang et al., 2015, Jain et al., 2018]. Woodworth et al. [2020] studied an acceleration of FEDAVG but was limited to quadratic objectives. More generally, Dieuleveut and Patel [2019] studied the convergence of FEDAVG under bounded 3$^{rd}$-derivative, but the bounds are still dominated by minibatch SGD baseline [Woodworth et al., 2020]. Recent work by Godichon-Baggioni and Saadane [2020] studied one-shot averaging under similar assumptions. Our analysis on FEDAVG (Theorem 3.4) allows for general $R$ and reduces to a comparable bound if $R = 1$, which is further improved by our analysis on FEDAC (Theorem 3.3).

FEDAVG has also been studied in other more general settings. A series of recent papers (*e.g.*, [Zhou and Cong, 2018, Haddadpour et al., 2019a, Wang and Joshi, 2019, Yu and Jin, 2019, Yu et al., 2019a,b]) studied the convergence of FEDAVG for non-convex objectives. We conjecture that FEDAC can be generalized to non-convex objectives to attain better efficiency by combining our result with recent non-convex acceleration algorithms (*e.g.*, [Carmon et al., 2018]). Numerous recent papers [Khaled et al., 2020, Li et al., 2020b, Haddadpour and Mahdavi, 2019, Koloskova et al., 2020] studied FEDAVG in heterogeneous settings, where each worker has access to stochastic gradient oracles from different distributions. Other variants of FEDAVG have been proposed in the face of heterogeneity [Pathak and Wainwright, 2020, Li et al., 2020a, Karimireddy et al., 2020, Wang et al., 2020]. We defer the analysis of FEDAC for heterogeneous settings for future work. Other techniques, such as quantization, can also reduce communication cost [Alistarh et al., 2017, Wen et al., 2017, Stich

et al., 2018, Basu et al., 2019, Mishchenko et al., 2019, Reisizadeh et al., 2020]. We refer readers to [Kairouz et al., 2019] for a more comprehensive survey of the recent development of algorithms in Federated Learning.

Stability is one of the major topics in machine learning and has been studied for a variety of purposes [Yu and Kumbier, 2020]. For example, Bousquet and Elisseeff [2002], Hardt et al. [2016] showed that algorithmic stability can be used to establish generalization bounds. Chen et al. [2018] provided the stability bound of standard Accelerated Gradient Descent (AGD) for *quadratic objectives*. To the best of our knowledge, there is no existing (positive or negative) result on the stability of AGD for general convex or strongly convex objectives. This work provides the first (negative) result on the stability of standard deterministic AGD, which suggests that standard AGD may not be initial-value stable even for strongly convex and smooth objectives (Theorem 4.2).[5] This result may be of broader interest. The tradeoff technique of FEDAC also provides a possible remedy to mitigate the instability issue, which may be applied to derive better generalization bounds for momentum-based methods.

The stochastic optimization problem $\min_{w \in \mathbb{R}^d} F(w) := \mathbb{E}_{\xi \sim \mathcal{D}} f(w; \xi)$ we consider in this paper is commonly referred to as the *stochastic approximation* (SA) problem [Kushner et al., 2003]. Another related question is the *empirical risk minimization* (ERM) problem [Vapnik, 1998], defined as $\min_{w \in \mathbb{R}^d} F(w) := \frac{1}{N} \sum_{i=1}^{N} f(w; \xi^{(i)})$. For ERM, it is possible to leverage variance reduction techniques [Johnson and Zhang, 2013] to accelerate convergence. For example, the Distributed Accelerated SVRG (DA-SVRG) [Lee et al., 2017] can attain $\varepsilon$-optimality within $\tilde{\mathcal{O}}(\frac{N}{M} \log(1/\varepsilon))$ parallel runtime and $\tilde{\mathcal{O}}(\log(1/\varepsilon))$ rounds of communication. If we were to apply FEDAC for ERM, it can attain expected $\varepsilon$-optimality with $\tilde{\mathcal{O}}(\frac{1}{M\varepsilon})$ parallel runtime and $\tilde{\mathcal{O}}(M^{\frac{1}{3}})$ rounds of communication (assuming Assumption 1 is satisfied). Therefore one can obtain low accuracy solution with FEDAC in a short parallel runtime, whereas DA-SVRG may be preferred if high accuracy is required and $N$ is relatively small. Note that FEDAC is not designed or validated for the distributed ERM setting, and we include this rough comparison for completeness. We conjecture that FEDAC can be incorporated with appropriate variance reduction techniques to attain better performance in federated ERM setting.

## 2 Preliminaries

We conduct our analysis on FEDAC in two settings with two sets of assumptions. The following Assumption 1 consists of a set of standard assumptions: convexity, smoothness and bounded variance. Comparable assumptions are assumed in existing studies on FEDAVG [Haddadpour et al., 2019b, Stich and Karimireddy, 2019, Khaled et al., 2020, Woodworth et al., 2020].[6]

**Assumption 1** ($\mu$-strong convexity, $L$-smoothness and $\sigma^2$-uniformly bounded gradient variance).

(a) *F is $\mu$-strongly convex, i.e.,* $F(u) \geq F(w) + \langle \nabla F(w), u - w \rangle + \frac{1}{2}\mu\|u - w\|^2$ *for any* $u, w \in \mathbb{R}^d$. *In addition, assume F attains a finite optimum* $w^* \in \mathbb{R}^d$. *(We will study both the strongly convex case* ($\mu > 0$) *and the general convex case* ($\mu = 0$)*, which will be clarified in the context.)*

(b) *F is $L$-smooth, i.e.,* $F(u) \leq F(w) + \langle \nabla F(w), u - w \rangle + \frac{1}{2}L\|u - w\|^2$ *for any* $u, w \in \mathbb{R}^d$.

(c) *$\nabla f(w; \xi)$ has $\sigma^2$-bounded variance, i.e.,* $\sup_{w \in \mathbb{R}^d} \mathbb{E}_{\xi \in \mathcal{D}} \|\nabla f(w; \xi) - \nabla F(w)\|^2 \leq \sigma^2$.

The following Assumption 2 consists of an additional set of assumptions: 3rd order smoothness and bounded 4th central moment.

**Assumption 2.** *In addition to Assumption 1, assume that*

(a) *F is Q-3rd-order-smooth, i.e.,* $F(u) \leq F(w) + \langle \nabla F(w), u - w \rangle + \frac{1}{2}\langle \nabla^2 F(w)(u - w), (u - w) \rangle + \frac{1}{6}Q\|u - w\|^3$ *for any* $u, w \in \mathbb{R}^d$.

(b) *$\nabla f(w; \xi)$ has $\sigma^4$-bounded 4th central moment, i.e,* $\sup_{w \in \mathbb{R}^d} \mathbb{E}_{\xi \in \mathcal{D}} \|\nabla f(w; \xi) - \nabla F(w)\|^4 \leq \sigma^4$.

**Notations.** We use $\|\cdot\|$ to denote the operator norm of a matrix or the $\ell_2$-norm of a vector, $[n]$ to denote the set $\{1, 2, \ldots, n\}$. Let $w^*$ be the optimum of $F$ and denote $F^* := F(w^*)$. Let $D_0 := \|w_0 - w^*\|$. For both FEDAC and FEDAVG, we use $M$ to denote the number of parallel workers, $R$ to denote synchronization rounds, $K$ to denote the synchronization interval (i.e., the number of local steps per synchronization round), and $T = KR$ to denote the parallel runtime. We use the subscript to denote timestep, italicized superscript to denote the index of worker and unitalicized superscript "md" or "ag" to denote modifier of iterates in FEDAC (see definition in Algorithm 1). We use overline to denote averaging over all workers, *e.g.*, $\overline{w_t^{\mathrm{ag}}} := \frac{1}{M} \sum_{m=1}^{M} w_t^{\mathrm{ag},m}$. We use $\tilde{\mathcal{O}}, \tilde{\Theta}$ to hide multiplicative polylog factors, which will be clarified in the formal context.

---

**Algorithm 1** Federated Accelerated Stochastic Gradient Descent (FEDAC)

---

1: **procedure** FEDAC($\alpha, \beta, \eta, \gamma$)          ▷ See Eqs. (3.1) and (3.2) for hyperparameter choices
2:      Initialize $w_0^{\mathrm{ag},m} = w_0^m = w_0$ for all $m \in [M]$
3:      **for** $t = 0, \ldots, T-1$ **do**
4:          **for** every worker $m \in [M]$ **in parallel do**
5:              $w_t^{\mathrm{md},m} \leftarrow \beta^{-1} w_t^m + (1 - \beta^{-1}) w_t^{\mathrm{ag},m}$        ▷ Compute $w_t^{\mathrm{md},m}$ by coupling
6:              $g_t^m \leftarrow \nabla f(w_t^{\mathrm{md},m}; \xi_t^m)$           ▷ Query gradient at $w_t^{\mathrm{md},m}$
7:              $v_{t+1}^{\mathrm{ag},m} \leftarrow w_t^{\mathrm{md},m} - \eta \cdot g_t^m$       ▷ Compute next iterate candidate $v_{t+1}^{\mathrm{ag},m}$
8:              $v_{t+1}^m \leftarrow (1 - \alpha^{-1}) w_t^m + \alpha^{-1} w_t^{\mathrm{md},m} - \gamma \cdot g_t^m$ ▷ Compute next iterate candidate $v_{t+1}^m$
9:          **if** sync (i.e., $t \bmod K = -1$) **then**
10:              $w_{t+1}^m \leftarrow \frac{1}{M} \sum_{m'=1}^{M} v_{t+1}^{m'}; \quad w_{t+1}^{\mathrm{ag},m} \leftarrow \frac{1}{M} \sum_{m'=1}^{M} v_{t+1}^{\mathrm{ag},m'}$ ▷ Average & broadcast
11:          **else**
12:              $w_{t+1}^m \leftarrow v_{t+1}^m; \quad w_{t+1}^{\mathrm{ag},m} \leftarrow v_{t+1}^{\mathrm{ag},m}$   ▷ Candidates assigned to be the next iterates

---

# 3 Main results

## 3.1 Main algorithm: Federated Accelerated Stochastic Gradient Descent (FEDAC)

We formally introduce our algorithm FEDAC in Algorithm 1. FEDAC parallelizes a generalized version of Accelerated SGD by Ghadimi and Lan [2012]. In FEDAC, each worker $m \in [M]$ maintains three intertwined sequences $\{w_t^m, w_t^{\mathrm{ag},m}, w_t^{\mathrm{md},m}\}$ at each step $t$. Here $w_t^{\mathrm{ag},m}$ aggregates the past iterates, $w_t^{\mathrm{md},m}$ is the auxiliary sequence of "middle points" on which the gradients are queried, and $w_t^m$ is the main sequence of iterates. At each step, candidate next iterates $v_{t+1}^{\mathrm{ag},m}$ and $v_{t+1}^m$ are computed. If this is a local (unsynchronized) step, they will be assigned to the next iterates $w_{t+1}^{\mathrm{ag},m}$ and $w_{t+1}^{\mathrm{ag},m}$. Otherwise, they will be collected, averaged, and broadcast to all the workers.

**Hyperparameter choice.** We note that the particular version of Accelerated SGD in FEDAC is more flexible than the most standard Nesterov version [Nesterov, 2018], as it has four hyperparameters instead of two. Our analysis suggests that this flexibility seems crucial for principled acceleration in the distributed setting to allow for acceleration-stability trade-off.

However, we note that our theoretical analysis gives a very concrete choice of hyperparameter $\alpha, \beta$, and $\gamma$ in terms of $\eta$. For $\mu$-strongly-convex objectives, we introduce the following two sets of hyperparameter choices, which are referred to as FEDAC-I and FEDAC-II, respectively. As we will see in the Section 3.2.1, under Assumption 1, FEDAC-I has a better dependency on condition number $L/\mu$, whereas FEDAC-II has better communication efficiency.

$$\text{FEDAC-I}: \quad \eta \in \left(0, \frac{1}{L}\right], \quad \gamma = \max\left\{\sqrt{\frac{\eta}{\mu K}}, \eta\right\}, \quad \alpha = \frac{1}{\gamma\mu}, \quad \beta = \alpha + 1; \tag{3.1}$$

$$\text{FEDAC-II}: \quad \eta \in \left(0, \frac{1}{L}\right], \quad \gamma = \max\left\{\sqrt{\frac{\eta}{\mu K}}, \eta\right\}, \quad \alpha = \frac{3}{2\gamma\mu} - \frac{1}{2}, \quad \beta = \frac{2\alpha^2 - 1}{\alpha - 1}. \tag{3.2}$$

Therefore, practically, if the strong convexity estimate $\mu$ is given (which is often taken to be the $\ell_2$ regularization strength), the only hyperparameter to be tuned is $\eta$, whose optimal value depends on the problem parameters.

## 3.2 Theorems on the convergence for strongly convex objectives

Now we present main theorems of FEDAC for strongly convex objectives under Assumption 1 or 2.

### 3.2.1 Convergence of FEDAC under Assumption 1

We first introduce the convergence theorem on FEDAC under Assumption 1. FEDAC-I and FEDAC-II lead to slightly different convergence rates.

**Theorem 3.1** (Convergence of FEDAC). *Let $F$ be $\mu > 0$-strongly convex, and assume Assumption 1.*

*(a) (Full version see Theorem B.1) For $\eta = \min\{\frac{1}{L}, \tilde{\Theta}(\frac{1}{\mu T R})\}$, FEDAC-I yields*

$$\mathbb{E}\left[F(\overline{w_T^{\text{ag}}}) - F^*\right] \le \exp\left(\min\left\{-\frac{\mu T}{L}, -\sqrt{\frac{\mu T R}{L}}\right\}\right) L D_0^2 + \tilde{\mathcal{O}}\left(\frac{\sigma^2}{\mu M T} + \frac{L\sigma^2}{\mu^2 T R^2}\right). \quad (3.3)$$

*(b) (Full version see Theorem C.13) For $\eta = \min\{\frac{1}{L}, \tilde{\Theta}(\frac{1}{\mu T R})\}$, FEDAC-II yields*

$$\mathbb{E}\left[F(\overline{w_T^{\text{ag}}}) - F^*\right] \le \exp\left(\min\left\{-\frac{\mu T}{3L}, -\sqrt{\frac{\mu T R}{9L}}\right\}\right) L D_0^2 + \tilde{\mathcal{O}}\left(\frac{\sigma^2}{\mu M T} + \frac{L^2\sigma^2}{\mu^3 T R^3}\right). \quad (3.4)$$

In comparison, the state-of-the-art FEDAVG analysis [Khaled et al., 2020, Woodworth et al., 2020] reveals the following result.[7]

**Proposition 3.2** (Convergence of FEDAVG under Assumption 1, adapted from Woodworth et al.). *In the settings of Theorem 3.1, for $\eta = \min\{\frac{1}{L}, \tilde{\Theta}(\frac{1}{\mu T})\}$, for appropriate non-negative $\{\rho_t\}_{t=0}^{T-1}$ with $\sum_{t=0}^{T-1} \rho_t = 1$, FEDAVG yields*

$$\mathbb{E}\left[F\left(\sum_{t=0}^{T-1} \rho_t \overline{w_t}\right) - F^*\right] \le \exp\left(-\frac{\mu T}{L}\right) L D_0^2 + \tilde{\mathcal{O}}\left(\frac{\sigma^2}{\mu M T} + \frac{L\sigma^2}{\mu^2 T R}\right). \quad (3.5)$$

**Remark.** *The bound for FEDAC-I (3.3) **asymptotically universally outperforms** FEDAVG (3.5). The first term in (3.3) cooresponds to the deterministic convergence, which is better than the one for FEDAVG. The second term corresponds to the stochasticity of the problem which is not improvable. The third term corresponds to the overhead of infrequent communication, which is also better than FEDAVG due to acceleration. On the other hand, FEDAC-II has better communication efficiency since the third term of (3.4) decays at rate $R^{-3}$.*

### 3.2.2 Convergence of FEDAC under Assumption 2 — faster when close to be quadratic

We establish stronger guarantees for FEDAC-II (3.2) under Assumption 2.

**Theorem 3.3** (Simplified version of Theorem C.1). *Let $F$ be $\mu > 0$-strongly convex, and assume Assumption 2, then for $R \ge \sqrt{\frac{L}{\mu}}$,[8] for $\eta = \min\{\frac{1}{L}, \tilde{\Theta}(\frac{1}{\mu T R})\}$, FEDAC-II yields*

$$\mathbb{E}\left[F(\overline{w_T^{\text{ag}}}) - F^*\right] \le \exp\left(\min\left\{-\frac{\mu T}{3L}, -\sqrt{\frac{\mu T R}{9L}}\right\}\right) 2 L D_0^2 + \tilde{\mathcal{O}}\left(\frac{\sigma^2}{\mu M T} + \frac{Q^2\sigma^4}{\mu^5 T^2 R^6}\right). \quad (3.6)$$

In comparison, we also establish and prove the convergence rate of FEDAVG under Assumption 2.

**Theorem 3.4** (Simplified version of Theorem D.1). *In the settings of Theorem 3.3, for $\eta = \min\left\{\frac{1}{4L}, \tilde{\Theta}\left(\frac{1}{\mu T}\right)\right\}$, for appropriate non-negative $\{\rho_t\}_{t=0}^{T-1}$ with $\sum_{t=0}^{T-1} \rho_t = 1$, FEDAVG yields*

$$\mathbb{E}\left[F\left(\sum_{t=0}^{T-1} \rho_t \overline{w_t}\right) - F^*\right] \le \exp\left(-\frac{\mu T}{8L}\right) 4 L D_0^2 + \tilde{\mathcal{O}}\left(\frac{\sigma^2}{\mu M T} + \frac{Q^2\sigma^4}{\mu^5 T^2 R^2}\right). \quad (3.7)$$

**Remark.** *Our results give a smooth interpolation of the results of [Woodworth et al., 2020] for quadratic objectives to broader function class — the third term regarding infrequent communication overhead will vanish when the objective is quadratic since $Q = 0$. The bound of FEDAC (3.6) outperforms the bound of FEDAVG (3.7) as long as $R \geq \sqrt{L/\mu}$ holds. Particularly in the case of $T \geq M$, our analysis suggests that only $\tilde{\mathcal{O}}(1)$ synchronization are required for linear speedup in $M$. We summarize our results on synchronization bounds and convergence rate in Tables 1 and 2, respectively.*

### 3.3 Convergence for general convex objectives

We also study the convergence of FEDAC for general convex objectives ($\mu = 0$). The idea is to apply FEDAC to $\ell_2$-augmented objective $\tilde{F}_\lambda(w) := F(w) + \frac{\lambda}{2}\|w - w_0\|^2$ as a $\lambda$-strongly-convex and $(L + \lambda)$-smooth objective for appropriate $\lambda$, which is similar to the technique of [Woodworth et al., 2020]. This augmented technique allows us to reuse most of the analysis for strongly-convex objectives. We conjecture that it is possible to construct direct versions of FEDAC for general convex objectives that attain the same rates, which we defer for the future work. We summarize the synchronization bounds in Table 1 and the convergence rates in Table 2. We defer the statement of formal theorems to Section E in Appendix.

## 4 Proof sketch

In this section we sketch the proof for two of our main results, namely Theorem 3.1(a) and 3.3.

### 4.1 Proof sketch of Theorem 3.1(a): FEDAC-I under Assumption 1

Our proof framework consists of the following four steps.

**Step 1: potential-based perturbed iterate analysis.** The first step is to study the difference between FEDAC and its fully synchronized idealization, namely the case of $K = 1$ (recall $K$ denotes the number of local steps). To this end, we extend the perturbed iterate analysis [Mania et al., 2017] to potential-based setting to analyze accelerated convergence. For FEDAC-I, we study the *decentralized* potential $\Psi_t := \frac{1}{M}\sum_{m=1}^{M} F(w_t^{\mathrm{ag},m}) - F^* + \frac{1}{2}\mu\|\overline{w_t} - w^*\|^2$ and establish the following lemma. $\Psi_t$ is adapted from the common potential for acceleration analysis [Bansal and Gupta, 2019].

**Lemma 4.1** (Simplified version of Lemma B.2, Potential-based perturbed iterate analysis for FEDAC-I). *In the same settings of Theorem 3.1(a), the following inequality holds*

$$\mathbb{E}\left[\Psi_T\right] \leq \exp\left(-\gamma\mu T\right)\Psi_0 + \frac{\eta^2 L \sigma^2}{2\gamma\mu} + \frac{\gamma\sigma^2}{2M} \quad \text{(Convergence rate in the case of } K = 1\text{)}$$

$$+ L \cdot \underbrace{\max_{0 \leq t < T} \mathbb{E}\left[\frac{1}{M}\sum_{m=1}^{M}\left\|\overline{w_t^{\mathrm{md}}} - w_t^{\mathrm{md},m}\right\|\left\|\frac{1}{1+\gamma\mu}(\overline{w_t} - w_t^m) + \frac{\gamma\mu}{1+\gamma\mu}(\overline{w_t^{\mathrm{ag}}} - w_t^{\mathrm{ag},m})\right\|\right]}_{\textit{Discrepancy overhead}}.$$

$$(4.1)$$

We refer to the last term of (4.1) as "discrepancy overhead" since it characterizes the dissimilarities among workers due to infrequent synchronization. The proof of Lemma 4.1 is deferred to Section B.2.

**Step 2: bounding discrepancy overhead.** The second step is to bound the discrepancy overhead in (4.1) via stability analysis. Before we look into FEDAC, let us first review the intuition for FEDAVG. There are two forces governing the growth of discrepancy of FEDAVG, namely the (negative) gradient and stochasticity. Thanks to the convexity, the gradient only makes the discrepancy lower. The stochasticity incurs $\mathcal{O}(\eta^2\sigma^2)$ variance per step, so the discrepancy $\mathbb{E}[\frac{1}{M}\sum_{m=1}^{M}\|\overline{w_t} - w_t^m\|^2]$ grows at rate $\mathcal{O}(\eta^2 K\sigma^2)$ linear in $K$. The detailed proof can be found in [Khaled et al., 2020, Woodworth et al., 2020].

For FEDAC, the discrepancy analysis is subtler since acceleration and stability are at odds — the momentum may amplify the discrepancy accumulated from previous steps. Indeed, we establish the following Theorem 4.2, which shows that the *standard deterministic* Accelerated GD (AGD) may *not* be initial-value stable even for strongly convex and smooth objectives, in the sense that initial infinitesimal difference may grow exponentially fast. We defer the formal setup and the proof of Theorem 4.2 to Section F in Appendix.

**Theorem 4.2** (Initial-value instability of deterministic standard AGD). *For any $L, \mu > 0$ such that $^L/_\mu \geq 25$, and for any $K \geq 1$, there exists a 1D objective $F$ that is $L$-smooth and $\mu$-strongly-convex, and an $\varepsilon_0 > 0$, such that for any positive $\varepsilon < \varepsilon_0$, there exists initialization $w_0, u_0, w_0^{\mathrm{ag}}, u_0^{\mathrm{ag}}$ such that $|w_0 - u_0| \leq \varepsilon$, $|w_0^{\mathrm{ag}} - u_0^{\mathrm{ag}}| \leq \varepsilon$, but the trajectories $\{w_t^{\mathrm{ag}}, w_t^{\mathrm{md}}, w_t\}_{t=0}^{3K}$, $\{u_t^{\mathrm{ag}}, u_t^{\mathrm{md}}, u_t\}_{t=0}^{3K}$ generated by applying deterministic AGD with initialization $(w_0, w_0^{\mathrm{ag}})$ and $(u_0, u_0^{\mathrm{ag}})$ satisfies*

$$|w_{3K} - u_{3K}| \geq \frac{1}{2}\varepsilon(1.02)^K, \qquad |w_{3K}^{\mathrm{ag}} - u_{3K}^{\mathrm{ag}}| \geq \varepsilon(1.02)^K.$$

Fortunately, we can show that the discrepancy can grow at a slower exponential rate via less aggressive acceleration, see Lemma 4.3. As we will discuss shortly, we adjust $\gamma$ according to $K$ to restrain the growth of discrepancy within the linear regime. The proof of Lemma 4.3 is deferred to Section B.3.

**Lemma 4.3** (Simplified version of Lemma B.3, Discrepancy overhead bounds for FEDAC-I). *In the same setting of Theorem 3.1(a), the following inequality holds*

$$\text{``Discrepancy overhead'' in Eq. (4.1)} \leq \begin{cases} 7\eta\gamma LK\sigma^2 \left(1 + \frac{2\gamma^2\mu}{\eta}\right)^{2K} & \text{if } \gamma \in (\eta, \sqrt{\frac{\eta}{\mu}}], \\ 7\eta^2 LK\sigma^2 & \text{if } \gamma = \eta. \end{cases}$$

**Step 3: trading-off acceleration and discrepancy.** Combining Lemmas 4.1 and 4.3 gives

$$\mathbb{E}[\Psi_T] \leq \underbrace{\exp\left(-\gamma\mu T\right)\Psi_0}_{(\mathrm{I})} + \frac{\eta^2 L\sigma^2}{2\gamma\mu} + \frac{\gamma\sigma^2}{2M} + \underbrace{\begin{cases} 7\eta\gamma LK\sigma^2 \left(1 + \frac{2\gamma^2\mu}{\eta}\right)^{2K} & \text{if } \gamma \in (\eta, \sqrt{\frac{\eta}{\mu}}], \\ 7\eta^2 LK\sigma^2 & \text{if } \gamma = \eta. \end{cases}}_{(\mathrm{II})} \quad (4.2)$$

The value of $\gamma \in [\eta, \sqrt{\eta/\mu}]$ controls the magnitude of acceleration in (I) and discrepancy growth in (II). The upper bound choice $\sqrt{\eta/\mu}$ gives full acceleration in (I) but makes (II) grow exponentially in $K$. On the other hand, the lower bound choice $\eta$ makes (II) linear in $K$ but loses all acceleration. We wish to attain as much acceleration in (I) as possible while keeping the discrepancy (II) grow moderately. Our balanced solution is to pick $\gamma = \max\{\sqrt{\eta/(\mu K)}, \eta\}$. One can verify that the discrepancy grows (at most) linearly in $K$. Substituting this choice of $\gamma$ to Eq. (4.2) leads to

$$\mathbb{E}[\Psi_T] \leq \underbrace{\exp\left(\min\left\{-\eta\mu T, -\frac{\eta^{\frac{1}{2}}\mu^{\frac{1}{2}}T}{K^{\frac{1}{2}}}\right\}\right)\Psi_0}_{\text{Monotonically decreasing } \varphi_\downarrow(\eta)} + \underbrace{\mathcal{O}\left(\frac{\eta^{\frac{1}{2}}\sigma^2}{\mu^{\frac{1}{2}}MK^{\frac{1}{2}}} + \frac{\eta\sigma^2}{M} + \frac{\eta^{\frac{3}{2}}LK^{\frac{1}{2}}\sigma^2}{\mu^{\frac{1}{2}}} + \eta^2 LK\sigma^2\right)}_{\text{Monotonically increasing } \varphi_\uparrow(\eta)}.$$

$$(4.3)$$

**Step 4: finding $\eta$ to optimize the RHS of Eq. (4.3).** It remains to show that (4.3) gives the desired bound with our choice of $\eta = \min\{\frac{1}{L}, \tilde{\Theta}(\frac{K}{\mu T^2})\}$. The increasing $\varphi_\uparrow(\eta)$ in (4.3) is bounded by $\tilde{\mathcal{O}}(\frac{\sigma^2}{\mu MT} + \frac{LK^2\sigma^2}{\mu^2 T^3})$. The decreasing term $\varphi_\downarrow(\eta)$ in (4.3) is bounded by $\varphi_\downarrow(\frac{1}{L}) + \varphi_\downarrow(\tilde{\Theta}(\frac{K}{\mu T^2}))$, where $\varphi_\downarrow(\frac{1}{L}) = \exp(\min\{-\frac{\mu T}{L}, -\frac{\mu^{\frac{1}{2}}T}{L^{\frac{1}{2}}K^{\frac{1}{2}}}\})$, and $\varphi_\downarrow(\tilde{\Theta}(\frac{K}{\mu T^2}))$ can be controlled by the bound of $\varphi_\uparrow(\eta)$ provided $\tilde{\Theta}$ has appropriate polylog factors. Replacing $K$ with $^T/_R$ completes the proof of Theorem 3.1(a). We defer the details to Section B.

### 4.2 Proof sketch of Theorem 3.3: convergence of FEDAC-II under Assumption 2

In this section, we outline the proof of Theorem 3.3 by explaining the differences with the proof in Section 4.1. The first difference is that for FEDAC-II we study an alternative *centralized potential* $\Phi_t = F(\overline{w_t^{\mathrm{ag}}}) - F^* + \frac{1}{6}\mu\|\overline{w_t} - w^*\|^2$, which leads to an alternative version of Lemma 4.1 as follows.

$$\mathbb{E}[\Phi_T] \leq \exp\left(-\frac{\gamma\mu T}{3}\right)\Phi_0 + \frac{3\eta^2 L\sigma^2}{2\gamma\mu M} + \frac{\gamma\sigma^2}{2M} + \frac{3}{\mu}\max_{0 \leq t < T}\mathbb{E}\left\|\frac{1}{M}\sum_{m=1}^{M}\nabla F(w_t^{\mathrm{md},m}) - \nabla F(\overline{w_t^{\mathrm{md}}})\right\|^2. \quad (4.4)$$

The second difference is that the particular discrepancy in (4.4) can be bounded via 3rd-order smoothness $Q$ since $\|\frac{1}{M}\sum_{m=1}^{M}\nabla F(w_t^{\mathrm{md},m}) - \nabla F(\overline{w_t^{\mathrm{md}}})\|^2 \leq \frac{Q^2}{4M}\sum_{m=1}^{M}\|w^{\mathrm{md},m} - \overline{w_t^{\mathrm{md}}}\|^4$. The proof then follows by analyzing the 4th-order stability of FEDAC. We defer the details to Section C.

# 5 Numerical experiments

In this section, we validate our theory and demonstrate the efficiency of FEDAC via experiments.[9] The performance of FEDAC is tested against FEDAVG (a.k.a., Local SGD), (distributed) Minibatch-SGD (MB-SGD) and Minibatch-Accelerated-SGD (MB-AC-SGD) [Dekel et al., 2012, Cotter et al., 2011] on $\ell_2$-regularized logistic regression for UCI a9a dataset [Dua and Graff, 2017] from LibSVM [Chang and Lin, 2011]. The regularization strength is set as $10^{-3}$. The hyperparameters $(\gamma, \alpha, \beta)$ of FEDAC follows FEDAC-I where strong-convexity $\mu$ is chosen as regularization strength $10^{-3}$. We test the settings of $M = 2^2, \ldots, 2^{13}$ workers and $K = 2^0, \ldots, 2^8$ synchronization interval. For all four algorithms, we tune the learning-rate $\eta$ *only* from the same set of levels within $[10^{-3}, 10]$. We choose $\eta$ based on the best suboptimality. We claim that the best $\eta$ lies in the range $[10^{-3}, 10]$ for all algorithms under all settings. We defer the rest of setup details to Section A. In Fig. 1, we compare the four algorithms by measuring the effect of linear speedup under variant $K$.

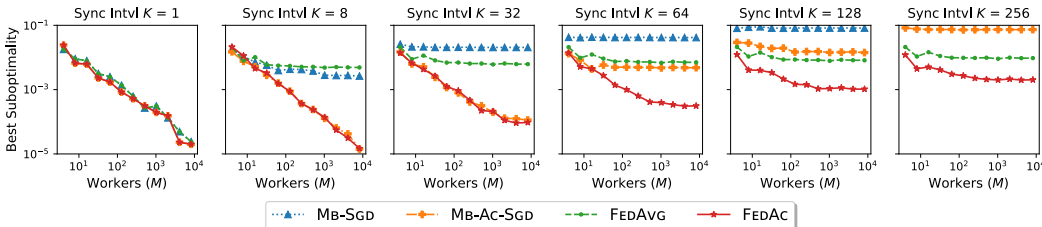

Figure 1: **Observed linear speedup with respect to the number of workers $M$ under various synchronization intervals $K$.** Our FEDAC is tested against three baselines FEDAVG, MB-SGD, and MB-AC-SGD. While all four algorithms attain linear speedup for the fully synchronized ($K = 1$) setting, FEDAVG and MB-SGD lose linear speedup for $K$ as low as 8. MB-AC-SGD is comparably better than the other two baselines but still deteriorates significantly for $K \geq 64$. FEDAC is most robust to infrequent synchronization and outperforms the baselines by a margin for $K \geq 64$.

In the next experiments, we provide an empirical example to show that the direct parallelization of standard accelerated SGD may indeed suffer from instability. This complements our Theorem 4.2 (or full version Theorem F.1) on the initial-value instability of standard AGD. Recall that FEDAC-I Eq. (3.1) and FEDAC-II Eq. (3.2) adopt an acceleration-stability tradeoff technique that takes $\gamma = \max\left\{\sqrt{\frac{\eta}{\mu K}}, \eta\right\}$. Formally, we denote the following direct acceleration of FEDAC without such tradeoff as "vanilla FEDAC": $\eta \in (0, \frac{1}{L}], \gamma = \sqrt{\frac{\eta}{\mu}}, \alpha = \frac{1}{\gamma\mu}, \beta = \alpha + 1$. In Fig. 2, the vanilla FEDAC is compared with (stable) FEDAC-I and the baseline MB-AC-SGD. We test on the UCI "adult" a9a dataset with $\ell_2$-regularization strength $\lambda$ taken to be $10^{-4}$. We test the settings of $M = 2^4, \ldots, 2^{13}$ and $K = 2^0, \ldots, 2^8$. $\eta$ is tuned from $[0.001, 5]$ and the best $\eta$ lies in this range for all algorithms under all settings. The results show that the vanilla FEDAC is consistently worse than the (stable) FEDAC-I when $K$ is large.

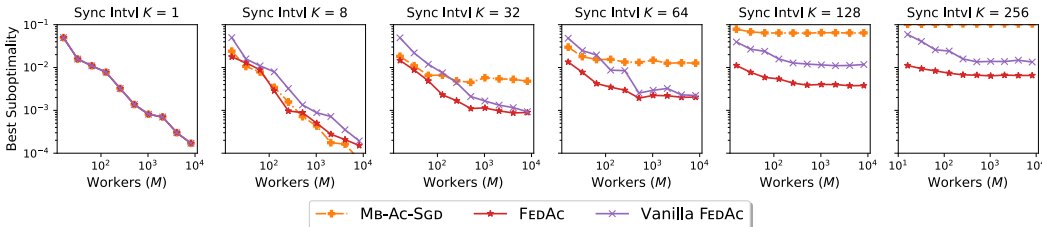

Figure 2: **Vanilla FEDAC versus (stable) FEDAC-I and baseline MB-AC-SGD on the observed linear speedup w.r.t. $M$ under various synchronization intervals $K$.** Observet that Vanilla FEDAC is indeed less robust to infrequent synchronization and thus worse than the (stable) FEDAC-I.

We include more experiments on various dataset, and more detailed analysis in Section A.

## Broader Impact

This work proposes FEDAC, a principled acceleration of FEDAVG, which provably improves convergence speed and communication efficiency. Our theory and experiments suggest that FEDAC saves computational resources and reduces communication overhead, especially in the setting of abundant workers and infrequent communication. Our analysis could promote a better understanding of federated / distributed optimization and acceleration theory. We expect FEDAC could be generalized to broader settings, *e.g.*, non-convex objective and/or heterogenous workers.

The opportunity for privacy-preserving learning is another advantage of Federated Learning beyond parallelization, since the user data are kept local during learning. While we do not analyze the privacy guarantee in this work, we conjecture that FEDAC could potentially enjoy better privacy-preserving property since less communication is required to achieve the same accuracy. However, this intuition should be applied with caution for high-risk data until theoretical privacy guarantee is established.

## Acknowledgements and Disclosure of Funding

Honglin Yuan would like to thank the support by the Total Innovation Fellowship. Tengyu Ma would like to thank the support by the Google Faculty Award. The work is also partially supported by SDSI and SAIL. We would like to thank Qian Li, Junzi Zhang, and Yining Chen for helpful discussions at various stages of this work. We would like to thank the anonymous reviewers for their suggestions and comments.

## Footnotes

[1]In the literature, FEDAVG usually runs on a randomly sampled subset of heterogeneous workers for each synchronization round, whereas Local SGD or Parallel SGD usually run on a fixed set of workers. In this paper we do not differentiate the terminology and assumed a fixed set of workers are deployed for simplicity.

[2]We hide varaibles other than $T, M, R$ for simplicity. The complete bound can be found in Table 2 and the corresponding theorems.

[3]"Synchronization required for linear speedup" is a simple and common measure of the communication efficiency, which can be derived from the raw convergence rate. It is defined as the minimum number of synchronization $R$, as a function of number of workers $M$ and parallel runtime $T$, required to achieve a linear speed up — the parallel runtime of $M$ workers is equal to the $^1/_M$ fraction of a sequential single worker runtime.

[4]Minibatch-SGD baseline corresponds to running SGD for $R$ steps with batch size $MT/R$, which can be implemented on $M$ parallel workers with $R$ communication and each worker queries $T$ gradients in total.

[5]We construct the counterexample for initial-value stability for simplicity and clarity. We conjecture that our counterexample also extends to other algorithmic stability notions (*e.g.*, uniform stability [Bousquet and Elisseeff, 2002]) since initial-value stability is usually milder than the others.

[6]In fact, Woodworth et al. [2020] imposes the same assumption in Assumption 1; Khaled et al. [2020] assumes $f(w; \xi)$ are convex and smooth for all $\xi$, which is more restricted; Stich and Karimireddy [2019] assumes quasi-convexity instead of convexity; Haddadpour et al. [2019b] assumes P-Ł condition instead of strong convexity. In this work we focus on standard (general or strong) convexity to simplify the analysis.

[7]Proposition 3.2 can be (easily) adapted from the Theorem 2 of [Woodworth et al., 2020] which analyzes a decaying learning rate with convergence rate $\mathcal{O}\left(\frac{L^2 D_0^2}{\mu T^2} + \frac{\sigma^2}{\mu M T}\right) + \tilde{\mathcal{O}}\left(\frac{L\sigma^2}{\mu^2 T R}\right)$. This bound has no log factor attached to $\frac{\sigma^2}{\mu M T}$ term but worse (polynomial) dependency on initial state $D_0$ than Proposition 3.2. We present Proposition 3.2 for consistency and the ease of comparison.

[8]The assumption $R \ge \sqrt{L/\mu}$ is removed in the full version (Theorem C.1).

[9]Code repository link: https://github.com/hongliny/FedAc-NeurIPS20.

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
