[Supplementary Material]

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

[10]FEDAC-II is qualitatively similar to FEDAC-I empirically so we show FEDAC-I only.

[11]We search for this range to guarantee that the optimal $\eta$ lies in this range for all algorithms and all settings. One could save effort in tuning if only one algorithm were implemented.

[12]Note that we state our full Theorem B.1 in terms of the synchronization gap $K$ instead of the synchronization round $R$ as in the simplified Theorem 3.1(a). This two quantities are trivially related as $T = KR$. In fact, our bound Theorem B.1 in terms of $K$ also holds for irregular synchronization setting as long as the maximum synchronization interval is bounded by $K$.

[13]Throughout this paper we do not optimize the polylog factors or the constants. We conjecture that certain polylog factors can be improved or removed via averaging techniques such as [Lacoste-Julien et al., 2012, Stich, 2019b].

[14]We assume this constant lower bound for technical simplification.

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

# Appendices

The appendices are structured as follows. In Section A, we include additional experiments with description of setup details. In Sections B and C, we prove the complete version of Theorems 3.1 and 3.3 on the convergence of FEDAC under Assumption 1 or 2. In Section D, we prove Theorem 3.4 on the convergence of FEDAVG under Assumption 2. In Section E, we prove the convergence of FEDAC (and FEDAVG) for general convex objectives. In Section F, we prove Theorem 4.2 on the initial-value instability of standard accelerated gradient descent. We include some helper lemmas in Section G.

## List of appendices

# A  Additional experiments and setup details

## A.1  Additional setup details

**Baselines.** FEDAC is tested against three baselines, namely FEDAVG (a.k.a., Local SGD), (distributed) Minibatch-SGD (MB-SGD), and (distributed) Minibatch-Accelerated-SGD (MB-AC-SGD) [Dekel et al., 2012, Cotter et al., 2011]. We fix the parallel runtime $T = 4096$, and test variant levels of synchronization interval $K$ and parallel workers $M$. MB-SGD and MB-AC-SGD baselines correspond to running SGD or accelerated SGD for $T/K$ steps with batch size $MK$. The comparison is fair since all algorithms can be parallelized to $M$ workers with $T/K$ rounds of communication where each worker queries $T$ gradients in total. We simulate the parallelization with a `NumPy` program on a local CPU cluster. We start from the same random initialization for all algorithms under all settings.

**Datasets.** The algorithms are tested on $\ell_2$-regularized logistic regression on the following two binary classification datasets from `LibSVM`. The preprocessing information and the download links can be found at `https://www.csie.ntu.edu.tw/~cjlin/libsvmtools/datasets/binary.html`.

1. The "adult" `a9a` dataset with 123 features and 32,561 training samples from the UCI Machine Learning Repository [Dua and Graff, 2017].

2. The `epsilon` dataset with 2,000 features and 400,000 training samples from the PASCAL Challenge 2008 [Sonnenburg et al., 2008].

**Evaluation.** For all algorithms and all settings, we evaluate the population loss every $512$ parallel timesteps (gradient queries). We compute the suboptimality by comparing with a pre-computed optimum $F^*$. We record the best suboptimality attained over the evaluations.

**Hyperparameter choice.** For all four algorithms, we tune the "learning-rate" hyperparameter $\eta$ only and record the best suboptimality attained. For MB-AC-SGD, the rest of hyperparameters are determined by the strong-convexity estimate $\mu$ which is taken to be the $\ell_2$-regularization strength $\lambda$. For FEDAC, the default choice is FEDAC-I Eq. (3.1),[10] where the strong-convexity estimate $\mu$ is also taken to be the $\ell_2$-regularization strength $\lambda$.

## A.2  Results on dataset `a9a`

We first test on the `a9a` dataset with $\ell_2$-regularization strength $10^{-3}$. We test the setting of $K = 2^0, \ldots, 2^8$ and $M = 2^2, \ldots, 2^{13}$. For all algorithms, we tune $\eta$ from the same sets: $\{0.001, 0.002, 0.005, 0.01, 0.02, 0.05, 0.1, 0.2, 0.5, 1, 2, 5, 10\}$. We claim that the best $\eta$ lies in $[0.001, 10]$ for all algorithms for all settings.[11] We plot the observed linear speedup figure in Fig. 1 in the main body. To better understand the dependency on synchronization intervals $K$, we plot the following Fig. 3. The results suggest that FEDAC is more robust to infrequent synchronization and thus more communication-efficient. For example, when using 8192 workers, FEDAC requries only 32 rounds of communication to attain $10^{-3}$ suboptimality, whereas MB-AC-SGD, MB-SGD and FEDAVG require 128, 1024, 4096 rounds, respectively.

Figure 3: **FEDAC versus baselines on the dependency of synchronization interval $K$ under various workers $M$.** For all tested $M$, FEDAVG and MB-SGD start to deteriorate once $K$ passes 2; MB-AC-SGD is more robust to moderate $K$ than FEDAVG and MB-SGD but sharply deteriorate once it passes a threshold at around $K = 32$. This is because MB-AC-SGD does not have enough gradient steps for convergence when the communication is too sparse. In comparison, FEDAC is more robust to infrequent communication. Dataset: `a9a`, $\ell_2$-regularization strength: $10^{-3}$.

We repeat the experiments with an alternative choice of $\lambda = 10^{-2}$. This problem is relatively "easier" in terms of optimization since the condition number $L/\mu$ is lower. We test the same levels of $M$, $K$ and tune the $\eta$ from the same set as above. The results are shown in Figs. 4 and 5. The results are qualitatively similar to the $\lambda = 10^{-3}$ case. For $K \leq 64$, the performance of FEDAC and MB-AC-SGD are similar, which both outperform the other two baselines FEDAVG and MB-SGD. For $K \geq 128$, the MB-AC-SGD drastically worsen because the gradient steps are too few, and FEDAC outperforms the other baselines by a margin.

Figure 4: **FEDAC versus baselines on the observed linear speedup w.r.t $M$ under various synchronization interval $K$.** The results are qualitatively similar to Fig. 1. Dataset: `a9a`, $\ell_2$-regularization strength: $10^{-2}$.

Figure 5: **FEDAC versus baselines on the dependency of synchronization interval $K$ under various workers $M$.** The results are qualitatively similar to Fig. 3. Dataset: `a9a`, $\ell_2$-regularization strength: $10^{-2}$.

### A.3 Results on dataset `epsilon`

In this section we repeat the experiments above on the larger `epsilon` dataset with $\ell_2$-regularization $\lambda$ taken to be $10^{-4}$. $\eta$ is tuned from $\{0.005, 0.01, 0.02, 0.05, 0.1, 0.2, 0.5, 1, 2, 5, 10, 20, 50\}$. The optimal $\eta$ lies in the corresponding range for all algorithm under all tested settings. The results are shown in Figs. 6 and 7. The results are qualitatively similar to the previous experiments on `a9a` dataset. FEDAC is more communication-efficient than the baselines. For example, when using 2048 workers, FEDAC requires only 64 rounds of communication (synchronization) to attain $10^{-4}$ suboptimality, whereas MB-AC-SGD, MB-SGD and FEDAVG require 256, 4096 and 4096 rounds of communication, respectively.

Figure 6: **FEDAC versus baselines on the observed linear speedup w.r.t $M$ under various synchronization interval $K$.** The results are qualitatively similar to Fig. 1. Dataset: `epsilon`, $\ell_2$-regularization strength: $10^{-4}$.

Figure 7: **FEDAC versus baselines on the dependency of synchronization interval $K$ under various workers $M$.** The results are qualitatively similar to Fig. 3. Dataset: `epsilon`, $\ell_2$-regularization strength: $10^{-4}$.

# B  Analysis of FEDAC-I under Assumption 1

In this section we study the convergence of FEDAC-I. We provide a complete, non-asymptotic version of Theorem 3.1(a) on the convergence of FEDAC-I under Assumption 1 and provide the detailed proof, which expands the proof sketch in Section 4.1. Recall that FEDAC-I is defined as the FEDAC (Algorithm 1) with the following hyperparameters choice

$$\eta \in \left(0, \frac{1}{L}\right], \quad \gamma = \max\left\{\sqrt{\frac{\eta}{\mu K}}, \eta\right\}, \quad \alpha = \frac{1}{\gamma\mu}, \quad \beta = \alpha + 1. \qquad \text{(FEDAC-I)}$$

We keep track of the convergence progress of FEDAC-I via the following decentralized potential $\Psi_t$.

$$\Psi_t := \frac{1}{M} \sum_{m=1}^{M} F(w_t^{\text{ag},m}) - F^* + \frac{1}{2}\mu\|\overline{w_t} - w^*\|^2. \qquad \text{(B.1)}$$

Recall $\overline{w_t}$ is defined as $\frac{1}{M} \sum_{m=1}^{M} w_t^m$. Formally, we use $\mathcal{F}_t$ to denote the $\sigma$-algebra generated by $\{w_\tau^m, w_\tau^{\text{ag},m}\}_{\tau \le t, m \in [M]}$. Since FEDAC is Markovian, conditioning on $\mathcal{F}_t$ is equivalent to conditioning on $\{w_t^m, w_t^{\text{ag},m}\}_{m \in [M]}$.

## B.1  Main theorem and lemmas: Complete version of Theorem 3.1(a)

Now we introduce the main theorem on the convergence of FEDAC-I. [12] [13]

**Theorem B.1** (Convergence of FEDAC-I, complete version of Theorem 3.1(a))**.** *Let $F$ be $\mu > 0$-strongly convex, and assume Assumption 1, then for*

$$\eta = \min\left\{\frac{1}{L}, \frac{K}{\mu T^2} \log^2\left(\text{e} + \min\left\{\frac{\mu MT\Psi_0}{\sigma^2}, \frac{\mu^2 T^3\Psi_0}{LK^2\sigma^2}\right\}\right)\right\},$$

FEDAC-I *yields*

$$\mathbb{E}[\Psi_T] \le \min\left\{\exp\left(-\frac{\mu T}{L}\right), \exp\left(-\frac{\mu^{\frac{1}{2}}T}{L^{\frac{1}{2}}K^{\frac{1}{2}}}\right)\right\}\Psi_0$$

$$+ \frac{2\sigma^2}{\mu MT} \log^2\left(\text{e} + \frac{\mu MT\Psi_0}{\sigma^2}\right) + \frac{400LK^2\sigma^2}{\mu^2 T^3} \log^4\left(\text{e} + \frac{\mu^2 T^3\Psi_0}{LK^2\sigma^2}\right),$$

*where $\Psi_t$ is the decentralized potential defined in Eq.* (B.1).

**Remark.** *The simplified version Theorem 3.1(a) in the main body can be obtained by replacing $K$ with $T/R$ and upper bound $\Psi_0$ by $LD_0^2$.*

The proof of Theorem B.1 is based on the following two lemmas regarding convergence and stability respectively. To clarify the hyperparameter dependency, we state these lemmas for general $\gamma \in \left[\eta, \sqrt{\frac{\eta}{\mu}}\right]$, which has one more degree of freedom than FEDAC-I where $\gamma = \max\left\{\sqrt{\frac{\eta}{\mu K}}, \eta\right\}$ is fixed.

**Lemma B.2** (Potential-based perturbed iterate analysis for FEDAC-I). *Let $F$ be $\mu > 0$-strongly convex, and assume Assumption 1, then for $\alpha = \frac{1}{\gamma\mu}$, $\beta = \alpha + 1$, $\gamma \in \left[\eta, \sqrt{\frac{\eta}{\mu}}\right]$, $\eta \in \left(0, \frac{1}{L}\right]$, FEDAC yields*

$$\mathbb{E}[\Psi_T] \leq \exp\left(-\gamma\mu T\right)\Psi_0 + \frac{\eta^2 L \sigma^2}{2\gamma\mu} + \frac{\gamma\sigma^2}{2M}$$

$$+ L \cdot \max_{0 \leq t < T} \mathbb{E}\left[\frac{1}{M}\sum_{m=1}^{M}\left\|\overline{w_t^{\mathrm{md}}} - w_t^{\mathrm{md},m}\right\|\left\|\frac{1}{1+\gamma\mu}(\overline{w_t} - w_t^m) + \frac{\gamma\mu}{1+\gamma\mu}(\overline{w_t^{\mathrm{ag}}} - w_t^{\mathrm{ag},m})\right\|\right],$$

*where $\Psi_t$ is the decentralized potential defined in Eq.* (B.1).

The proof of Lemma B.2 is deferred to Section B.2.

**Lemma B.3** (Discrepancy overhead bound). *In the same setting of Lemma B.2,* FEDAC *satisfies*

$$\mathbb{E}\left[\frac{1}{M}\sum_{m=1}^{M}\left\|\overline{w_t^{\mathrm{md}}} - w_t^{\mathrm{md},m}\right\|\left\|\frac{1}{1+\gamma\mu}(\overline{w_t} - w_t^m) + \frac{\gamma\mu}{1+\gamma\mu}(\overline{w_t^{\mathrm{ag}}} - w_t^{\mathrm{ag},m})\right\|\right]$$

$$\leq \begin{cases} 7\eta\gamma K\sigma^2\left(1 + \frac{2\gamma^2\mu}{\eta}\right)^{2K} & \text{if } \gamma \in \left(\eta, \sqrt{\frac{\eta}{\mu}}\right], \\ 7\eta^2 K\sigma^2 & \text{if } \gamma = \eta. \end{cases}$$

The proof of Lemma B.3 is deferred to Section B.3.

Now we plug in the choice of $\gamma = \max\left\{\sqrt{\frac{\eta}{\mu K}}, \eta\right\}$ to Lemmas B.2 and B.3, which leads to the following lemma.

**Lemma B.4** (Convergence of FEDAC-I for general $\eta$). *Let $F$ be $\mu > 0$-strongly convex, and assume Assumption 1, then for any $\eta \in \left(0, \frac{1}{L}\right]$,* FEDAC-I *yields*

$$\mathbb{E}[\Psi_T] \leq \exp\left(-\max\left\{\eta\mu, \sqrt{\frac{\eta\mu}{K}}\right\}T\right)\Psi_0 + \frac{\eta^{\frac{1}{2}}\sigma^2}{2\mu^{\frac{1}{2}}MK^{\frac{1}{2}}} + \frac{\eta\sigma^2}{2M} + \frac{390\eta^{\frac{3}{2}}LK^{\frac{1}{2}}\sigma^2}{\mu^{\frac{1}{2}}} + 7\eta^2 LK\sigma^2,$$
(B.2)

*where $\Psi_t$ is the decentralized potential defined in Eq.* (B.1).

*Proof of Lemma B.4.* It is direct to verify that $\gamma = \max\left\{\eta, \sqrt{\frac{\eta}{\mu K}}\right\} \in \left[\eta, \sqrt{\frac{\eta}{\mu}}\right]$ so both Lemmas B.2 and B.3 are applicable. Applying Lemma B.2 yields

$$\mathbb{E}[\Psi_T] \leq \exp\left(-\max\left\{\eta\mu, \sqrt{\frac{\eta\mu}{K}}\right\}T\right)\Psi_0 + \min\left\{\frac{\eta L \sigma^2}{2\mu}, \frac{\eta^{\frac{3}{2}}LK^{\frac{1}{2}}\sigma^2}{2\mu^{\frac{1}{2}}}\right\} + \max\left\{\frac{\eta\sigma^2}{2M}, \frac{\eta^{\frac{1}{2}}\sigma^2}{2\mu^{\frac{1}{2}}MK^{\frac{1}{2}}}\right\}$$

$$+ L \cdot \max_{0 \leq t < T} \mathbb{E}\left[\frac{1}{M}\sum_{m=1}^{M}\left\|\overline{w_t^{\mathrm{md}}} - w_t^{\mathrm{md},m}\right\|\left\|\frac{1}{1+\gamma\mu}(\overline{w_t} - w_t^m) + \frac{\gamma\mu}{1+\gamma\mu}(\overline{w_t^{\mathrm{ag}}} - w_t^{\mathrm{ag},m})\right\|\right].$$
(B.3)

We bound $\max\left\{\frac{\eta\sigma^2}{2M}, \frac{\eta^{\frac{1}{2}}\sigma^2}{2\mu^{\frac{1}{2}}MK^{\frac{1}{2}}}\right\}$ by $\frac{\eta\sigma^2}{2M} + \frac{\eta^{\frac{1}{2}}\sigma^2}{2\mu^{\frac{1}{2}}MK^{\frac{1}{2}}}$, and bound $\min\left\{\frac{\eta L \sigma^2}{2\mu}, \frac{\eta^{\frac{3}{2}}LK^{\frac{1}{2}}\sigma^2}{2\mu^{\frac{1}{2}}}\right\}$ by $\frac{\eta^{\frac{3}{2}}LK^{\frac{1}{2}}\sigma^2}{2\mu^{\frac{1}{2}}}$, which gives

$$\min\left\{\frac{\eta L \sigma^2}{2\mu}, \frac{\eta^{\frac{3}{2}}LK^{\frac{1}{2}}\sigma^2}{2\mu^{\frac{1}{2}}}\right\} + \max\left\{\frac{\eta\sigma^2}{2M}, \frac{\eta^{\frac{1}{2}}\sigma^2}{2\mu^{\frac{1}{2}}MK^{\frac{1}{2}}}\right\} \leq \frac{\eta^{\frac{3}{2}}LK^{\frac{1}{2}}\sigma^2}{2\mu^{\frac{1}{2}}} + \frac{\eta\sigma^2}{2M} + \frac{\eta^{\frac{1}{2}}\sigma^2}{2\mu^{\frac{1}{2}}MK^{\frac{1}{2}}}.$$
(B.4)

Applying Lemma B.3 with $\gamma = \max\left\{\eta, \sqrt{\frac{\eta}{\mu K}}\right\}$ gives

$$\mathbb{E}\left[\frac{1}{M}\sum_{m=1}^{M}\left\|\overline{w_t^{\mathrm{md}}} - w_t^{\mathrm{md},m}\right\|\left\|\frac{1}{1+\gamma\mu}(\overline{w_t} - w_t^m) + \frac{\gamma\mu}{1+\gamma\mu}(\overline{w_t^{\mathrm{ag}}} - w_t^{\mathrm{ag},m})\right\|\right]$$

$$\leq \begin{cases} 7\eta\sqrt{\frac{\eta}{\mu K}}K\sigma^2\left(1+\frac{2}{K}\right)^{2K} & \text{if } \gamma = \sqrt{\frac{\eta}{\mu K}} \\ 7\eta^2 K\sigma^2 & \text{if } \gamma = \eta \end{cases}$$

$$\leq \frac{7\mathrm{e}^4\eta^{\frac{3}{2}}K^{\frac{1}{2}}\sigma^2}{\mu^{\frac{1}{2}}} + 7\eta^2 K\sigma^2. \tag{B.5}$$

Combining Eqs. (B.3), (B.4) and (B.5) yields

$$\mathbb{E}[\Psi_T] \leq \exp\left(-\max\left\{\eta\mu, \sqrt{\frac{\eta\mu}{K}}\right\}T\right)\Psi_0 + \frac{\eta^{\frac{1}{2}}\sigma^2}{2\mu^{\frac{1}{2}}MK^{\frac{1}{2}}} + \frac{\eta\sigma^2}{2M} + \frac{(7\mathrm{e}^4+\frac{1}{2})\eta^{\frac{3}{2}}LK^{\frac{1}{2}}\sigma^2}{\mu^{\frac{1}{2}}} + 7\eta^2 LK\sigma^2.$$

The lemma then follows by leveraging the estimate $7\mathrm{e}^4 + \frac{1}{2} < 390$ for the coefficient of $\frac{\eta^{\frac{3}{2}}LK^{\frac{1}{2}}\sigma^2}{\mu^{\frac{1}{2}}}$.

$\square$

The main Theorem B.1 then follows by plugging an appropriate $\eta$ to Lemma B.4.

*Proof of Theorem B.1.* To simplify the notation, we denote the decreasing term in Eq. (B.2) as $\varphi_\downarrow(\eta)$ and the increasing term as $\varphi_\uparrow(\eta)$, namely

$$\varphi_\downarrow(\eta) := \exp\left(-\max\left\{\eta\mu, \sqrt{\frac{\eta\mu}{K}}\right\}T\right)\Psi_0, \quad \varphi_\uparrow(\eta) := \frac{\eta^{\frac{1}{2}}\sigma^2}{2\mu^{\frac{1}{2}}MK^{\frac{1}{2}}} + \frac{\eta\sigma^2}{2M} + \frac{390\eta^{\frac{3}{2}}LK^{\frac{1}{2}}\sigma^2}{\mu^{\frac{1}{2}}} + 7\eta^2 LK\sigma^2.$$

Now let

$$\eta_0 := \frac{K}{\mu T^2}\log^2\left(\mathrm{e} + \min\left\{\frac{\mu MT\Psi_0}{\sigma^2}, \frac{\mu^2 T^3\Psi_0}{LK^2\sigma^2}\right\}\right),$$

and then $\eta = \min\left\{\frac{1}{L}, \eta_0\right\}$. Therefore, the decreasing term $\varphi_\downarrow(\eta)$ is upper bounded by $\varphi_\downarrow(\frac{1}{L}) + \varphi_\downarrow(\eta_0)$, where

$$\varphi_\downarrow\left(\frac{1}{L}\right) = \min\left\{\exp\left(-\frac{\mu T}{L}\right), \exp\left(-\frac{\mu^{\frac{1}{2}}T}{L^{\frac{1}{2}}K^{\frac{1}{2}}}\right)\right\}\Psi_0, \tag{B.6}$$

and

$$\varphi_\downarrow(\eta_0) \leq \exp\left(-\sqrt{\frac{\eta_0\mu}{K}}T\right)\Psi_0 = \left(\mathrm{e} + \min\left\{\frac{\mu MT\Psi_0}{\sigma^2}, \frac{\mu^2 T^3\Psi_0}{LK^2\sigma^2}\right\}\right)^{-1}\Psi_0 \leq \frac{\sigma^2}{\mu MT} + \frac{LK^2\sigma^2}{\mu^2 T^3}. \tag{B.7}$$

On the other hand

$$\varphi_\uparrow(\eta) \leq \varphi_\uparrow(\eta_0) \leq \frac{\sigma^2}{2\mu MT}\log\left(\mathrm{e} + \frac{\mu MT\Psi_0}{\sigma^2}\right) + \frac{K\sigma^2}{2\mu MT^2}\log^2\left(\mathrm{e} + \frac{\mu MT\Psi_0}{\sigma^2}\right)$$

$$+ \frac{390LK^2\sigma^2}{\mu^2 T^3}\log^3\left(\mathrm{e} + \frac{\mu^2 T^3\Psi_0}{LK^2\sigma^2}\right) + \frac{7LK^3\sigma^2}{\mu^2 T^4}\log^4\left(\mathrm{e} + \frac{\mu^2 T^3\Psi_0}{LK^2\sigma^2}\right)$$

$$\leq \frac{\sigma^2}{\mu MT}\log^2\left(\mathrm{e} + \frac{\mu MT\Psi_0}{\sigma^2}\right) + \frac{397LK^2\sigma^2}{\mu^2 T^3}\log^4\left(\mathrm{e} + \frac{\mu^2 T^3\Psi_0}{LK^2\sigma^2}\right), \tag{B.8}$$

where the last inequality is due to $\frac{K\sigma^2}{2\mu MT} \leq \frac{\sigma^2}{\mu MT}$ and $\frac{7LK^3\sigma^2}{\mu^2 T^4} \leq \frac{7LK^2\sigma^2}{\mu^2 T^3}$ since $K \leq T$.

Combining Lemma B.4 and Eqs. (B.6), (B.7) and (B.8) gives

$$\mathbb{E}[\Psi_T] \leq \varphi_\downarrow\left(\frac{1}{L}\right) + \varphi_\downarrow(\eta_0) + \varphi_\uparrow(\eta)$$

$$\leq \min\left\{\exp\left(-\frac{\mu T}{L}\right), \exp\left(-\frac{\mu^{\frac{1}{2}}T}{L^{\frac{1}{2}}K^{\frac{1}{2}}}\right)\right\}\Psi_0 + \frac{2\sigma^2}{\mu MT}\log^2\left(\mathrm{e} + \frac{\mu MT\Psi_0}{\sigma^2}\right) + \frac{400LK^2\sigma^2}{\mu^2 T^3}\log^4\left(\mathrm{e} + \frac{\mu^2 T^3\Psi_0}{LK^2\sigma^2}\right),$$

completing the proof of main Theorem B.1.

$\square$

## B.2 Perturbed iterate analysis for FEDAC-I: Proof of Lemma B.2

In this section we will prove Lemma B.2. We start by the one-step analysis of the decentralized potential $\Psi_t$ defined in Eq. (B.1). The following two propositions establish the one-step analysis of the two quantities in $\Psi_t$, namely $\|\overline{w_t} - w^*\|^2$ and $\frac{1}{M}\sum_{m=1}^{M} F(w_t^{\mathrm{ag},m}) - F^*$. We only require minimal hyperparameter assumptions, namely $\alpha \geq 1, \beta \geq 1, \eta \leq \frac{1}{L}$, for these two propositions. We will then show how the choice of $\alpha, \beta$ is determined towards the proof of Lemma B.2 in order to couple the two quantities into potential $\Psi_t$.

**Proposition B.5.** *Let $F$ be $\mu > 0$-strongly convex, and assume Assumption 1, then for* FEDAC *with hyperparameters assumptions $\alpha \geq 1$, $\beta \geq 1$, $\eta \leq \frac{1}{L}$, the following inequality holds*

$$\mathbb{E}[\|\overline{w_{t+1}} - w^*\|^2 | \mathcal{F}_t]$$

$$\leq (1 - \alpha^{-1})\|\overline{w_t} - w^*\|^2 + \alpha^{-1}\|\overline{w_t^{\mathrm{md}}} - w^*\|^2 + \gamma^2 \left\|\frac{1}{M}\sum_{m=1}^{M} \nabla F(w_t^{\mathrm{md},m})\right\|^2 + \frac{1}{M}\gamma^2\sigma^2$$

$$- 2\gamma \cdot \frac{1}{M}\sum_{m=1}^{M} \left\langle \nabla F(w_t^{\mathrm{md},m}), (1 - \alpha^{-1}(1 - \beta^{-1}))w_t^m + \alpha^{-1}(1 - \beta^{-1})w_t^{\mathrm{ag},m} - w^* \right\rangle$$

$$+ 2\gamma L \frac{1}{M}\sum_{m=1}^{M} \left\|\overline{w_t^{\mathrm{md}}} - w_t^{\mathrm{md},m}\right\| \left\|(1 - \alpha^{-1}(1 - \beta^{-1}))(\overline{w_t} - w_t^m) + \alpha^{-1}(1 - \beta^{-1})(\overline{w_t^{\mathrm{ag}}} - w_t^{\mathrm{ag},m})\right\|.$$

**Proposition B.6.** *In the same setting of Proposition B.5, the following inequality holds*

$$\mathbb{E}\left[\frac{1}{M}\sum_{m=1}^{M} F(w_{t+1}^{\mathrm{ag},m}) - F^* \middle| \mathcal{F}_t\right]$$

$$\leq (1 - \alpha^{-1})\left(\frac{1}{M}\sum_{m=1}^{M} F(w_t^{\mathrm{ag},m}) - F^*\right) - \frac{1}{2}\eta \left\|\frac{1}{M}\sum_{m=1}^{M} \nabla F(w_t^{\mathrm{md},m})\right\|^2 + \frac{1}{2}\eta^2 L\sigma^2$$

$$+ \alpha^{-1}\frac{1}{M}\sum_{m=1}^{M} \left\langle \nabla F(w_t^{\mathrm{md},m}), \alpha\beta^{-1}w_t^m + (1 - \alpha\beta^{-1})w_t^{\mathrm{ag},m} - w^* \right\rangle - \frac{1}{2}\mu\alpha^{-1}\|\overline{w_t^{\mathrm{md}}} - w^*\|^2.$$

We defer the proofs of Propositions B.5 and B.6 to Sections B.2.1 and B.2.2, respectively.

With Propositions B.5 and B.6 at hand we are ready to prove Lemma B.2.

*Proof of Lemma B.2.* Applying Proposition B.5 with the specified $\alpha = \frac{1}{\gamma\mu}, \beta = \alpha + 1$ yields (for any $t$)

$$\mathbb{E}[\|\overline{w_{t+1}} - w^*\|^2 | \mathcal{F}_t]$$

$$\leq (1 - \gamma\mu)\|\overline{w_t} - w^*\|^2 + \gamma\mu\|\overline{w_t^{\mathrm{md}}} - w^*\|^2 + \gamma^2 \left\|\frac{1}{M}\sum_{m=1}^{M} \nabla F(w_t^{\mathrm{md},m})\right\|^2 + \frac{1}{M}\gamma^2\sigma^2$$

$$- 2\gamma \cdot \frac{1}{M}\sum_{m=1}^{M} \left\langle \nabla F(w_t^{\mathrm{md},m}), \frac{1}{1 + \gamma\mu}w_t^m + \frac{\gamma\mu}{1 + \gamma\mu}w_t^{\mathrm{ag},m} - w^* \right\rangle$$

$$+ 2\gamma L \cdot \frac{1}{M}\sum_{m=1}^{M} \left\|\overline{w_t^{\mathrm{md}}} - w_t^{\mathrm{md},m}\right\| \left\|\frac{1}{1 + \gamma\mu}(\overline{w_t} - w_t^m) + \frac{\gamma\mu}{1 + \gamma\mu}(\overline{w_t^{\mathrm{ag}}} - w_t^{\mathrm{ag},m})\right\|. \quad \text{(B.9)}$$

Applying Proposition B.6 with the specified $\alpha = \frac{1}{\gamma\mu}, \beta = \alpha + 1$ yields (for any $t$)

$$\mathbb{E}\left[\frac{1}{M}\sum_{m=1}^{M}F(w_{t+1}^{\mathrm{ag},m}) - F^* \middle| \mathcal{F}_t\right]$$

$$\leq (1-\gamma\mu)\left(\frac{1}{M}\sum_{m=1}^{M}F(w_t^{\mathrm{ag},m}) - F^*\right) - \frac{1}{2}\eta\left\|\frac{1}{M}\sum_{m=1}^{M}\nabla F(w_t^{\mathrm{md},m})\right\|^2 + \frac{1}{2}\eta^2 L\sigma^2$$

$$+ \gamma\mu \cdot \frac{1}{M}\sum_{m=1}^{M}\left\langle \nabla F(w_t^{\mathrm{md},m}), \frac{1}{1+\gamma\mu}w_t^m + \frac{\gamma\mu}{1+\gamma\mu}w_t^{\mathrm{ag},m} - w^*\right\rangle - \frac{1}{2}\gamma\mu^2\|\overline{w_t^{\mathrm{md}}} - w^*\|^2.$$

(B.10)

Adding Eq. (B.10) with $\frac{1}{2}\mu$ times of Eq. (B.9) yields

$$\mathbb{E}[\Psi_{t+1}|\mathcal{F}_t] \leq (1-\gamma\mu)\Psi_t + \frac{1}{2}\left(\eta^2 L + \frac{1}{M}\gamma^2\mu\right)\sigma^2 + \frac{1}{2}\left(\gamma^2\mu - \eta\right)\left\|\frac{1}{M}\sum_{m=1}^{M}\nabla F(w_t^{\mathrm{md},m})\right\|^2$$

$$+ \gamma\mu L \cdot \frac{1}{M}\sum_{m=1}^{M}\left\|\overline{w_t^{\mathrm{md}}} - w_t^{\mathrm{md},m}\right\|\left\|\frac{1}{1+\gamma\mu}(\overline{w_t} - w_t^m) + \frac{\gamma\mu}{1+\gamma\mu}(\overline{w_t^{\mathrm{ag}}} - w_t^{\mathrm{ag},m})\right\|.$$

Since $\gamma^2\mu \leq \eta$, the coefficient of $\left\|\frac{1}{M}\sum_{m=1}^{M}\nabla F(w_t^{\mathrm{md},m})\right\|^2$ is non-positive. Thus

$$\mathbb{E}[\Psi_{t+1}|\mathcal{F}_t] \leq (1-\gamma\mu)\Psi_t + \frac{1}{2}\left(\eta^2 L + \frac{1}{M}\gamma^2\mu\right)\sigma^2$$

$$+ \gamma\mu L \cdot \frac{1}{M}\sum_{m=1}^{M}\left\|\overline{w_t^{\mathrm{md}}} - w_t^{\mathrm{md},m}\right\|\left\|\frac{1}{1+\gamma\mu}(\overline{w_t} - w_t^m) + \frac{\gamma\mu}{1+\gamma\mu}(\overline{w_t^{\mathrm{ag}}} - w_t^{\mathrm{ag},m})\right\|.$$

Telescoping the above inequality up to timestep $T$ yields

$$\mathbb{E}\left[\Psi_T\right] \leq (1-\gamma\mu)^T \Psi_0 + \left(\sum_{t=0}^{T-1}(1-\gamma\mu)^t\right) \cdot \frac{1}{2}\left(\eta^2 L + \frac{1}{M}\gamma^2\mu\right)\sigma^2$$

$$+ \gamma\mu L \cdot \sum_{t=0}^{T-1}\left\{(1-\gamma\mu)^{T-t-1} \cdot \mathbb{E}\left[\frac{1}{M}\sum_{m=1}^{M}\left\|\overline{w_t^{\mathrm{md}}} - w_t^{\mathrm{md},m}\right\|\left\|\frac{1}{1+\gamma\mu}(\overline{w_t} - w_t^m) + \frac{\gamma\mu}{1+\gamma\mu}(\overline{w_t^{\mathrm{ag}}} - w_t^{\mathrm{ag},m})\right\|\right]\right\}$$

$$\leq \exp\left(-\gamma\mu T\right)\Psi_0 + \frac{\eta^2 L\sigma^2}{2\gamma\mu} + \frac{\gamma\sigma^2}{2M}$$

$$+ L \cdot \max_{0\leq t < T}\mathbb{E}\left[\frac{1}{M}\sum_{m=1}^{M}\left\|\overline{w_t^{\mathrm{md}}} - w_t^{\mathrm{md},m}\right\|\left\|\frac{1}{1+\gamma\mu}(\overline{w_t} - w_t^m) + \frac{\gamma\mu}{1+\gamma\mu}(\overline{w_t^{\mathrm{ag}}} - w_t^{\mathrm{ag},m})\right\|\right],$$

where in the last inequality we used the fact that $(1-\gamma\mu)^T \leq \exp(-\gamma\mu T)$ and $\sum_{t=0}^{T-1}(1-\gamma\mu)^t \leq \frac{1}{\gamma\mu}$. $\square$

### B.2.1  Proof of Proposition B.5

*Proof of Proposition B.5.* By definition of the FEDAC procedure (Algorithm 1), for all $m \in [M]$ (recall $v_{t+1}^m$ is the candidate for next step),

$$v_{t+1}^m = (1-\alpha^{-1})w_t^m + \alpha^{-1}w_t^{\mathrm{md},m} - \gamma \cdot \nabla f(w_t^{\mathrm{md},m}; \xi_t^m).$$

Taking average over $m = 1, \ldots, M$ gives

$$\overline{w_{t+1}} - w^* = (1-\alpha^{-1})\overline{w_t} + \alpha^{-1}\overline{w_t^{\mathrm{md}}} - \gamma \cdot \frac{1}{M}\sum_{m=1}^{M}\nabla f(w_t^{\mathrm{md},m}; \xi_t^m) - w^*.$$

Taking conditional expectation gives

$$\mathbb{E}[\|\overline{w_{t+1}} - w^*\|^2 | \mathcal{F}_t]$$

$$= \left\| (1-\alpha^{-1})\overline{w_t} + \alpha^{-1}\overline{w_t^{\mathrm{md}}} - \gamma \cdot \frac{1}{M}\sum_{m=1}^{M} \nabla F(w_t^{\mathrm{md},m}) - w^* \right\|^2$$

$$+ \mathbb{E}\left[ \left\| \frac{1}{M}\sum_{m=1}^{M}\left( \nabla f(w_t^{\mathrm{md},m};\xi_t^m) - \nabla F(w_t^{\mathrm{md};m}) \right) \right\|^2 \Bigg| \mathcal{F}_t \right] \qquad \text{(independence)}$$

$$\leq \left\| (1-\alpha^{-1})\overline{w_t} + \alpha^{-1}\overline{w_t^{\mathrm{md}}} - \gamma \cdot \frac{1}{M}\sum_{m=1}^{M} \nabla F(w_t^{\mathrm{md},m}) - w^* \right\|^2 + \frac{1}{M}\gamma^2\sigma^2, \qquad \text{(B.11)}$$

where the last inequality of Eq. (B.11) is due to the bounded variance assumption (Assumption 1(c)) and independence. Expanding the squared norm term of Eq. (B.11) and applying Jensen's inequality,

$$\left\| (1-\alpha^{-1})\overline{w_t} + \alpha^{-1}\overline{w_t^{\mathrm{md}}} - \gamma \cdot \frac{1}{M}\sum_{m=1}^{M} \nabla F(w_t^{\mathrm{md},m}) - w^* \right\|^2$$

$$= \left\| (1-\alpha^{-1})\overline{w_t} + \alpha^{-1}\overline{w_t^{\mathrm{md}}} - w^* \right\|^2 + \gamma^2 \left\| \frac{1}{M}\sum_{m=1}^{M} \nabla F(w_t^{\mathrm{md},m}) \right\|^2$$

$$- 2\gamma \cdot \frac{1}{M}\sum_{m=1}^{M}\left\langle \nabla F(w_t^{\mathrm{md},m}), (1-\alpha^{-1})\overline{w_t} + \alpha^{-1}\overline{w_t^{\mathrm{md}}} - w^* \right\rangle \quad \text{(expansion of squared norm)}$$

$$\leq (1-\alpha^{-1})\|\overline{w_t} - w^*\|^2 + \alpha^{-1}\|\overline{w_t^{\mathrm{md}}} - w^*\|^2 + \gamma^2 \left\| \frac{1}{M}\sum_{m=1}^{M} \nabla F(w_t^{\mathrm{md},m}) \right\|^2$$

$$- 2\gamma \cdot \frac{1}{M}\sum_{m=1}^{M}\left\langle \nabla F(w_t^{\mathrm{md},m}), (1-\alpha^{-1})\overline{w_t} + \alpha^{-1}\overline{w_t^{\mathrm{md}}} - w^* \right\rangle, \qquad \text{(B.12)}$$

It remains to analyze the inner product term of Eq. (B.12). Note that

$$- \frac{1}{M}\sum_{m=1}^{M}\left\langle \nabla F(w_t^{\mathrm{md},m}), (1-\alpha^{-1})\overline{w_t} + \alpha^{-1}\overline{w_t^{\mathrm{md}}} - w^* \right\rangle$$

$$= - \frac{1}{M}\sum_{m=1}^{M}\left\langle \nabla F(w_t^{\mathrm{md},m}), (1-\alpha^{-1}(1-\beta^{-1}))\overline{w_t} + \alpha^{-1}(1-\beta^{-1})\overline{w_t^{\mathrm{ag}}} - w^* \right\rangle$$

$$\text{(definition of } \overline{w_t^{\mathrm{md}}})$$

$$= - \frac{1}{M}\sum_{m=1}^{M}\left\langle \nabla F(w_t^{\mathrm{md},m}), (1-\alpha^{-1}(1-\beta^{-1}))(\overline{w_t} - w_t^m) + \alpha^{-1}(1-\beta^{-1})(\overline{w_t^{\mathrm{ag}}} - w_t^{\mathrm{ag},m}) \right\rangle$$

$$- \frac{1}{M}\sum_{m=1}^{M}\left\langle \nabla F(w_t^{\mathrm{md},m}), (1-\alpha^{-1}(1-\beta^{-1}))w_t^m + \alpha^{-1}(1-\beta^{-1})w_t^{\mathrm{ag},m} - w^* \right\rangle$$

$$= \frac{1}{M}\sum_{m=1}^{M}\left\langle \nabla F(\overline{w_t^{\mathrm{md}}}) - \nabla F(w_t^{\mathrm{md},m}), (1-\alpha^{-1}(1-\beta^{-1}))(\overline{w_t} - w_t^m) + \alpha^{-1}(1-\beta^{-1})(\overline{w_t^{\mathrm{ag}}} - w_t^{\mathrm{ag},m}) \right\rangle$$

$$- \frac{1}{M}\sum_{m=1}^{M}\left\langle \nabla F(w_t^{\mathrm{md},m}), (1-\alpha^{-1}(1-\beta^{-1}))w_t^m + \alpha^{-1}(1-\beta^{-1})w_t^{\mathrm{ag},m} - w^* \right\rangle$$

$$\leq L \cdot \frac{1}{M}\sum_{m=1}^{M} \left\| \overline{w_t^{\mathrm{md}}} - w_t^{\mathrm{md},m} \right\| \left\| (1-\alpha^{-1}(1-\beta^{-1}))(\overline{w_t} - w_t^m) + \alpha^{-1}(1-\beta^{-1})(\overline{w_t^{\mathrm{ag}}} - w_t^{\mathrm{ag},m}) \right\|$$

$$- \frac{1}{M}\sum_{m=1}^{M}\left\langle \nabla F(w_t^{\mathrm{md},m}), (1-\alpha^{-1}(1-\beta^{-1}))w_t^m + \alpha^{-1}(1-\beta^{-1})w_t^{\mathrm{ag},m} - w^* \right\rangle, \qquad \text{(B.13)}$$

where the last equality is due to the $L$-smoothness (Assumption 1(b)). Combining Eqs. (B.11), (B.12) and (B.13) completes the proof of Proposition B.5. □

### B.2.2 Proof of Proposition B.6

Before stating the proof of Proposition B.6, we first introduce and prove the following claim for a single worker $m \in [M]$.

**Claim B.7.** *Under the same assumptions of Proposition B.6, for any $m \in [M]$, the following inequality holds (recall that $v_{t+1}^{\mathrm{ag},m}$ is defined as the candidate next update (see Algorithm 1) before possible synchronization)*

$$\mathbb{E}\left[F(v_{t+1}^{\mathrm{ag},m}) - F^* | \mathcal{F}_t\right] \leq (1 - \alpha^{-1})\left(F(w_t^{\mathrm{ag},m}) - F^*\right) - \frac{1}{2}\eta \left\|\nabla F(w_t^{\mathrm{md},m})\right\|^2 + \frac{1}{2}\eta^2 L\sigma^2$$
$$- \frac{1}{2}\mu\alpha^{-1}\|w_t^{\mathrm{md},m} - w^*\|^2 + \alpha^{-1}\left\langle \nabla F(w_t^{\mathrm{md},m}), \alpha\beta^{-1}w_t^m + (1 - \alpha\beta^{-1})w_t^{\mathrm{ag},m} - w^*\right\rangle.$$

*Proof of Claim B.7.* By definition of FEDAC (Algorithm 1), $v_{t+1}^{\mathrm{ag},m} = w_t^{\mathrm{md},m} - \eta \cdot \nabla f(w_t^{\mathrm{md},m}; \xi_t^m)$. Thus, by $L$-smoothness (Assumption 1(b)),

$$F(v_{t+1}^{\mathrm{ag},m}) \leq F(w_t^{\mathrm{md},m}) - \eta \left\langle \nabla F(w_t^{\mathrm{md},m}), \nabla f(w_t^{\mathrm{md},m}; \xi_t^m)\right\rangle + \frac{1}{2}\eta^2 L \left\|\nabla f(w_t^{\mathrm{md},m}; \xi_t^m)\right\|^2.$$

Taking conditional expectation gives

$$\mathbb{E}\left[F(v_{t+1}^{\mathrm{ag},m}) | \mathcal{F}_t\right] \leq F(w_t^{\mathrm{md},m}) - \eta \left\|\nabla F(w_t^{\mathrm{md},m})\right\|^2 + \frac{1}{2}\eta^2 L \left\|\nabla F(w_t^{\mathrm{md},m})\right\|^2 + \frac{1}{2}\eta^2 L\sigma^2$$
$$= F(w_t^{\mathrm{md},m}) - \eta\left(1 - \frac{1}{2}\eta L\right)\left\|\nabla F(w_t^{\mathrm{md},m})\right\|^2 + \frac{1}{2}\eta^2 L\sigma^2.$$

Since $\eta \leq \frac{1}{L}$ we have $1 - \frac{1}{2}\eta L \geq \frac{1}{2}$. Thus

$$\mathbb{E}\left[F(v_{t+1}^{\mathrm{ag},m}) | \mathcal{F}_t\right] \leq F(w_t^{\mathrm{md},m}) - \frac{1}{2}\eta \left\|\nabla F(w_t^{\mathrm{md},m})\right\|^2 + \frac{1}{2}\eta^2 L\sigma^2. \tag{B.14}$$

Now we connect $F(w_t^{\mathrm{md},m})$ with $F(w_t^{\mathrm{ag},m})$ as follows.

$$F(w_t^{\mathrm{md},m}) - F^*$$
$$= (1 - \alpha^{-1})\left(F(w_t^{\mathrm{ag},m}) - F^*\right) + \alpha^{-1}\left(F(w_t^{\mathrm{md},m}) - F^*\right) + (1 - \alpha^{-1})\left(F(w_t^{\mathrm{md},m}) - F(w_t^{\mathrm{ag},m})\right)$$
$$\leq (1 - \alpha^{-1})\left(F(w_t^{\mathrm{ag},m}) - F^*\right) - \frac{1}{2}\mu\alpha^{-1}\|w_t^{\mathrm{md},m} - w^*\|^2 + \alpha^{-1}\left\langle \nabla F(w_t^{\mathrm{md},m}), w_t^{\mathrm{md},m} - w^*\right\rangle$$
$$\quad + (1 - \alpha^{-1})\left\langle \nabla F(w_t^{\mathrm{md},m}), w_t^{\mathrm{md},m} - w_t^{\mathrm{ag},m}\right\rangle \qquad (\mu\text{-strong-convexity})$$
$$= (1 - \alpha^{-1})\left(F(w_t^{\mathrm{ag},m}) - F^*\right) - \frac{1}{2}\mu\alpha^{-1}\|w_t^{\mathrm{md},m} - w^*\|^2$$
$$\quad + \alpha^{-1}\left\langle \nabla F(w_t^{\mathrm{md},m}), \alpha\beta^{-1}w_t^m + (1 - \alpha\beta^{-1})w_t^{\mathrm{ag},m} - w^*\right\rangle, \tag{B.15}$$

where the last equality is due to the definition of $w_t^{\mathrm{md},m}$. Plugging Eq. (B.15) to Eq. (B.14) completes the proof of Claim B.7. □

Now we complete the proof of Proposition B.6 by assembling the bound for all workers in Claim B.7.

*Proof of Proposition B.6.* If $t + 1$ is a synchronized step, then $w_{t+1}^{\mathrm{ag},m} = \overline{v_{t+1}^{\mathrm{ag}}}$ for all $m$. Then by convexity,

$$\frac{1}{M}\sum_{m=1}^{M} F(w_{t+1}^{\mathrm{ag},m}) = \frac{1}{M} \cdot M \cdot F\left(\overline{v_{t+1}^{\mathrm{ag}}}\right) = F\left(\overline{v_{t+1}^{\mathrm{ag}}}\right) \leq \frac{1}{M}\sum_{m=1}^{M} F(v_{t+1}^{\mathrm{ag},m}).$$

If $t + 1$ is not a synchronized step, then trivially $\frac{1}{M}\sum_{m=1}^{M} F(w_{t+1}^{\mathrm{ag},m}) = \frac{1}{M}\sum_{m=1}^{M} F(v_{t+1}^{\mathrm{ag},m})$.

Hence in either case

$$\frac{1}{M}\sum_{m=1}^{M} F(w_{t+1}^{\mathrm{ag,m}}) \leq \frac{1}{M}\sum_{m=1}^{M} F(v_{t+1}^{\mathrm{ag,m}}).$$

Now we average the bounds of Claim B.7 for $m = 1, \ldots, M$, which gives

$$\mathbb{E}\left[\frac{1}{M}\sum_{m=1}^{M} F(w_{t+1}^{\mathrm{ag},m}) - F^* \Big| \mathcal{F}_t\right] \leq \mathbb{E}\left[\frac{1}{M}\sum_{m=1}^{M} F(v_{t+1}^{\mathrm{ag},m}) - F^* \Big| \mathcal{F}_t\right]$$

$$\leq (1-\alpha^{-1})\left(\frac{1}{M}\sum_{m=1}^{M} F(w_t^{\mathrm{ag},m}) - F^*\right) - \frac{1}{2}\eta \cdot \frac{1}{M}\sum_{m=1}^{M}\left\|\nabla F(w_t^{\mathrm{md},m})\right\|^2 + \frac{1}{2}\eta^2 L\sigma^2$$

$$+\alpha^{-1}\frac{1}{M}\sum_{m=1}^{M}\left\langle \nabla F(w_t^{\mathrm{md},m}), \alpha\beta^{-1}w_t^m + (1-\alpha\beta^{-1})w_t^{\mathrm{ag},m} - w^*\right\rangle - \frac{1}{2}\mu\alpha^{-1}\frac{1}{M}\sum_{m=1}^{M}\|w_t^{\mathrm{md},m} - w^*\|^2$$

$$\leq (1-\alpha^{-1})\left(\frac{1}{M}\sum_{m=1}^{M} F(w_t^{\mathrm{ag},m}) - F^*\right) - \frac{1}{2}\eta\left\|\frac{1}{M}\sum_{m=1}^{M}\nabla F(w_t^{\mathrm{md},m})\right\|^2 + \frac{1}{2}\eta^2 L\sigma^2$$

$$+\alpha^{-1}\frac{1}{M}\sum_{m=1}^{M}\left\langle \nabla F(w_t^{\mathrm{md},m}), \alpha\beta^{-1}w_t^m + (1-\alpha\beta^{-1})w_t^{\mathrm{ag},m} - w^*\right\rangle - \frac{1}{2}\mu\alpha^{-1}\|\overline{w_t^{\mathrm{md}}} - w^*\|^2,$$

where the last inequality is due to Jensen's inequality on the convex function $\|\cdot\|^2$. $\qquad\square$

## B.3 Discrepancy overhead bound for FEDAC-I: Proof of Lemma B.3

In this subsection we prove Lemma B.3 regarding the growth of discrepancy overhead introduced in Lemma B.2.

We first introduce a few more notations to simplify the discussions throughout this subsection. Let $m_1, m_2 \in [M]$ be two arbitrary distinct workers. For any timestep $t$, denote $\Delta_t := w_t^{m_1} - w_t^{m_2}$, $\Delta_t^{\mathrm{ag}} := w_t^{\mathrm{ag},m_1} - w_t^{\mathrm{ag},m_2}$ and $\Delta_t^{\mathrm{md}} := w_t^{\mathrm{md},m_1} - w_t^{\mathrm{md},m_2}$ be the corresponding vector differences. Let $\Delta_t^\varepsilon = \varepsilon_t^{m_1} - \varepsilon_t^{m_2}$, where $\varepsilon_t^m := \nabla f(w_t^{\mathrm{md},m}; \xi_t^m) - \nabla F(w_t^{\mathrm{md},m})$ be the noise of the stochastic gradient oracle of the $m$-th worker evaluated at $w_t^{\mathrm{md}}$.

The proof of Lemma B.3 is based on the following propositions.

The following Proposition B.8 studies the growth of $\begin{bmatrix}\Delta_t^{\mathrm{ag}}\\\Delta_t\end{bmatrix}$ at each step. The proof of Proposition B.8 is deferred to Section B.3.1.

**Proposition B.8.** *In the same setting of Lemma B.3, suppose $t+1$ is not a synchronized step, then there exists a matrix $H_t$ such that $\mu I \preceq H_t \preceq LI$ satisfying*

$$\begin{bmatrix}\Delta_{t+1}^{\mathrm{ag}}\\\Delta_{t+1}\end{bmatrix} = \mathcal{A}(\mu, \gamma, \eta, H_t)\begin{bmatrix}\Delta_t^{\mathrm{ag}}\\\Delta_t\end{bmatrix} - \begin{bmatrix}\eta I\\\gamma I\end{bmatrix}\Delta_t^\varepsilon,$$

*where $\mathcal{A}(\mu, \gamma, \eta, H)$ is a matrix-valued function defined as*

$$\mathcal{A}(\mu, \gamma, \eta, H) = \frac{1}{1+\gamma\mu}\begin{bmatrix} I - \eta H & \gamma\mu(I - \eta H) \\ -\gamma(H - \mu I) & I - \gamma^2\mu H \end{bmatrix}. \tag{B.16}$$

Let us pause for a moment and discuss the intuition of the next steps of our plan. Our goal is to bound the product of several $\mathcal{A}(\mu, \gamma, \eta, H_i)$ where the $H_i$ matrix may be different. The natural idea is to bound the uniform norm bound of $\mathcal{A}$ for some norm $\|\cdot\|_\star$: $\sup_{\mu I \preceq H \preceq LI}\|\mathcal{A}\|_\star$. It is worth noticing that the matrix operator norm will not give the desired bound — $\sup_{\mu I \preceq H \preceq LI}\|\mathcal{A}\|_2$ is not sufficiently small for our purpose. Our approach is to leverage the "transformed" norm [Golub and Van Loan, 2013] $\|\mathcal{A}\|_{\mathcal{X}} := \|\mathcal{X}^{-1}\mathcal{A}\mathcal{X}\|_2$ for certain non-singular $\mathcal{X}$ and analyze the uniform norm bound for $\sup_{\mu I \preceq H \preceq LI}\|\mathcal{X}^{-1}\mathcal{A}\mathcal{X}\|_2$.

Formally, the following Proposition B.9 studies the uniform norm bound of $\mathcal{A}$ under the proposed transformation $\mathcal{X}$. The proof of Proposition B.9 is deferred to Section B.3.2.

**Proposition B.9** (Uniform norm bound of $\mathcal{A}$ under transformation $\mathcal{X}$). *Let $\mathcal{A}(\mu, \gamma, \eta, H)$ be defined in Eq. (B.16). and assume $\mu > 0$, $\gamma \in [\eta, \sqrt{\frac{\eta}{\mu}}]$, $\eta \in (0, \frac{1}{L}]$. Then the following uniform norm bound holds*

$$\sup_{\mu I \preceq H \preceq LI} \left\| \mathcal{X}(\gamma, \eta)^{-1} \mathcal{A}(\mu, \gamma, \eta, H) \mathcal{X}(\gamma, \eta) \right\| \leq \begin{cases} 1 + \frac{2\gamma^2 \mu}{\eta} & \text{if } \gamma \in \left(\eta, \sqrt{\frac{\eta}{\mu}}\right], \\ 1 & \text{if } \gamma = \eta, \end{cases}$$

*where $\mathcal{X}(\gamma, \eta)$ is a matrix-valued function defined as*

$$\mathcal{X}(\gamma, \eta) := \begin{bmatrix} \frac{\eta}{\gamma} I & 0 \\ I & I \end{bmatrix}. \tag{B.17}$$

Propositions B.8 and B.9 suggest the one step growth of $\left\| \mathcal{X}(\gamma, \eta)^{-1} \begin{bmatrix} \Delta_t^{\text{ag}} \\ \Delta_t \end{bmatrix} \right\|^2$ as follows.

**Proposition B.10.** *In the same setting of Lemma B.3, the following inequality holds (for all possible $t$)*

$$\mathbb{E}\left[ \left\| \mathcal{X}(\gamma, \eta)^{-1} \begin{bmatrix} \Delta_{t+1}^{\text{ag}} \\ \Delta_{t+1} \end{bmatrix} \right\|^2 \middle| \mathcal{F}_t \right] \leq 2\gamma^2 \sigma^2 + \left\| \mathcal{X}(\gamma, \eta)^{-1} \begin{bmatrix} \Delta_t^{\text{ag}} \\ \Delta_t \end{bmatrix} \right\|^2 \cdot \begin{cases} \left(1 + \frac{2\gamma^2 \mu}{\eta}\right)^2 & \text{if } \gamma \in \left(\eta, \sqrt{\frac{\eta}{\mu}}\right], \\ 1 & \text{if } \gamma = \eta, \end{cases}$$

*where $\mathcal{X}$ is the matrix-valued function defined in Eq. (B.17).*

The proof of Proposition B.10 is deferred to Section B.3.3.

The following Proposition B.11 relates the discrepancy overhead we wish to bound for Lemma B.3 with the quantity analyzed in Proposition B.10. The proof of Proposition B.11 is deferred to Section B.3.4.

**Proposition B.11.** *In the same setting of Lemma B.3, the following inequality holds (for all $t$)*

$$\frac{1}{M} \sum_{m=1}^{M} \left\| \overline{w_t^{\text{md}}} - w_t^{\text{md},m} \right\| \left\| \frac{1}{1 + \gamma\mu}(\overline{w_t} - w_t^m) + \frac{\gamma\mu}{1 + \gamma\mu}(\overline{w_t^{\text{ag}}} - w_t^{\text{ag},m}) \right\| \leq \frac{\sqrt{10}\eta}{\gamma} \left\| \mathcal{X}(\gamma, \eta)^{-1} \begin{bmatrix} \Delta_t^{\text{ag}} \\ \Delta_t \end{bmatrix} \right\|^2,$$

*where $\mathcal{X}$ is the matrix-valued function defined in Eq. (B.17).*

We are ready to finish the proof of Lemma B.3.

*Proof of Lemma B.3.* Let $t_0$ be the latest synchronized step prior to $t$ (note that the initial state $t = 0$ is always synchronized so $t_0$ is well-defined), then telescoping Proposition B.10 from $t_0$ to $t$ gives (note that $\Delta_{t_0}^{\text{ag}} = \Delta_{t_0} = 0$ due to synchronization)

$$\mathbb{E}\left[ \left\| \mathcal{X}(\gamma, \eta)^{-1} \begin{bmatrix} \Delta_t^{\text{ag}} \\ \Delta_t \end{bmatrix} \right\|^2 \middle| \mathcal{F}_{t_0} \right] \leq 2\gamma^2 \sigma^2 (t - t_0) \cdot \begin{cases} \left(1 + \frac{2\gamma^2 \mu}{\eta}\right)^{2(t-t_0)} & \text{if } \gamma \in \left(\eta, \sqrt{\frac{\eta}{\mu}}\right], \\ 1 & \text{if } \gamma = \eta \end{cases}$$

$$\leq 2\gamma^2 \sigma^2 K \cdot \begin{cases} \left(1 + \frac{2\gamma^2 \mu}{\eta}\right)^{2K} & \text{if } \gamma \in \left(\eta, \sqrt{\frac{\eta}{\mu}}\right], \\ 1 & \text{if } \gamma = \eta, \end{cases}$$

where the last inequality is due to $t - t_0 \leq K$ since $K$ is the synchronization interval.

Consequently, by Proposition B.11 we have

$$\frac{1}{M} \sum_{m=1}^{M} \mathbb{E}\left[ \left\| \overline{w_t^{\text{md}}} - w_t^{\text{md},m} \right\| \left\| \frac{1}{1 + \gamma\mu}(\overline{w_t} - w_t^m) + \frac{\gamma\mu}{1 + \gamma\mu}(\overline{w_t^{\text{ag}}} - w_t^{\text{ag},m}) \right\| \middle| \mathcal{F}_{t_0} \right]$$

$$\leq \frac{\sqrt{10}\eta}{\gamma} \mathbb{E}\left[ \left\| \mathcal{X}(\gamma, \eta)^{-1} \begin{bmatrix} \Delta_t^{\text{ag}} \\ \Delta_t \end{bmatrix} \right\|^2 \middle| \mathcal{F}_{t_0} \right] \leq \begin{cases} 7\eta\gamma K\sigma^2 \left(1 + \frac{2\gamma^2 \mu}{\eta}\right)^{2K} & \text{if } \gamma \in \left(\eta, \sqrt{\frac{\eta}{\mu}}\right], \\ 7\eta^2 K\sigma^2 & \text{if } \gamma = \eta, \end{cases}$$

where in the last inequality we used the estimate that $2\sqrt{10} < 7$. $\qquad\square$

### B.3.1 Proof of Proposition B.8

In this section we will prove Proposition B.8. Let us first state and prove a more general version of Proposition B.8 regarding FEDAC with general hyperparameter assumptions $\alpha \geq 1$, $\beta \geq 1$ .

**Claim B.12.** *Assume Assumption 1 and assume $F$ to be $\mu > 0$-strongly convex. Suppose $t + 1$ is not a synchronized step, then there exists a matrix $H_t$ such that $\mu I \preceq H_t \preceq LI$ satisfying*

$$\begin{bmatrix} \Delta_{t+1}^{\mathrm{ag}} \\ \Delta_{t+1} \end{bmatrix} = \begin{bmatrix} (1-\beta^{-1})(I-\eta H_t) & \beta^{-1}(I-\eta H_t) \\ (1-\beta^{-1})(\alpha^{-1}-\gamma H_t) & \beta^{-1}(\alpha^{-1}I-\gamma H_t) + (1-\alpha^{-1})I \end{bmatrix} \begin{bmatrix} \Delta_t^{\mathrm{ag}} \\ \Delta_t \end{bmatrix} - \begin{bmatrix} \eta I \\ \gamma I \end{bmatrix} \Delta_t^{\varepsilon}.$$

*Proof of Claim B.12.* First note that FEDAC can be written as the following two-point recursions.

$$w_{t+1}^{\mathrm{ag},m} = (1-\beta^{-1})w_t^{\mathrm{ag},m} + \beta^{-1}w_t^m - \eta \cdot \nabla F(w_t^{\mathrm{md},m}) - \eta\varepsilon_t^m;$$
$$w_{t+1}^m = \alpha^{-1}w_t^{\mathrm{md},m} + (1-\alpha^{-1})w_t^m - \gamma \cdot \nabla F(w_t^{\mathrm{md},m}) - \gamma\varepsilon_t^m$$
$$= \alpha^{-1}(1-\beta^{-1})w_t^{\mathrm{ag},m} + (1-\alpha^{-1}+\alpha^{-1}\beta^{-1})w_t^m - \gamma \cdot \nabla F(w_t^{\mathrm{md},m}) - \gamma\varepsilon_t^m.$$

Taking difference gives

$$\Delta_{t+1}^{\mathrm{ag}} = (1-\beta^{-1})\Delta_t^{\mathrm{ag}} + \beta^{-1}\Delta_t - \eta\left(\nabla F(w_t^{\mathrm{md},m_1}) - \nabla F(w_t^{\mathrm{md},m_2})\right) - \eta\Delta_t^{\varepsilon};$$

$$\Delta_{t+1} = \alpha^{-1}(1-\beta^{-1})\Delta_t^{\mathrm{ag}} + (1-\alpha^{-1}+\alpha^{-1}\beta^{-1})\Delta_t - \gamma\left(\nabla F(w_t^{\mathrm{md},m_1}) - \nabla F(w_t^{\mathrm{md},m_2})\right) - \gamma\Delta_t^{\varepsilon}.$$

By mean-value theorem, there exists a symmetric positive-definite matrix $H_t$ such that $\mu I \preceq H_t \preceq LI$ satisfying

$$\nabla F(w_t^{\mathrm{md},m_1}) - \nabla F(w_t^{\mathrm{md},m_2}) = H_t\Delta_t^{\mathrm{md}} = H_t\left((1-\beta^{-1})\Delta_t^{\mathrm{ag}} + \beta^{-1}\Delta_t\right).$$

Thus

$$\Delta_{t+1}^{\mathrm{ag}} = (1-\beta^{-1})\Delta_t^{\mathrm{ag}} + \beta^{-1}\Delta_t - \eta H_t\left((1-\beta^{-1})\Delta_t^{\mathrm{ag}} + \beta^{-1}\Delta_t\right) - \eta\Delta_t^{\varepsilon}$$
$$\Delta_{t+1} = \alpha^{-1}(1-\beta^{-1})\Delta_t^{\mathrm{ag}} + (1-\alpha^{-1}+\alpha^{-1}\beta^{-1})\Delta_t - \gamma H_t\left((1-\beta^{-1})\Delta_t^{\mathrm{ag}} + \beta^{-1}\Delta_t\right) - \gamma\Delta_t^{\varepsilon}$$

Rearranging into matrix form completes the proof of Claim B.12. $\square$

Proposition B.8 is a special case of Claim B.12.

*Proof of Proposition B.8.* The proof follows instantly by applying Claim B.12 with particular choice $\alpha = \frac{1}{\gamma\mu}$ and $\beta = \alpha + 1 = \frac{1+\gamma\mu}{\gamma\mu}$. $\square$

### B.3.2 Proof of Proposition B.9: uniform norm bound

*Proof of Proposition B.9.* Define another matrix-valued function $\mathcal{B}$ as

$$\mathcal{B}(\mu,\gamma,\eta,H) := \mathcal{X}(\gamma,\eta)^{-1}\mathcal{A}(\mu,\gamma,\eta,H)\mathcal{X}(\gamma,\eta).$$

Since $\mathcal{X}(\gamma,\eta)^{-1} = \begin{bmatrix} \frac{\gamma}{\eta}I & 0 \\ -\frac{\gamma}{\eta}I & I \end{bmatrix}$ we can compute that

$$\mathcal{B}(\mu,\gamma,\eta,H) = \frac{1}{(1+\gamma\mu)\eta}\begin{bmatrix} (\eta+\gamma^2\mu)(I-\eta H) & \gamma^2\mu(I-\eta H) \\ -\mu(\gamma^2-\eta^2)I & \eta-\gamma^2\mu \end{bmatrix}.$$

Define the four blocks of $\mathcal{B}(\mu,\gamma,\eta,H)$ as $\mathcal{B}_{11}(\mu,\gamma,\eta,H)$, $\mathcal{B}_{12}(\mu,\gamma,\eta,H)$, $\mathcal{B}_{21}(\mu,\gamma,\eta)$, $\mathcal{B}_{22}(\mu,\gamma,\eta)$ (note that the lower two blocks do not involve $H$), *i.e.,*

$$\mathcal{B}_{11}(\mu,\gamma,\eta,H) = \frac{\eta+\gamma^2\mu}{(1+\gamma\mu)\eta}(I-\eta H), \qquad \mathcal{B}_{12}(\mu,\gamma,\eta,H) = \frac{\gamma^2\mu}{(1+\gamma\mu)\eta}(I-\eta H),$$

$$\mathcal{B}_{21}(\mu,\gamma,\eta) = -\frac{\mu(\gamma^2-\eta^2)}{(1+\gamma\mu)\eta}I, \qquad \mathcal{B}_{22}(\mu,\gamma,\eta) = \frac{\eta-\gamma^2\mu}{(1+\gamma\mu)\eta}I.$$

**Case I:** $\eta < \gamma \le \sqrt{\frac{\eta}{\mu}}$. In this case we have

$$\|\mathcal{B}_{11}(\mu,\gamma,\eta,H)\| \le \frac{\eta + \gamma^2\mu}{(1+\gamma\mu)\eta}(1-\eta\mu) \le \frac{\eta+\gamma^2\mu}{\eta} = 1 + \frac{\gamma^2\mu}{\eta}, \qquad \text{(since } \eta\mu \le 1)$$

$$\|\mathcal{B}_{12}(\mu,\gamma,\eta,H)\| \le \frac{\gamma^2\mu}{(1+\gamma\mu)\eta}(1-\eta\mu) \le \frac{\gamma^2\mu}{\eta}, \qquad \text{(since } \eta\mu \le 1)$$

$$\|\mathcal{B}_{21}(\mu,\gamma,\eta)\| = \frac{\mu(\gamma^2-\eta^2)}{(1+\gamma\mu)\eta} \le \frac{\gamma^2\mu}{\eta}, \qquad \text{(since } \eta < \gamma \le \sqrt{\frac{\eta}{\mu}})$$

$$\|\mathcal{B}_{22}(\mu,\gamma,\eta)\| = \frac{\eta - \gamma^2\mu}{(1+\gamma\mu)\eta} \le \frac{1}{1+\gamma\mu} \le 1. \qquad \text{(since } \gamma \le \sqrt{\frac{\eta}{\mu}})$$

The operator norm of $\mathcal{B}$ can be bounded via its blocks via helper Lemma G.1 as

$$\mathcal{B}(\mu,\gamma,\eta,H)$$
$$\le \max\left\{\|\mathcal{B}_{11}(\mu,\gamma,\eta,H)\|, \|\mathcal{B}_{22}(\mu,\gamma,\eta)\|\right\} + \max\left\{\|\mathcal{B}_{12}(\mu,\gamma,\eta,H)\|, \|\mathcal{B}_{21}(\mu,\gamma,\eta)\|\right\}$$
$$\text{(Lemma G.1)}$$
$$\le \max\left\{1 + \frac{\gamma^2\mu}{\eta}, 1\right\} + \max\left\{\frac{\gamma^2\mu}{\eta}, \frac{\gamma^2\mu}{\eta}\right\} = 1 + \frac{2\gamma^2\mu}{\eta}.$$

**Case II:** $\gamma = \eta$. In this case we have

$$\|\mathcal{B}_{11}(\mu,\gamma,\eta,H)\| \le \frac{\eta + \eta^2\mu}{(1+\eta\mu)\eta}(1-\eta\mu) = 1 - \eta\mu,$$

$$\|\mathcal{B}_{12}(\mu,\gamma,\eta,H)\| \le \frac{\eta^2\mu}{(1+\eta\mu)\eta}(1-\eta\mu) = \frac{(1-\eta\mu)\eta\mu}{1+\eta\mu},$$

$$\|\mathcal{B}_{21}(\mu,\gamma,\eta)\| = 0,$$

$$\|\mathcal{B}_{22}(\mu,\gamma,\eta)\| = \frac{\eta - \eta^2\mu}{(1+\eta\mu)\eta} = \frac{1-\eta\mu}{1+\eta\mu}.$$

Similarly the operator norm of block matrix $\mathcal{B}$ can be bounded via its blocks via helper Lemma G.1 as

$$\mathcal{B}(\mu,\gamma,\eta,H)$$
$$\le \max\left\{\|\mathcal{B}_{11}(\mu,\gamma,\eta,H)\|, \|\mathcal{B}_{22}(\mu,\gamma,\eta)\|\right\} + \max\left\{\|\mathcal{B}_{12}(\mu,\gamma,\eta,H)\|, \|\mathcal{B}_{21}(\mu,\gamma,\eta)\|\right\}$$
$$\text{(Lemma G.1)}$$
$$\le \max\left\{1-\eta\mu, \frac{1-\eta\mu}{1+\eta\mu}\right\} + \frac{\eta\mu(1-\eta\mu)}{1+\eta\mu} = 1 - \eta\mu + \frac{\eta\mu(1-\eta\mu)}{1+\eta\mu} = \frac{1+\eta\mu - 2\eta^2\mu^2}{1+\eta\mu} \le 1.$$

Summarizing the above two cases completes the proof of Proposition B.9. $\qquad\square$

### B.3.3 Proof of Proposition B.10

In this section we apply Propositions B.8 and B.9 to establish Proposition B.10.

*Proof of Proposition B.10.* If $t+1$ is a synchronized step, then the bound trivially holds since $\Delta_{t+1}^{\text{ag}} = \Delta_{t+1} = 0$ due to synchronization.

Now assume $t+1$ is not a synchronized step, for which Proposition B.8 is applicable. Multiplying $\mathcal{X}(\gamma,\eta)^{-1}$ to the left on both sides of Proposition B.8 gives

$$\mathcal{X}(\gamma,\eta)^{-1}\begin{bmatrix}\Delta_{t+1}^{\text{ag}} \\ \Delta_{t+1}\end{bmatrix} = \mathcal{X}(\gamma,\eta)^{-1}\mathcal{A}(\mu,\gamma,\eta,H)\begin{bmatrix}\Delta_t^{\text{ag}} \\ \Delta_t\end{bmatrix} - \mathcal{X}(\gamma,\eta)^{-1}\begin{bmatrix}\eta I \\ \gamma I\end{bmatrix}\Delta_t^\varepsilon$$

$$= \mathcal{X}(\gamma,\eta)^{-1}\mathcal{A}(\mu,\gamma,\eta,H_t)\mathcal{X}(\gamma,\eta)^{-1}\left(\mathcal{X}(\gamma,\eta)\begin{bmatrix}\Delta_t^{\text{ag}} \\ \Delta_t\end{bmatrix}\right) - \begin{bmatrix}\gamma I \\ 0\end{bmatrix}\Delta_t^\varepsilon,$$

where the last equality is due to

$$\mathcal{X}(\gamma,\eta)^{-1} = \begin{bmatrix} \frac{\gamma}{\eta}I & 0 \\ -\frac{\gamma}{\eta}I & I \end{bmatrix}, \qquad \mathcal{X}(\gamma,\eta)^{-1}\begin{bmatrix}\eta I \\ \gamma I\end{bmatrix} = \begin{bmatrix}\gamma I \\ 0\end{bmatrix}.$$

Taking conditional expectation,

$$\mathbb{E}\left[\left\|\mathcal{X}(\gamma,\eta)^{-1}\begin{bmatrix}\Delta_{t+1}^{\mathrm{ag}} \\ \Delta_{t+1}\end{bmatrix}\right\|^2\middle|\mathcal{F}_t\right]$$

$$=\left\|\mathcal{X}^{-1}\mathcal{A}\mathcal{X}\left(\mathcal{X}^{-1}\begin{bmatrix}\Delta_t^{\mathrm{ag}} \\ \Delta_t\end{bmatrix}\right)\right\|^2 + \mathbb{E}\left[\left\|\begin{bmatrix}\gamma I \\ 0\end{bmatrix}\Delta_t^\varepsilon\right\|^2\middle|\mathcal{F}_t\right] \qquad \text{(independence)}$$

$$\leq\|\mathcal{X}^{-1}\mathcal{A}\mathcal{X}\|^2\left\|\mathcal{X}^{-1}\begin{bmatrix}\Delta_t^{\mathrm{ag}} \\ \Delta_t\end{bmatrix}\right\|^2 + 2\gamma^2\sigma^2 \qquad \text{(bounded variance, sub-multiplicativity)}$$

$$\leq 2\gamma^2\sigma^2 + \left\|\mathcal{X}(\gamma,\eta)^{-1}\begin{bmatrix}\Delta_t^{\mathrm{ag}} \\ \Delta_t\end{bmatrix}\right\|^2\cdot\begin{cases}\left(1+\frac{2\gamma^2\mu}{\eta}\right)^2 & \text{if } \gamma\in\left(\eta,\sqrt{\frac{\eta}{\mu}}\right], \\ 1 & \text{if } \gamma=\eta.\end{cases} \qquad \text{(by Proposition B.9)}$$

$\square$

### B.3.4 Proof of Proposition B.11

In this section we will prove Proposition B.11 in three steps via the following three claims. For all the three claims $\mathcal{X}$ stands for the matrix-valued functions defined in Eq. (B.17).

**Claim B.13.** *In the same setting of Proposition B.11,*

$$\frac{1}{M}\sum_{m=1}^M\left\|\overline{w_t^{\mathrm{md}}}-w_t^{\mathrm{md},m}\right\|\left\|\frac{1}{1+\gamma\mu}(\overline{w_t}-w_t^m)+\frac{\gamma\mu}{1+\gamma\mu}(\overline{w_t^{\mathrm{ag}}}-w_t^{\mathrm{ag},m})\right\|$$

$$\leq\left\|\begin{bmatrix}\frac{1}{1+\gamma\mu}I \\ \frac{\gamma\mu}{1+\gamma\mu}I\end{bmatrix}^\intercal\mathcal{X}(\gamma,\eta)\right\|\cdot\left\|\begin{bmatrix}\frac{\gamma\mu}{1+\gamma\mu}I \\ \frac{1}{1+\gamma\mu}I\end{bmatrix}^\intercal\mathcal{X}(\gamma,\eta)\right\|\cdot\left\|\mathcal{X}(\gamma,\eta)^{-1}\begin{bmatrix}\Delta_t^{\mathrm{ag}} \\ \Delta_t\end{bmatrix}\right\|^2.$$

**Claim B.14.** *Assume $\mu>0$, $\gamma\in[\eta,\sqrt{\frac{\eta}{\mu}}]$, $\eta\in(0,\frac{1}{L}]$, then $\left\|\mathcal{X}(\gamma,\eta)^\intercal\begin{bmatrix}\frac{1}{1+\gamma\mu}I \\ \frac{\gamma\mu}{1+\gamma\mu}I\end{bmatrix}\right\|\leq\frac{\sqrt{5}\eta}{\gamma}.$*

**Claim B.15.** *Assume $\mu>0$, $\gamma\in[\eta,\sqrt{\frac{\eta}{\mu}}]$, $\eta\in(0,\frac{1}{L}]$, then $\left\|\mathcal{X}(\gamma,\eta)^\intercal\begin{bmatrix}\frac{\gamma\mu}{1+\gamma\mu}I \\ \frac{1}{1+\gamma\mu}I\end{bmatrix}\right\|\leq\sqrt{2}.$*

Proposition B.11 follows immediately once we have Claims B.13, B.14 and B.15.

*Proof of Proposition B.11.* Follows trivially with Claims B.13, B.14 and B.15.

$$\frac{1}{M}\sum_{m=1}^M\left\|\overline{w_t^{\mathrm{md}}}-w_t^{\mathrm{md},m}\right\|\left\|\frac{1}{1+\gamma\mu}(\overline{w_t}-w_t^m)+\frac{\gamma\mu}{1+\gamma\mu}(\overline{w_t^{\mathrm{ag}}}-w_t^{\mathrm{ag},m})\right\|\leq\frac{\sqrt{10}\eta}{\gamma}\left\|\mathcal{X}(\gamma,\eta)^{-1}\begin{bmatrix}\Delta_t^{\mathrm{ag}} \\ \Delta_t\end{bmatrix}\right\|^2.$$

$\square$

Now we finish the proof of the three claims.

*Proof of Claim B.13.* Note that

$$\frac{1}{M}\sum_{m=1}^M\left\|\overline{w_t^{\mathrm{md}}}-w_t^{\mathrm{md},m}\right\|^2 \leq\|\Delta_t^{\mathrm{md}}\|^2 \qquad \text{(convexity of } \|\cdot\|^2)$$

$$=\left\|\begin{bmatrix}(1-\beta^{-1})I \\ \beta^{-1}I\end{bmatrix}^\intercal\begin{bmatrix}\Delta_t^{\mathrm{ag}} \\ \Delta_t\end{bmatrix}\right\|^2 = \left\|\begin{bmatrix}\frac{1}{1+\gamma\mu}I \\ \frac{\gamma\mu}{1+\gamma\mu}I\end{bmatrix}^\intercal\begin{bmatrix}\Delta_t^{\mathrm{ag}} \\ \Delta_t\end{bmatrix}\right\|^2 \qquad \text{(definition of ``md'')}$$

$$\leq\left\|\begin{bmatrix}\frac{1}{1+\gamma\mu}I \\ \frac{\gamma\mu}{1+\gamma\mu}I\end{bmatrix}^\intercal\mathcal{X}(\gamma,\eta)\right\|^2\left\|\mathcal{X}(\gamma,\eta)^{-1}\begin{bmatrix}\Delta_t^{\mathrm{ag}} \\ \Delta_t\end{bmatrix}\right\|^2, \qquad \text{(sub-multiplicativity)}$$

and similarly

$$\frac{1}{M} \sum_{m=1}^{M} \left\| \frac{1}{1+\gamma\mu}(\overline{w_t} - w_t^m) + \frac{\gamma\mu}{1+\gamma\mu}(\overline{w_t^{\mathrm{ag}}} - w_t^{\mathrm{ag},m}) \right\|^2$$

$$\leq \left\| \begin{bmatrix} \frac{\gamma\mu}{1+\gamma\mu}I \\ \frac{1}{1+\gamma\mu}I \end{bmatrix}^{\mathsf{T}} \begin{bmatrix} \Delta_t^{\mathrm{ag}} \\ \Delta_t \end{bmatrix} \right\|^2 \qquad \text{(convexity of } \|\cdot\|^2)$$

$$\leq \left\| \begin{bmatrix} \frac{\gamma\mu}{1+\gamma\mu}I \\ \frac{1}{1+\gamma\mu}I \end{bmatrix}^{\mathsf{T}} \mathcal{X}(\gamma,\eta) \right\|^2 \left\| \mathcal{X}(\gamma,\eta)^{-1} \begin{bmatrix} \Delta_t^{\mathrm{ag}} \\ \Delta_t \end{bmatrix} \right\|^2. \qquad \text{(sub-multiplicativity)}$$

Thus, by Cauchy-Schwarz inequality,

$$\frac{1}{M} \sum_{m=1}^{M} \left\| \overline{w_t^{\mathrm{md}}} - w_t^{\mathrm{md},m} \right\| \left\| \frac{1}{1+\gamma\mu}(\overline{w_t} - w_t^m) + \frac{\gamma\mu}{1+\gamma\mu}(\overline{w_t^{\mathrm{ag}}} - w_t^{\mathrm{ag},m}) \right\|$$

$$\leq \left( \frac{1}{M} \sum_{m=1}^{M} \left\| \overline{w_t^{\mathrm{md}}} - w_t^{\mathrm{md},m} \right\|^2 \right)^{\frac{1}{2}} \left( \frac{1}{M} \sum_{m=1}^{M} \left\| \frac{1}{1+\gamma\mu}(\overline{w_t} - w_t^m) + \frac{\gamma\mu}{1+\gamma\mu}(\overline{w_t^{\mathrm{ag}}} - w_t^{\mathrm{ag},m}) \right\|^2 \right)^{\frac{1}{2}}$$

$$\text{(Cauchy-Schwarz)}$$

$$\leq \left\| \begin{bmatrix} \frac{1}{1+\gamma\mu}I \\ \frac{\gamma\mu}{1+\gamma\mu}I \end{bmatrix}^{\mathsf{T}} \mathcal{X}(\gamma,\eta) \right\| \cdot \left\| \begin{bmatrix} \frac{\gamma\mu}{1+\gamma\mu}I \\ \frac{1}{1+\gamma\mu}I \end{bmatrix}^{\mathsf{T}} \mathcal{X}(\gamma,\eta) \right\| \cdot \left\| \mathcal{X}(\gamma,\eta)^{-1} \begin{bmatrix} \Delta_t^{\mathrm{ag}} \\ \Delta_t \end{bmatrix} \right\|^2,$$

completing the proof of Claim B.13. $\qquad \square$

*Proof of Claim B.14.* Direct calculation shows that

$$\mathcal{X}(\gamma,\eta)^{\mathsf{T}} \begin{bmatrix} \frac{1}{1+\gamma\mu}I \\ \frac{\gamma\mu}{1+\gamma\mu}I \end{bmatrix} = \begin{bmatrix} \frac{\eta}{\gamma}I & I \\ 0 & I \end{bmatrix} \begin{bmatrix} \frac{1}{1+\gamma\mu}I \\ \frac{\gamma\mu}{1+\gamma\mu}I \end{bmatrix} = \frac{1}{1+\gamma\mu} \begin{bmatrix} (\frac{\eta}{\gamma} + \gamma\mu)I \\ \gamma\mu I \end{bmatrix}.$$

Since

$$\left\| \begin{bmatrix} (\frac{\eta}{\gamma} + \gamma\mu)I \\ \gamma\mu I \end{bmatrix} \right\| = \sqrt{\left( \frac{\eta}{\gamma} + \gamma\mu \right)^2 + (\gamma\mu)^2} \leq \sqrt{\left( \frac{2\eta}{\gamma} \right)^2 + \left( \frac{\eta}{\gamma} \right)^2} = \frac{\sqrt{5}\eta}{\gamma}. \qquad \text{(since } \gamma\mu \leq \frac{\eta}{\gamma})$$

We conclude that

$$\left\| \mathcal{X}(\gamma,\eta)^{\mathsf{T}} \begin{bmatrix} \frac{1}{1+\gamma\mu}I \\ \frac{\gamma\mu}{1+\gamma\mu}I \end{bmatrix} \right\| \leq \frac{1}{1+\gamma\mu} \cdot \frac{\sqrt{5}\eta}{\gamma} \leq \frac{\sqrt{5}\eta}{\gamma}.$$

$\qquad \square$

*Proof of Claim B.15.* Direct calculation shows that

$$\mathcal{X}(\gamma,\eta)^{\mathsf{T}} \begin{bmatrix} \frac{\gamma\mu}{1+\gamma\mu}I \\ \frac{1}{1+\gamma\mu}I \end{bmatrix} = \begin{bmatrix} \frac{\eta}{\gamma}I & I \\ 0 & I \end{bmatrix} \begin{bmatrix} \frac{\gamma\mu}{1+\gamma\mu}I \\ \frac{1}{1+\gamma\mu}I \end{bmatrix} = \begin{bmatrix} \frac{1+\eta\mu}{1+\gamma\mu}I \\ \frac{1}{1+\gamma\mu}I \end{bmatrix},$$

and

$$\left\| \begin{bmatrix} \frac{1+\eta\mu}{1+\gamma\mu}I \\ \frac{1}{1+\gamma\mu}I \end{bmatrix} \right\| = \sqrt{\left( \frac{1+\eta\mu}{1+\gamma\mu} \right)^2 + \left( \frac{1}{1+\gamma\mu} \right)^2} \leq \sqrt{2}, \qquad \text{(since } \eta \leq \gamma)$$

completing the proof of Claim B.15. $\qquad \square$

## C  Analysis of FEDAC-II under Assumption 1 or 2

In this section we study the convergence of FEDAC-II. We provide a complete, non-asymptotic version of Theorem 3.3 on the convergence of FEDAC-II under Assumption 2 and provide the detailed proof, which expands the proof sketch in Section 4.2. We also study the convergence of FEDAC-II under Assumption 1, which we defer to the end of this section (see Section C.4) since the analysis is mostly shared.

Recall that FEDAC-II is defined as the FEDAC algorithm with the following hyperparameter choice:

$$\eta \in \left(0, \frac{1}{L}\right], \quad \gamma = \max\left\{\sqrt{\frac{\eta}{\mu K}}, \eta\right\}, \quad \alpha = \frac{3}{2\gamma\mu} - \frac{1}{2}, \quad \beta = \frac{2\alpha^2 - 1}{\alpha - 1}. \quad \text{(FEDAC-II)}$$

As we discussed in the proof sketch Section 4.2, for FEDAC-II, we keep track of the convergence via the "centralized" potential $\Phi_t$.

$$\Phi_t := F(\overline{w_t^{\mathrm{ag}}}) - F^* + \frac{1}{6}\mu\|\overline{w_t} - w^*\|^2. \quad \text{(C.1)}$$

Recall $\overline{w_t}$ is defined as $\frac{1}{M}\sum_{m=1}^{M} w_t^m$ and $\overline{w_t^{\mathrm{ag}}}$ is defined as $\frac{1}{M}\sum_{m=1}^{M} w_t^{\mathrm{ag},m}$. We use $\mathcal{F}_t$ to denote the $\sigma$-algebra generated by $\{w_\tau^m, w_\tau^{\mathrm{ag},m}\}_{\tau \le t, m \in [M]}$. Since FEDAC is Markovian, conditioning on $\mathcal{F}_t$ is equivalent to conditioning on $\{w_t^m, w_t^{\mathrm{ag},m}\}_{m \in [M]}$.

## C.1 Main theorem and lemmas: Complete version of Theorem 3.3

Now we introduce the main theorem on the convergence of FEDAC-II under Assumption 2.

**Theorem C.1** (Convergence of FEDAC-II under Assumption 2, complete version of Theorem 3.3)**.**
*Let $F$ be $\mu > 0$ strongly convex, and assume Assumption 2, then for*

$$\eta := \min\left\{\frac{1}{L}, \frac{9K}{\mu T^2}\log^2\left(\mathrm{e} + \min\left\{\frac{\mu MT\Phi_0}{\sigma^2} + \frac{\mu^2 MT^3\Phi_0}{LK^2\sigma^2}, \frac{\mu^5 T^8\Phi_0}{Q^2 K^6\sigma^4}\right\}\right)\right\},$$

FEDAC-II *yields*

$$\mathbb{E}[\Phi_T] \le \min\left\{\exp\left(-\frac{\mu T}{3L}\right), \exp\left(-\frac{\mu^{\frac{1}{2}}T}{3L^{\frac{1}{2}}K^{\frac{1}{2}}}\right)\right\}\Phi_0 + \frac{4\sigma^2}{\mu MT}\log\left(\mathrm{e} + \frac{\mu MT\Phi_0}{\sigma^2}\right)$$

$$+ \frac{55LK^2\sigma^2}{\mu^2 MT^3}\log^3\left(\mathrm{e} + \frac{\mu^2 MT^3\Phi_0}{LK^2\sigma^2}\right) + \frac{\mathrm{e}^{18}Q^2 K^6\sigma^4}{\mu^5 T^8}\log^8\left(\mathrm{e} + \frac{\mu^5 T^8\Phi_0}{Q^2 K^6\sigma^4}\right),$$

*where $\Phi_t$ is the "centralized" potential defined in Eq.* (C.1).

**Remark.** *The simplified version Theorem 3.3 in main body can be obtained by replacing $K$ with $T/R$ and upper bound $\Phi_0$ by $LD_0^2$.*

The proof of Theorem C.1 is based on the following two lemmas regarding convergence and stability respectively. To clarify the hyperparameter dependency, we state our lemma for general $\gamma \in \left[\eta, \sqrt{\frac{\eta}{\mu}}\right]$, which has one more degree of freedom than FEDAC-II where $\gamma = \max\left\{\sqrt{\frac{\eta}{\mu K}}, \eta\right\}$ is fixed.

**Lemma C.2** (Potential-based perturbed iterate analysis for FEDAC-II)**.** *Let $F$ be $\mu > 0$-strongly convex, and assume Assumption 1, then for $\alpha = \frac{3}{2\gamma\mu} - \frac{1}{2}, \beta = \frac{2\alpha^2-1}{\alpha-1}, \gamma \in \left[\eta, \sqrt{\frac{\eta}{\mu}}\right], \eta \in (0, \frac{1}{L}]$, FEDAC yields*

$$\mathbb{E}[\Phi_T] \le \exp\left(-\frac{1}{3}\gamma\mu T\right)\Phi_0 + \frac{3\eta^2 L\sigma^2}{2\gamma\mu M} + \frac{\gamma\sigma^2}{2M} + \frac{3}{\mu}\max_{0 \le t < T}\mathbb{E}\left[\left\|\nabla F(\overline{w_t^{\mathrm{md}}}) - \frac{1}{M}\sum_{m=1}^{M}\nabla F(w_t^{\mathrm{md},m})\right\|^2\right],$$

*where $\Phi_t$ is the decentralized potential defined in Eq.* (C.1).

The proof of Lemma C.2 is deferred to Section C.2. Note that Lemma C.2 only requires Assumption 1 (recall that Assumption 1 is strictly weaker than Assumption 2), which enables us to recycle this Lemma towards the convergence proof of FEDAC-II under Assumption 1 (see Section C.4).

The following lemma studies the discrepancy overhead by 4th-th order stability, which requires Assumption 2.

**Lemma C.3** (Discrepancy overhead bounds)**.** *Let $F$ be $\mu > 0$-strongly convex, and assume Assumption 2, then for the same hyperparameter choice as in Lemma C.2, FEDAC satisfies (for all $t$)*

$$\mathbb{E}\left[\left\|\nabla F(\overline{w_t^{\mathrm{md}}}) - \frac{1}{M}\sum_{m=1}^{M}\nabla F(w_t^{\mathrm{md},m})\right\|^2\right] \le \begin{cases} 44\eta^4 Q^2 K^2\sigma^4\left(1 + \frac{\gamma^2\mu}{\eta}\right)^{4K} & \text{if } \gamma \in \left(\eta, \sqrt{\frac{\eta}{\mu}}\right], \\ 44\eta^4 Q^2 K^2\sigma^4 & \text{if } \gamma = \eta. \end{cases}$$

The proof of Lemma C.3 is deferred to Section C.3.

Now we plug in the choice of $\gamma = \max\left\{\sqrt{\frac{\eta}{\mu K}}, \eta\right\}$ to Lemmas C.2 and C.3, which leads to the following lemma.

**Lemma C.4** (Convergence of FEDAC-II for general $\eta$). *Let $F$ be $\mu > 0$-strongly convex, and assume Assumption 2, then for any $\eta \in (0, \frac{1}{L}]$, FEDAC-II yields*

$$\mathbb{E}[\Phi_T] \leq \exp\left(-\frac{1}{3}\max\left\{\eta\mu, \sqrt{\frac{\eta\mu}{K}}\right\}T\right)\Phi_0 + \frac{\eta^{\frac{1}{2}}\sigma^2}{\mu^{\frac{1}{2}}MK^{\frac{1}{2}}} + \frac{2\eta^{\frac{3}{2}}LK^{\frac{1}{2}}\sigma^2}{\mu^{\frac{1}{2}}M} + \frac{e^9\eta^4Q^2K^2\sigma^4}{\mu},$$
(C.2)

*where $\Phi_t$ is the decentralized potential defined in Eq. (C.1).*

*Proof of Lemma C.4.* It is direct to verify that $\gamma = \max\left\{\eta, \sqrt{\frac{\eta}{\mu K}}\right\} \in \left[\eta, \sqrt{\frac{\eta}{\mu}}\right]$ so both Lemmas C.2 and C.3 are applicable. Applying Lemma C.2 yields

$$\mathbb{E}[\Phi_T] \leq \exp\left(-\frac{1}{3}\max\left\{\eta\mu, \sqrt{\frac{\eta\mu}{K}}\right\}T\right)\Phi_0 + \min\left\{\frac{3\eta L\sigma^2}{2\mu M}, \frac{3\eta^{\frac{3}{2}}LK^{\frac{1}{2}}\sigma^2}{2\mu^{\frac{1}{2}}M}\right\}$$
$$+ \max\left\{\frac{\eta\sigma^2}{2M}, \frac{\eta^{\frac{1}{2}}\sigma^2}{2\mu^{\frac{1}{2}}MK^{\frac{1}{2}}}\right\} + \frac{3}{\mu}\max_{0\leq t<T}\mathbb{E}\left[\left\|\nabla F(\overline{w_t^{\mathrm{md}}}) - \frac{1}{M}\sum_{m=1}^{M}\nabla F(w_t^{\mathrm{md},m})\right\|^2\right].$$
(C.3)

We bound $\min\left\{\frac{3\eta L\sigma^2}{2\mu M}, \frac{3\eta^{\frac{3}{2}}LK^{\frac{1}{2}}\sigma^2}{2\mu^{\frac{1}{2}}M}\right\}$ with $\frac{3\eta^{\frac{3}{2}}LK^{\frac{1}{2}}\sigma^2}{2\mu^{\frac{1}{2}}M}$, and bound $\max\left\{\frac{\eta\sigma^2}{2M}, \frac{\eta^{\frac{1}{2}}\sigma^2}{2\mu^{\frac{1}{2}}MK^{\frac{1}{2}}}\right\}$ with $\frac{\eta\sigma^2}{2M} + \frac{\eta^{\frac{1}{2}}\sigma^2}{2\mu^{\frac{1}{2}}MK^{\frac{1}{2}}}$. By AM-GM inequality and $\mu \leq L$, we have

$$\frac{\eta\sigma^2}{2M} \leq \frac{\eta^{\frac{3}{2}}\mu^{\frac{1}{2}}K^{\frac{1}{2}}\sigma^2}{4M} + \frac{\eta^{\frac{1}{2}}\sigma^2}{4\mu^{\frac{1}{2}}MK^{\frac{1}{2}}} \leq \frac{\eta^{\frac{3}{2}}LK^{\frac{1}{2}}\sigma^2}{4\mu^{\frac{1}{2}}M} + \frac{\eta^{\frac{1}{2}}\sigma^2}{4\mu^{\frac{1}{2}}MK^{\frac{1}{2}}}$$

Thus

$$\min\left\{\frac{3\eta L\sigma^2}{2\mu M}, \frac{3\eta^{\frac{3}{2}}LK^{\frac{1}{2}}\sigma^2}{2\mu^{\frac{1}{2}}M}\right\} + \max\left\{\frac{\eta\sigma^2}{2M}, \frac{\eta^{\frac{1}{2}}\sigma^2}{2\mu^{\frac{1}{2}}MK^{\frac{1}{2}}}\right\}$$
$$\leq \frac{3\eta^{\frac{3}{2}}LK^{\frac{1}{2}}\sigma^2}{2\mu^{\frac{1}{2}}M} + \frac{\eta\sigma^2}{2M} + \frac{\eta^{\frac{1}{2}}\sigma^2}{2\mu^{\frac{1}{2}}MK^{\frac{1}{2}}} \leq \frac{7\eta^{\frac{3}{2}}LK^{\frac{1}{2}}\sigma^2}{4\mu^{\frac{1}{2}}M} + \frac{3\eta^{\frac{1}{2}}\sigma^2}{4\mu^{\frac{1}{2}}MK^{\frac{1}{2}}},$$
(C.4)

Applying Lemma C.3 yields (for all $t$)

$$\frac{3}{\mu}\mathbb{E}\left[\left\|\nabla F(\overline{w_t^{\mathrm{md}}}) - \frac{1}{M}\sum_{m=1}^{M}\nabla F(w_t^{\mathrm{md},m})\right\|^2\right] \leq \begin{cases} \frac{132}{\mu}\eta^4Q^2K^2\sigma^4\left(1+\frac{1}{K}\right)^{4K} & \text{if } \gamma = \sqrt{\frac{\eta}{\mu K}} \\ \frac{132}{\mu}\eta^4Q^2K^2\sigma^4, & \text{if } \gamma = \eta \end{cases}$$
$$\leq 132e^4\mu^{-1}\eta^4Q^2K^2\sigma^4 \leq e^9\mu^{-1}\eta^4Q^2K^2\sigma^4,$$
(C.5)

where in the last inequality we used the estimation that $132e^4 < e^9$.

Combining Eqs. (C.3), (C.4) and (C.5) yields

$$\mathbb{E}[\Phi_T] \leq \exp\left(-\frac{1}{3}\max\left\{\eta\mu, \sqrt{\frac{\eta\mu}{K}}\right\}T\right)\Phi_0 + \frac{\eta^{\frac{1}{2}}\sigma^2}{\mu^{\frac{1}{2}}MK^{\frac{1}{2}}} + \frac{2\eta^{\frac{3}{2}}LK^{\frac{1}{2}}\sigma^2}{\mu^{\frac{1}{2}}M} + \frac{e^9\eta^4Q^2K^2\sigma^4}{\mu}.$$

$\square$

The main Theorem C.1 then follows by plugging the appropriate $\eta$ to Lemma C.4.

*Proof of Theorem C.1.* To simplify the notation, we denote the decreasing term in Eq. (C.2) in Lemma C.4 as $\varphi_\downarrow(\eta)$ and the increasing term as $\varphi_\uparrow(\eta)$, namely

$$\varphi_\downarrow(\eta) := \exp\left(-\frac{1}{3}\max\left\{\eta\mu, \sqrt{\frac{\eta\mu}{K}}\right\}T\right)\Phi_0, \quad \varphi_\uparrow(\eta) := \frac{\eta^{\frac{1}{2}}\sigma^2}{\mu^{\frac{1}{2}}MK^{\frac{1}{2}}} + \frac{2\eta^{\frac{3}{2}}LK^{\frac{1}{2}}\sigma^2}{\mu^{\frac{1}{2}}M} + \frac{\mathrm{e}^9\eta^4Q^2K^2\sigma^4}{\mu}.$$

Now let

$$\eta_0 := \frac{9K}{\mu T^2}\log^2\left(\mathrm{e} + \min\left\{\frac{\mu MT\Phi_0}{\sigma^2} + \frac{\mu^2 MT^3\Phi_0}{LK^2\sigma^2}, \frac{\mu^5 T^8\Phi_0}{Q^2K^6\sigma^4}\right\}\right)$$

then $\eta := \min\left\{\frac{1}{L}, \eta_0\right\}$. Therefore, the decreasing term $\varphi_\downarrow(\eta)$ is upper bounded by $\varphi_\downarrow(\frac{1}{L}) + \varphi_\downarrow(\eta_0)$, where

$$\varphi_\downarrow\left(\frac{1}{L}\right) \leq \min\left\{\exp\left(-\frac{\mu T}{3L}\right), \exp\left(-\frac{\mu^{\frac{1}{2}}T}{3L^{\frac{1}{2}}K^{\frac{1}{2}}}\right)\right\}\Phi_0, \tag{C.6}$$

and

$$\varphi_\downarrow(\eta_0) \leq \exp\left(-\frac{1}{3}\sqrt{\frac{\eta_0\mu}{K}}T\right)\Phi_0 = \left(\mathrm{e} + \min\left\{\frac{\mu MT\Phi_0}{\sigma^2} + \frac{\mu^2 MT^3\Phi_0}{LK^2\sigma^2}, \frac{\mu^5 T^8\Phi_0}{Q^2K^6\sigma^4}\right\}\right)^{-1}\Phi_0$$

$$\leq \frac{\sigma^2}{\mu MT} + \frac{LK^2\sigma^2}{\mu^2 MT^3} + \frac{Q^2K^6\sigma^4}{\mu^5 T^8}. \tag{C.7}$$

On the other hand

$$\varphi_\uparrow(\eta) \leq \varphi_\uparrow(\eta_0) \leq \frac{3\sigma^2}{\mu MT}\log\left(\mathrm{e} + \frac{\mu MT\Phi_0}{\sigma^2}\right) + \frac{54LK^2\sigma^2}{\mu^2 MT^3}\log^3\left(\mathrm{e} + \frac{\mu^2 MT^3\Phi_0}{LK^2\sigma^2}\right)$$

$$+ \frac{9^4\mathrm{e}^9Q^2K^6\sigma^4}{\mu^5 T^8}\log^8\left(\mathrm{e} + \frac{\mu^5 T^8\Phi_0}{Q^2K^6\sigma^4}\right). \tag{C.8}$$

Combining Lemma C.4 and Eqs. (C.6), (C.7) and (C.8) gives

$$\mathbb{E}[\Phi_T] \leq \varphi_\downarrow\left(\frac{1}{L}\right) + \varphi_\downarrow(\eta_0) + \varphi_\uparrow(\eta_0)$$

$$\leq \min\left\{\exp\left(-\frac{\mu T}{3L}\right), \exp\left(-\frac{\mu^{\frac{1}{2}}T}{3L^{\frac{1}{2}}K^{\frac{1}{2}}}\right)\right\}\Phi_0 + \frac{4\sigma^2}{\mu MT}\log\left(\mathrm{e} + \frac{\mu MT\Phi_0}{\sigma^2}\right)$$

$$+ \frac{55LK^2\sigma^2}{\mu^2 MT^3}\log^3\left(\mathrm{e} + \frac{\mu^2 MT^3\Phi_0}{LK^2\sigma^2}\right) + \frac{\mathrm{e}^{18}Q^2K^6\sigma^4}{\mu^5 T^8}\log^8\left(\mathrm{e} + \frac{\mu^5 T^8\Phi_0}{Q^2K^6\sigma^4}\right),$$

where in the last inequality we used the estimate $9^4\mathrm{e}^9 + 1 < \mathrm{e}^{18}$. $\qquad\square$

## C.2  Perturbed iterate analysis for FEDAC-II: Proof of Lemma C.2

In this subsection we will prove Lemma C.2. We start by the one-step analysis of the centralized potential defined in Eq. (C.1). The following two propositions establish the one-step analysis of the two quantities in $\Phi_t$, namely $\|\overline{w}_t - w^*\|^2$ and $F(\overline{w_t^{\mathrm{ag}}}) - F^*$. We only require minimal hyperparameter assumptions, namely $\alpha \geq 1, \beta \geq 1, \eta \leq \frac{1}{L}$ for these two propositions. We will then show how the choice of $\alpha, \beta$ are determined towards the proof of Lemma C.2 in order to couple the two quantities into potential $\Phi_t$.

**Proposition C.5.** *Let $F$ be $\mu > 0$-strongly convex, and assume Assumption 1, then for FEDAC with hyperparameters assumptions $\alpha \geq 1$, $\beta \geq 1$, $\eta \leq \frac{1}{L}$, the following inequality holds*

$$\mathbb{E}[\|\overline{w_{t+1}} - w^*\|^2|\mathcal{F}_t]$$

$$\leq \left(1 - \frac{1}{2}\alpha^{-1}\right)\|\overline{w_t} - w^*\|^2 + \frac{3}{2}\alpha^{-1}\|\overline{w_t^{\mathrm{md}}} - w^*\|^2 + \frac{3}{2}\gamma^2\left\|\nabla F(\overline{w_t^{\mathrm{md}}})\right\|^2$$

$$- 2\gamma\left(1 + \frac{1}{2}\alpha^{-1}\right)\left\langle\nabla F(\overline{w_t^{\mathrm{md}}}), (1 - \alpha^{-1}(1 - \beta^{-1}))\overline{w_t} + \alpha^{-1}(1 - \beta^{-1})\overline{w_t^{\mathrm{ag}}} - w^*\right\rangle$$

$$+ \gamma^2(1 + 2\alpha)\left\|\nabla F(\overline{w_t^{\mathrm{md}}}) - \frac{1}{M}\sum_{m=1}^{M}\nabla F(w_t^{\mathrm{md},m})\right\|^2 + \frac{\gamma^2\sigma^2}{M}. \tag{C.9}$$

**Proposition C.6.** *In the same setting of Proposition C.5, the following inequality holds*

$$
\mathbb{E}\left[F(\overline{w_{t+1}^{\mathrm{ag}}}) - F^* | \mathcal{F}_t\right]
$$

$$
\leq \left(1 - \frac{1}{2}\alpha^{-1}\right)\left(F(\overline{w_t^{\mathrm{ag}}}) - F^*\right) - \frac{1}{4}\mu\alpha^{-1}\left\|\overline{w_t^{\mathrm{md}}} - w^*\right\|^2 - \frac{1}{2}\eta\left\|\nabla F(\overline{w_t^{\mathrm{md}}})\right\|^2
$$

$$
+ \frac{1}{2}\alpha^{-1}\left\langle\nabla F(\overline{w_t^{\mathrm{md}}}), 2\alpha\beta^{-1}\overline{w_t} + (1 - 2\alpha\beta^{-1})\overline{w_t^{\mathrm{ag}}} - w^*\right\rangle
$$

$$
+ \frac{1}{2}\eta\left\|\nabla F(\overline{w_t^{\mathrm{md}}}) - \frac{1}{M}\sum_{m=1}^{M}\nabla F(w_t^{\mathrm{md},m})\right\|^2 + \frac{\eta^2 L\sigma^2}{2M}. \tag{C.10}
$$

We defer the proofs of Propositions C.5 and C.6 to Sections C.2.1 and C.2.2, respectively.

Now we are ready to prove Lemma C.2.

*Proof of Lemma C.2.* Since $\gamma \leq \sqrt{\frac{\eta}{\mu}} \leq \sqrt{\frac{1}{\mu L}} \leq \frac{1}{\mu}$, we have $\alpha = \frac{3}{2\gamma\mu} - \frac{1}{2} \geq 1$, and therefore $\beta = \frac{2\alpha^2 - 1}{\alpha - 1} \geq 1$. Hence both Propositions C.5 and C.6 are applicable.

Adding Eq. (C.10) with $\frac{1}{6}\mu$ times of Eq. (C.9) gives (note that the $\|\overline{w_t^{\mathrm{md}}} - w^*\|^2$ term is cancelled because $\frac{1}{4}\mu\alpha^{-1} = \frac{1}{6}\mu \cdot \frac{3}{2}\alpha^{-1}$)

$$
\mathbb{E}\left[\Phi_{t+1}|\mathcal{F}_t\right] \leq \underbrace{\left(1 - \frac{1}{2}\alpha^{-1}\right)\Phi_t}_{(\mathrm{I})} + \underbrace{\left(\frac{1}{4}\gamma^2\mu - \frac{1}{2}\eta\right)\left\|\nabla F(\overline{w_t^{\mathrm{md}}})\right\|^2}_{(\mathrm{II})}
$$

$$
+ \underbrace{\frac{1}{2}\alpha^{-1}\left\langle\nabla F(\overline{w_t^{\mathrm{md}}}), 2\alpha\beta^{-1}\overline{w_t} + (1 - 2\alpha\beta^{-1})\overline{w_t^{\mathrm{ag}}} - w^*\right\rangle}_{(\mathrm{III})}
$$

$$
- \underbrace{\frac{1}{3}\gamma\mu\left(1 + \frac{1}{2}\alpha^{-1}\right)\left\langle\nabla F(\overline{w_t^{\mathrm{md}}}), (1 - \alpha^{-1}(1 - \beta^{-1}))\overline{w_t} + \alpha^{-1}(1 - \beta^{-1})\overline{w_t^{\mathrm{ag}}} - w^*\right\rangle}_{(\mathrm{IV})}
$$

$$
+ \underbrace{\left(\frac{1}{2}\eta + \frac{1}{6}\gamma^2\mu(1 + 2\alpha)\right)\left\|\nabla F(\overline{w_t^{\mathrm{md}}}) - \frac{1}{M}\sum_{m=1}^{M}\nabla F(w_t^{\mathrm{md},m})\right\|^2}_{(\mathrm{V})} + \frac{\eta^2 L\sigma^2}{2M} + \frac{\gamma^2\mu\sigma^2}{6M}. \tag{C.11}
$$

Now we analyze the RHS of Eq. (C.11) term by term.

**Term (I) of Eq. (C.11)** Note that $\alpha^{-1} = \frac{2\gamma\mu}{3 - \gamma\mu} \geq \frac{2}{3}\gamma\mu$, we have

$$
\left(1 - \frac{1}{2}\alpha^{-1}\right)\Phi_t \leq \left(1 - \frac{1}{3}\gamma\mu\right)\Phi_t. \tag{C.12}
$$

**Term (II) of Eq. (C.11)** Since $\gamma^2\mu \leq \eta$ we have

$$
\left(\frac{1}{4}\gamma^2\mu - \frac{1}{2}\eta\right)\left\|\nabla F(\overline{w_t^{\mathrm{md}}})\right\|^2 \leq 0. \tag{C.13}
$$

**Term (III) and (IV) of Eq. (C.11)** Since $\beta = \frac{2\alpha^2 - 1}{\alpha - 1}$, we have $2\alpha\beta^{-1} = \frac{2\alpha(\alpha - 1)}{2\alpha^2 - 1} = (1 - \alpha^{-1}(1 - \beta^{-1}))$, and $1 - 2\alpha\beta^{-1} = \frac{2\alpha - 1}{2\alpha^2 - 1} = \alpha^{-1}(1 - \beta^{-1})$. Therefore, the two inner-product terms are

cancelled:

$$\frac{1}{2}\alpha^{-1}\left\langle \nabla F(\overline{w_t^{\mathrm{md}}}), 2\alpha\beta^{-1}\overline{w_t} + (1 - 2\alpha\beta^{-1})\overline{w_t^{\mathrm{ag}}} - w^*\right\rangle$$

$$-\frac{1}{3}\gamma\mu\left(1 + \frac{1}{2}\alpha^{-1}\right)\left\langle \nabla F(\overline{w_t^{\mathrm{md}}}), (1 - \alpha^{-1}(1 - \beta^{-1}))\overline{w_t} + \alpha^{-1}(1 - \beta^{-1})\overline{w_t^{\mathrm{ag}}} - w^*\right\rangle$$

$$= \left(\frac{1}{2}\alpha^{-1} - \frac{1}{3}\gamma\mu\left(1 + \frac{1}{2}\alpha^{-1}\right)\right)\left\langle \nabla F(\overline{w_t^{\mathrm{md}}}), \frac{2\alpha - 1}{2\alpha^2 - 1}\overline{w_t^{\mathrm{ag}}} + \left(\frac{2\alpha^2 - 2\alpha}{2\alpha^2 - 1}\right)\overline{w_t} - w^*\right\rangle$$

$$= \left(\frac{\gamma\mu}{3 - \gamma\mu} - \frac{1}{3}\gamma\mu\left(1 + \frac{\gamma\mu}{3 - \gamma\mu}\right)\right)\left\langle \nabla F(\overline{w_t^{\mathrm{md}}}), \frac{2\alpha - 1}{2\alpha^2 - 1}\overline{w_t^{\mathrm{ag}}} + \left(\frac{2\alpha^2 - 2\alpha}{2\alpha^2 - 1}\right)\overline{w_t} - w^*\right\rangle$$

$$\text{(since } \alpha^{-1} = \tfrac{2\gamma\mu}{3 - \gamma\mu})$$

$$= 0. \tag{C.14}$$

**Term (V) of Eq. (C.11)**   Since $\alpha = \frac{3 - \gamma\mu}{2\gamma\mu}$ and $\gamma \geq \eta$ we have

$$\left(\frac{1}{2}\eta + \frac{1}{6}\gamma^2\mu(1 + 2\alpha)\right) = \frac{1}{2}\eta + \frac{1}{6}\gamma^2\mu\left(\frac{6}{2\gamma\mu}\right) = \frac{1}{2}(\eta + \gamma) \leq \gamma. \tag{C.15}$$

Plugging Eqs. (C.12), (C.13), (C.14) and (C.15) to Eq. (C.11) gives

$$\mathbb{E}\left[\Phi_{t+1}|\mathcal{F}_t\right] \leq \left(1 - \frac{1}{3}\gamma\mu\right)\Phi_t + \frac{\eta^2 L\sigma^2}{2M} + \frac{\gamma^2\mu\sigma^2}{6M} + \gamma\left\|\nabla F(\overline{w_t^{\mathrm{md}}}) - \frac{1}{M}\sum_{m=1}^{M}\nabla F(w_t^{\mathrm{md},m})\right\|^2.$$

Telescoping the above inequality up to timestep $T$ yields

$$\mathbb{E}\left[\Phi_T\right] \leq \left(1 - \frac{1}{3}\gamma\mu\right)^T \Phi_0 + \left(\sum_{t=0}^{T-1}\left(1 - \frac{1}{3}\gamma\mu\right)^t\right) \cdot \left(\frac{\eta^2 L\sigma^2}{2M} + \frac{\gamma^2\mu\sigma^2}{6M}\right)$$

$$+ \gamma\sum_{t=0}^{T-1}\left(1 - \frac{1}{3}\gamma\mu\right)^{T-t-1}\mathbb{E}\left[\left\|\nabla F(\overline{w_t^{\mathrm{md}}}) - \frac{1}{M}\sum_{m=1}^{M}\nabla F(w_t^{\mathrm{md},m})\right\|^2\right]$$

$$\leq \exp\left(-\frac{1}{3}\gamma\mu T\right)\Phi_0 + \left(\frac{3\eta^2 L\sigma^2}{2\gamma\mu M} + \frac{\gamma\sigma^2}{2M}\right) + \frac{3}{\mu}\cdot\max_{0 \leq t < T}\mathbb{E}\left[\left\|\nabla F(\overline{w_t^{\mathrm{md}}}) - \frac{1}{M}\sum_{m=1}^{M}\nabla F(w_t^{\mathrm{md},m})\right\|^2\right],$$

where in the last inequality we used the fact that $(1 - \frac{1}{3}\gamma\mu)^T \leq \exp(-\frac{1}{3}\gamma\mu T)$ and $\sum_{t=0}^{T-1}\left(1 - \frac{1}{3}\gamma\mu\right)^t \leq \sum_{t=0}^{T-1}\left(1 - \frac{1}{3}\gamma\mu\right)^\infty = \frac{3}{\gamma\mu}$. $\qquad\square$

### C.2.1   Proof of Proposition C.5

*Proof of Proposition C.5.* By definition of the FEDAC procedure (Algorithm 1),

$$\overline{w_{t+1}} - w^* = (1 - \alpha^{-1})\overline{w_t} + \alpha^{-1}\overline{w_t^{\mathrm{md}}} - \gamma \cdot \frac{1}{M}\sum_{m=1}^{M}\nabla f(w_t^{\mathrm{md},m}; \xi_t^m) - w^*.$$

Taking conditional expectation gives

$$\mathbb{E}\left[\|\overline{w_{t+1}} - w^*\|^2\big|\mathcal{F}_t\right] \leq \left\|(1 - \alpha^{-1})\overline{w_t} + \alpha^{-1}\overline{w_t^{\mathrm{md}}} - \gamma \cdot \frac{1}{M}\sum_{m=1}^{M}\nabla F(w_t^{\mathrm{md},m}) - w^*\right\|^2 + \frac{1}{M}\gamma^2\sigma^2.$$

$$\tag{C.16}$$

The squared norm in Eq. (C.16) is bounded as

$$\left\| (1-\alpha^{-1})\overline{w_t} + \alpha^{-1}\overline{w_t^{\mathrm{md}}} - \gamma \cdot \frac{1}{M} \sum_{m=1}^{M} \nabla F(w_t^{\mathrm{md},m}) - w^* \right\|^2$$

$$= \left\| (1-\alpha^{-1})\overline{w_t} + \alpha^{-1}\overline{w_t^{\mathrm{md}}} - \gamma \nabla F(\overline{w_t^{\mathrm{md}}}) - w^* + \gamma \left( \nabla F(\overline{w_t^{\mathrm{md}}}) - \frac{1}{M} \sum_{m=1}^{M} \nabla F(w_t^{\mathrm{md}}) \right) \right\|^2$$

$$\leq \left(1 + \frac{1}{2}\alpha^{-1}\right) \left\| (1-\alpha^{-1})\overline{w_t} + \alpha^{-1}\overline{w_t^{\mathrm{md}}} - w^* - \gamma \nabla F(\overline{w_t^{\mathrm{md}}}) \right\|^2$$

$$+ \gamma^2(1+2\alpha) \left\| \nabla F(\overline{w_t^{\mathrm{md}}}) - \frac{1}{M} \sum_{m=1}^{M} \nabla F(w_t^{\mathrm{md},m}) \right\|^2$$

$$\qquad\qquad\qquad\qquad\qquad\qquad\qquad \text{(apply helper Lemma G.2 with } \zeta = \tfrac{1}{2}\alpha^{-1})$$

$$= \underbrace{\left(1 + \frac{1}{2}\alpha^{-1}\right) \left\| (1-\alpha^{-1})\overline{w_t} + \alpha^{-1}\overline{w_t^{\mathrm{md}}} - w^* \right\|^2}_{\text{(I)}} + \underbrace{\gamma^2 \left(1 + \frac{1}{2}\alpha^{-1}\right) \left\| \nabla F(\overline{w_t^{\mathrm{md}}}) \right\|^2}_{\text{(II)}}$$

$$\underbrace{-2\gamma\left(1+\frac{1}{2}\alpha^{-1}\right) \left\langle \nabla F(\overline{w_t^{\mathrm{md}}}), (1-\alpha^{-1})\overline{w_t} + \alpha^{-1}\overline{w_t^{\mathrm{md}}} - w^* \right\rangle}_{\text{(III)}}$$

$$+ \gamma^2 (1+2\alpha) \left\| \nabla F(\overline{w_t^{\mathrm{md}}}) - \frac{1}{M} \sum_{m=1}^{M} \nabla F(w_t^{\mathrm{md},m}) \right\|^2. \tag{C.17}$$

The first term (I) of Eq. (C.17) is bounded via Jensen's inequality as follows:

$$\left(1 + \frac{1}{2}\alpha^{-1}\right) \left\| (1-\alpha^{-1})\overline{w_t} + \alpha^{-1}\overline{w_t^{\mathrm{md}}} - w^* \right\|^2$$

$$\leq \left(1 + \frac{1}{2}\alpha^{-1}\right) \left( (1-\alpha^{-1})\|\overline{w_t} - w^*\|^2 + \alpha^{-1}\|\overline{w_t^{\mathrm{md}}} - w^*\|^2 \right) \qquad \text{(Jensen's inequality)}$$

$$\leq \left(1 - \frac{1}{2}\alpha^{-1}\right) \|\overline{w_t} - w^*\|^2 + \frac{3}{2}\alpha^{-1}\|\overline{w_t^{\mathrm{md}}} - w^*\|^2. \tag{C.18}$$

where in the last inequality of Eq. (C.18) we used the fact that $(1 + \frac{1}{2}\alpha^{-1})(1 - \alpha^{-1}) = 1 - \frac{1}{2}\alpha^{-1} - \frac{1}{2}\alpha^{-2} < 1 - \frac{1}{2}\alpha^{-1}$, and $(1 + \frac{1}{2}\alpha^{-1})\alpha^{-1} \leq \frac{3}{2}\alpha^{-1}$ as $\alpha \geq 1$.

The second term (II) of Eq. (C.17) is bounded as (since $\alpha \geq 1$)

$$\gamma^2 \left(1 + \frac{1}{2}\alpha^{-1}\right) \left\| \nabla F(\overline{w_t^{\mathrm{md}}}) \right\|^2 \leq \frac{3}{2}\gamma^2 \left\| \nabla F(\overline{w_t^{\mathrm{md}}}) \right\|^2. \tag{C.19}$$

To analyze the third term (III) of Eq. (C.17), we note that by definition of $\overline{w_t^{\mathrm{md}}}$,

$$-2\gamma\left(1+\frac{1}{2}\alpha^{-1}\right) \left\langle \nabla F(\overline{w_t^{\mathrm{md}}}), (1-\alpha^{-1})\overline{w_t} + \alpha^{-1}\overline{w_t^{\mathrm{md}}} - w^* \right\rangle$$

$$= -2\gamma\left(1+\frac{1}{2}\alpha^{-1}\right) \left\langle \nabla F(\overline{w_t^{\mathrm{md}}}), (1-\alpha^{-1}(1-\beta^{-1}))\overline{w_t} + \alpha^{-1}(1-\beta^{-1})\overline{w_t^{\mathrm{ag}}} - w^* \right\rangle. \tag{C.20}$$

Plugging Eqs. (C.17), (C.18), (C.19) and (C.20) back to Eq. (C.16) yields

$$
\mathbb{E}[\|\overline{w_{t+1}} - w^*\|^2 | \mathcal{F}_t]
$$

$$
\leq \left(1 - \frac{1}{2}\alpha^{-1}\right)\|\overline{w_t} - w^*\|^2 + \frac{3}{2}\alpha^{-1}\|\overline{w_t^{\mathrm{md}}} - w^*\|^2 + \frac{3}{2}\gamma^2 \left\|\nabla F(\overline{w_t^{\mathrm{md}}})\right\|^2
$$

$$
- 2\gamma\left(1 + \frac{1}{2}\alpha^{-1}\right)\left\langle \nabla F(\overline{w_t^{\mathrm{md}}}), (1 - \alpha^{-1}(1 - \beta^{-1}))\overline{w_t} + \alpha^{-1}(1 - \beta^{-1})\overline{w_t^{\mathrm{ag}}} - w^*\right\rangle
$$

$$
+ \gamma^2(1 + 2\alpha)\left\|\nabla F(\overline{w_t^{\mathrm{md}}}) - \frac{1}{M}\sum_{m=1}^{M}\nabla F(w_t^{\mathrm{md},m})\right\|^2 + \frac{\gamma^2\sigma^2}{M},
$$

completing the proof of Proposition C.5.                                         $\square$

### C.2.2 Proof of Proposition C.6

*Proof of Proposition C.6.* By definition of the FEDAC procedure we have

$$
\overline{w_{t+1}^{\mathrm{ag}}} = \overline{w_t^{\mathrm{md}}} - \eta \cdot \frac{1}{M}\sum_{m=1}^{M}\nabla f(w_t^{\mathrm{md},m}; \xi_t^m),
$$

and thus by $L$-smoothness (Assumption 1(b)) we obtain

$$
F(\overline{w_{t+1}^{\mathrm{ag}}}) \leq F(\overline{w_t^{\mathrm{md}}}) - \eta\left\langle \nabla F(\overline{w_t^{\mathrm{md}}}), \frac{1}{M}\sum_{m=1}^{M}\nabla f(w_t^{\mathrm{md},m}; \xi_t^m)\right\rangle + \frac{\eta^2 L}{2}\left\|\frac{1}{M}\sum_{m=1}^{M}\nabla f(w_t^{\mathrm{md},m}; \xi_t^m)\right\|^2.
$$

Taking conditional expectation, and by bounded variance (Assumption 1(c))

$$
\mathbb{E}\left[F(\overline{w_{t+1}^{\mathrm{ag}}})|\mathcal{F}_t\right] \leq F(\overline{w_t^{\mathrm{md}}}) - \eta\left\langle \nabla F(\overline{w_t^{\mathrm{md}}}), \frac{1}{M}\sum_{m=1}^{M}\nabla F(w_t^{\mathrm{md},m})\right\rangle + \frac{\eta^2 L}{2}\left\|\frac{1}{M}\sum_{m=1}^{M}\nabla F(w_t^{\mathrm{md},m})\right\|^2 + \frac{\eta^2 L\sigma^2}{2M}.
$$
(C.21)

By polarization identity we have

$$
\left\langle \nabla F(\overline{w_t^{\mathrm{md}}}), \frac{1}{M}\sum_{m=1}^{M}\nabla F(w_t^{\mathrm{md},m})\right\rangle
$$

$$
= \frac{1}{2}\left(\left\|\nabla F(\overline{w_t^{\mathrm{md}}})\right\|^2 + \left\|\frac{1}{M}\sum_{m=1}^{M}\nabla F(w_t^{\mathrm{md},m})\right\|^2 - \left\|\nabla F(\overline{w_t^{\mathrm{md}}}) - \frac{1}{M}\sum_{m=1}^{M}\nabla F(w_t^{\mathrm{md},m})\right\|^2\right).
$$
(C.22)

Combining Eqs. (C.21) and (C.22) gives

$$
\mathbb{E}\left[F(\overline{w_{t+1}^{\mathrm{ag}}})|\mathcal{F}_t\right]
$$

$$
= F(\overline{w_t^{\mathrm{md}}}) - \frac{1}{2}\eta\left\|\nabla F(\overline{w_t^{\mathrm{md}}})\right\|^2 + \frac{1}{2}\eta\left\|\nabla F(\overline{w_t^{\mathrm{md}}}) - \frac{1}{M}\sum_{m=1}^{M}\nabla F(w_t^{\mathrm{md},m})\right\|^2
$$

$$
- \frac{1}{2}\eta(1 - \eta L)\left\|\frac{1}{M}\sum_{m=1}^{M}\nabla F(w_t^{\mathrm{md},m})\right\|^2 + \frac{\eta^2 L\sigma^2}{2M}
$$

$$
\leq F(\overline{w_t^{\mathrm{md}}}) - \frac{1}{2}\eta\left\|\nabla F(\overline{w_t^{\mathrm{md}}})\right\|^2 + \frac{1}{2}\eta\left\|\nabla F(\overline{w_t^{\mathrm{md}}}) - \frac{1}{M}\sum_{m=1}^{M}\nabla F(w_t^{\mathrm{md},m})\right\|^2 + \frac{\eta^2 L\sigma^2}{2M}, \quad \text{(C.23)}
$$

where the last inequality is due to the assumption that $\eta \leq \frac{1}{L}$.

Now we relate $F(\overline{w_t^{\mathrm{md}}})$ and $F(\overline{w_t^{\mathrm{ag}}})$ as follows

$$
\begin{aligned}
& F(\overline{w_t^{\mathrm{md}}}) - F^* \\
&= \left(1 - \frac{1}{2}\alpha^{-1}\right)\left(F(\overline{w_t^{\mathrm{ag}}}) - F^*\right) + \left(1 - \frac{1}{2}\alpha^{-1}\right)\left(F(\overline{w_t^{\mathrm{md}}}) - F(\overline{w_t^{\mathrm{ag}}})\right) + \frac{1}{2}\alpha^{-1}\left(F(\overline{w_t^{\mathrm{md}}}) - F^*\right) \\
&\le \left(1 - \frac{1}{2}\alpha^{-1}\right)\left(F(\overline{w_t^{\mathrm{ag}}}) - F^*\right) + \left(1 - \frac{1}{2}\alpha^{-1}\right)\left\langle\nabla F(\overline{w_t^{\mathrm{md}}}), \overline{w_t^{\mathrm{md}}} - \overline{w_t^{\mathrm{ag}}}\right\rangle \\
&\quad + \frac{1}{2}\alpha^{-1}\left(\left\langle\nabla F(\overline{w_t^{\mathrm{md}}}), \overline{w_t^{\mathrm{md}}} - w^*\right\rangle - \frac{\mu}{2}\left\|\overline{w_t^{\mathrm{md}}} - w^*\right\|^2\right) \quad (\mu\text{-strong convexity}) \\
&= \left(1 - \frac{1}{2}\alpha^{-1}\right)\left(F(\overline{w_t^{\mathrm{ag}}}) - F^*\right) - \frac{1}{4}\mu\alpha^{-1}\left\|\overline{w_t^{\mathrm{md}}} - w^*\right\|^2 \\
&\quad + \frac{1}{2}\alpha^{-1}\left\langle\nabla F(\overline{w_t^{\mathrm{md}}}), 2\alpha\overline{w_t^{\mathrm{md}}} - (2\alpha-1)\overline{w_t^{\mathrm{ag}}} - w^*\right\rangle \quad (\text{rearranging}) \\
&= \left(1 - \frac{1}{2}\alpha^{-1}\right)\left(F(\overline{w_t^{\mathrm{ag}}}) - F^*\right) - \frac{1}{4}\mu\alpha^{-1}\left\|\overline{w_t^{\mathrm{md}}} - w^*\right\|^2 \\
&\quad + \frac{1}{2}\alpha^{-1}\left\langle\nabla F(\overline{w_t^{\mathrm{md}}}), 2\alpha\beta^{-1}\overline{w_t} + (1 - 2\alpha\beta^{-1})\overline{w_t^{\mathrm{ag}}} - w^*\right\rangle, \qquad\qquad (\text{C.24})
\end{aligned}
$$

where the last equality is due to the definition of $\overline{w_t^{\mathrm{md}}}$.

Plugging Eq. (C.24) back to Eq. (C.23) yields

$$
\begin{aligned}
& \mathbb{E}\left[F(\overline{w_{t+1}^{\mathrm{ag}}}) - F^* \middle| \mathcal{F}_t\right] \\
&\le \left(1 - \frac{1}{2}\alpha^{-1}\right)\left(F(\overline{w_t^{\mathrm{ag}}}) - F^*\right) - \frac{1}{4}\mu\alpha^{-1}\left\|\overline{w_t^{\mathrm{md}}} - w^*\right\|^2 - \frac{1}{2}\eta\left\|\nabla F(\overline{w_t^{\mathrm{md}}})\right\|^2 \\
&\quad + \frac{1}{2}\alpha^{-1}\left\langle\nabla F(\overline{w_t^{\mathrm{md}}}), 2\alpha\beta^{-1}\overline{w_t} + (1 - 2\alpha\beta^{-1})\overline{w_t^{\mathrm{ag}}} - w^*\right\rangle + \frac{1}{2}\eta\left\|\nabla F(\overline{w_t^{\mathrm{md}}}) - \frac{1}{M}\sum_{m=1}^M \nabla F(w_t^{\mathrm{md},m})\right\|^2 + \frac{\eta^2 L\sigma^2}{2M},
\end{aligned}
$$

completing the proof of Proposition C.6. $\qquad\qquad\square$

## C.3 Discrepancy overhead bound for FEDAC-II: Proof of Lemma C.3

In this subsection we prove Lemma C.3 regarding the regarding the growth of discrepancy overhead introduced in Lemma C.2. The core of the proof is the $4^{\mathrm{th}}$-order stability of FEDAC-II. Note that most of the analysis in this subsection follows closely with the analysis on FEDAC-I (see Section B.3), but the analysis is technically more complicated.

We will reuse a set of notations defined in Section B.3, which we restate here for clearance. Let $m_1, m_2 \in [M]$ be two arbitrary distinct machines. For any timestep $t$, denote $\Delta_t := w_t^{m_1} - w_t^{m_2}$, $\Delta_t^{\mathrm{ag}} := w_t^{\mathrm{ag},m_1} - w_t^{\mathrm{ag},m_2}$ and $\Delta_t^{\mathrm{md}} := w_t^{\mathrm{md},m_1} - w_t^{\mathrm{md},m_2}$ be the corresponding vector differences. Let $\Delta_t^\varepsilon = \varepsilon_t^{m_1} - \varepsilon_t^{m_2}$, where $\varepsilon_t^m := \nabla f(w_t^{\mathrm{md},m}; \xi_t^m) - \nabla F(w_t^{\mathrm{md},m})$ be the bias of the gradient oracle of the $m$-th worker evaluated at $w_t^{\mathrm{md}}$.

The proof of Lemma C.3 is based on the following propositions.

The following Proposition C.7 studies the growth of $\begin{bmatrix}\Delta_t^{\mathrm{ag}} \\ \Delta_t\end{bmatrix}$ at each step. Proposition C.7 is analogous to Proposition B.8, but the $\mathcal{A}$ is different. Note that Proposition C.7 requires only Assumption 1.

**Proposition C.7.** *Let $F$ be $\mu > 0$-strongly convex, assume Assumption 1 and assume the same hyperparameter choice is taken as in Lemma C.3 (namely $\alpha = \frac{3}{2\gamma\mu} - \frac{1}{2}$, $\beta = \frac{2\alpha^2-1}{\alpha-1}$, $\gamma \in [\eta, \sqrt{\frac{\eta}{\mu}}]$, $\eta \in (0, \frac{1}{L}]$). Suppose $t+1$ is not a synchronization gap, then there exists a matrix $H_t$ such that $\mu I \preceq H_t \preceq LI$ satisfying*

$$
\begin{bmatrix}\Delta_{t+1}^{\mathrm{ag}} \\ \Delta_{t+1}\end{bmatrix} = \mathcal{A}(\mu, \gamma, \eta, H_t)\begin{bmatrix}\Delta_t^{\mathrm{ag}} \\ \Delta_t\end{bmatrix} - \begin{bmatrix}\eta I \\ \gamma I\end{bmatrix}\Delta_t^\varepsilon,
$$

where $\mathcal{A}(\mu, \gamma, \eta, H)$ is a matrix-valued function defined as

$$\mathcal{A}(\mu, \gamma, \eta, H) = \frac{1}{9 - \gamma\mu(6 + \gamma\mu)} \begin{bmatrix} (3 - \gamma\mu)(3 - 2\gamma\mu)(I - \eta H) & 3\gamma\mu(1 - \gamma\mu)(I - \eta H) \\ (3 - 2\gamma\mu)(2\gamma\mu - (3 - \gamma\mu)\gamma H) & 3(1 - \gamma\mu)((3 - \gamma\mu)I - \gamma^2\mu H) \end{bmatrix}.$$
(C.25)

The proof of Proposition C.7 is almost identical with Proposition B.8 except the choice of $\alpha$ and $\beta$ are different. We include this proof in Section C.3.1 for completeness.

The following Proposition C.8 studies the uniform norm bound of $\mathcal{A}$ under the proposed transformation $\mathcal{X}$. The transformation $\mathcal{X}$ is the same as the one studied in FEDAC-I, which we restate here for the ease of reference. The bound is also similar to the corresponding bound for on FEDAC-I as shown in Proposition B.9, though the proof is technically more complicated due to the complexity of $\mathcal{A}$. We defer the proof of Proposition C.8 to Section C.3.2.

**Proposition C.8** (Uniform norm bound of $\mathcal{A}$ under transformation $\mathcal{X}$). *Let $\mathcal{A}(\mu, \gamma, \eta, H)$ be defined as in Eq. (C.25). and assume $\mu > 0$, $\gamma \in [\eta, \sqrt{\frac{\eta}{\mu}}]$, $\eta \in (0, \frac{1}{L}]$. Then the following uniform norm bound holds*

$$\sup_{\mu I \preceq H \preceq LI} \left\| \mathcal{X}(\gamma, \eta)^{-1} \mathcal{A}(\mu, \gamma, \eta, H) \mathcal{X}(\gamma, \eta) \right\| \leq \begin{cases} 1 + \frac{\gamma^2\mu}{\eta} & \text{if } \gamma \in \left(\eta, \sqrt{\frac{\eta}{\mu}}\right], \\ 1 & \text{if } \gamma = \eta, \end{cases}$$

*where $\mathcal{X}(\gamma, \eta)$ is a matrix-valued function defined as*

$$\mathcal{X}(\gamma, \eta) := \begin{bmatrix} \frac{\eta}{\gamma}I & 0 \\ I & I \end{bmatrix}.$$
(C.26)

Propositions C.7 and C.8 suggest the one-step growth of $\left\| \mathcal{X}(\gamma, \eta)^{-1} \begin{bmatrix} \Delta_t^{\text{ag}} \\ \Delta_t \end{bmatrix} \right\|^4$ as follows.

**Proposition C.9.** *In the same setting of Lemma C.3, the following inequality holds (for all possible $t$)*

$$\sqrt{\mathbb{E}\left[ \left\| \mathcal{X}(\gamma, \eta)^{-1} \begin{bmatrix} \Delta_{t+1}^{\text{ag}} \\ \Delta_{t+1} \end{bmatrix} \right\|^4 \middle| \mathcal{F}_t \right]} \leq 7\gamma^2\sigma^2 + \left\| \mathcal{X}(\gamma, \eta)^{-1} \begin{bmatrix} \Delta_t^{\text{ag}} \\ \Delta_t \end{bmatrix} \right\|^2 \cdot \begin{cases} \left(1 + \frac{\gamma^2\mu}{\eta}\right)^2 & \text{if } \gamma \in \left(\eta, \sqrt{\frac{\eta}{\mu}}\right], \\ 1 & \text{if } \gamma = \eta, \end{cases}$$

*where $\mathcal{X}$ is the matrix-valued function defined in Eq. (C.26).*

We defer the proof of Proposition C.9 to Section C.3.3.

The following Proposition C.10 links the discrepancy overhead we wish to bound for Lemma C.3 with the quantity analyzed in Proposition C.9 via $3^{\text{rd}}$-order-smoothness (Assumption 2(a)). The proof of Proposition C.10 is deferred to Section C.3.4.

**Proposition C.10.** *In the same setting of Lemma C.3, the following inequality holds (for all possible $t$)*

$$\left\| \nabla F(\overline{w_t^{\text{md}}}) - \frac{1}{M} \sum_{m=1}^{M} \nabla F(w_t^{\text{md},m}) \right\|^2 \leq \frac{289\eta^4 Q^2}{324\gamma^4} \left\| \mathcal{X}(\gamma, \eta)^{-1} \begin{bmatrix} \Delta_t^{\text{ag}} \\ \Delta_t \end{bmatrix} \right\|^4,$$

*where $\mathcal{X}$ is the matrix-valued function defined in Eq. (C.26).*

We are ready to complete the proof of Lemma C.3.

*Proof of Lemma C.3.* Let $t_0$ be the latest synchronized step prior to $t$. Applying Proposition C.9 gives

$$\sqrt{\mathbb{E}\left[ \left\| \mathcal{X}(\gamma, \eta)^{-1} \begin{bmatrix} \Delta_{t+1}^{\text{ag}} \\ \Delta_{t+1} \end{bmatrix} \right\|^4 \middle| \mathcal{F}_{t_0} \right]}$$

$$\leq 7\gamma^2\sigma^2 + \sqrt{\mathbb{E}\left[ \left\| \mathcal{X}(\gamma, \eta)^{-1} \begin{bmatrix} \Delta_t^{\text{ag}} \\ \Delta_t \end{bmatrix} \right\|^2 \middle| \mathcal{F}_{t_0} \right]} \cdot \begin{cases} \left(1 + \frac{\gamma^2\mu}{\eta}\right)^2 & \text{if } \gamma \in \left(\eta, \sqrt{\frac{\eta}{\mu}}\right], \\ 1 & \text{if } \gamma = \eta. \end{cases}$$

Telescoping from $t_0$ to $t$ gives (note that $\Delta^{\mathrm{ag}}_{t_0} = \Delta_{t_0} = 0$)

$$\mathbb{E}\left[\left\|\mathcal{X}(\gamma,\eta)^{-1}\begin{bmatrix}\Delta^{\mathrm{ag}}_t\\\Delta_t\end{bmatrix}\right\|^4 \middle| \mathcal{F}_{t_0}\right] \le 49\gamma^4\sigma^4(t-t_0)^2 \cdot \begin{cases}\left(1+\frac{\gamma^2\mu}{\eta}\right)^{4(t-t_0)} & \text{if } \gamma \in \left(\eta, \sqrt{\frac{\eta}{\mu}}\right], \\ 1 & \text{if } \gamma = \eta\end{cases}$$

$$\le 49\gamma^4\sigma^4 K^2 \cdot \begin{cases}\left(1+\frac{\gamma^2\mu}{\eta}\right)^{4K} & \text{if } \gamma \in \left(\eta, \sqrt{\frac{\eta}{\mu}}\right], \\ 1 & \text{if } \gamma = \eta,\end{cases}$$

where the last inequality is due to $t - t_0 \le K$ since $K$ is the maximum synchronization interval.

Consequently, by Proposition C.10 we have

$$\mathbb{E}\left[\left\|\nabla F(\overline{w^{\mathrm{md}}_t}) - \frac{1}{M}\sum_{m=1}^M \nabla F(w^{\mathrm{md},m}_t)\right\|^2 \middle| \mathcal{F}_{t_0}\right] \le \frac{289\eta^4 Q^2}{324\gamma^4}\mathbb{E}\left[\left\|\mathcal{X}(\gamma,\eta)^{-1}\begin{bmatrix}\Delta^{\mathrm{ag}}_t\\\Delta_t\end{bmatrix}\right\|^4 \middle| \mathcal{F}_{t_0}\right]$$

$$\le \begin{cases}44\eta^4 Q^2 K^2 \sigma^4\left(1+\frac{\gamma^2\mu}{\eta}\right)^{4K} & \text{if } \gamma \in \left(\eta, \sqrt{\frac{\eta}{\mu}}\right], \\ 44\eta^4 Q^2 K^2 \sigma^4 & \text{if } \gamma = \eta,\end{cases}$$

where in the last inequality we used the estimate that $\frac{289}{324} \cdot 49 < 44$. □

### C.3.1 Proof of Proposition C.7

*Proof of Proposition C.7.* The proof of Proposition C.7 follows instantly by plugging $\alpha = \frac{3}{2\gamma\mu} - \frac{1}{2}$, $\beta = \frac{2\alpha^2-1}{\alpha-1} = \frac{9-\gamma\mu(6+\gamma\mu)}{3\gamma\mu(1-\gamma\mu)}$ to the general claim on FEDAC Claim B.12:

$$\begin{bmatrix}(1-\beta^{-1})(I-\eta H) & \beta^{-1}(I-\eta H)\\(1-\beta^{-1})(\alpha^{-1}-\gamma H) & \beta^{-1}(\alpha^{-1}I-\gamma H)+(1-\alpha^{-1})I\end{bmatrix}$$

$$= \frac{1}{9-\gamma\mu(6+\gamma\mu)}\begin{bmatrix}(3-\gamma\mu)(3-2\gamma\mu)(I-\eta H) & 3\gamma\mu(1-\gamma\mu)(I-\eta H)\\(3-2\gamma\mu)(2\gamma\mu-(3-\gamma\mu)\gamma H) & 3(1-\gamma\mu)((3-\gamma\mu)I-\gamma^2\mu H)\end{bmatrix}.$$

□

### C.3.2 Proof of Proposition C.8: uniform norm bound

The proof idea of this proposition is very similar to Proposition B.9, though more complicated technically.

*Proof.* Define another matrix-valued function $\mathcal{B}$ as
$$\mathcal{B}(\mu,\gamma,\eta,H) := \mathcal{X}(\gamma,\eta)^{-1}\mathcal{A}(\mu,\gamma,\eta,H)\mathcal{X}(\gamma,\eta).$$
Since $\mathcal{X}(\gamma,\eta)^{-1} = \begin{bmatrix}\frac{\gamma}{\eta}I & 0\\-\frac{\gamma}{\eta}I & I\end{bmatrix}$ we have

$\mathcal{B}(\mu,\gamma,\eta,H)$

$$= \frac{1}{(9-(6+\gamma\mu)\gamma\mu)\eta}\begin{bmatrix}\left(3\gamma^2\mu(1-\gamma\mu)+\eta(3-\gamma\mu)(3-2\gamma\mu)\right)(I-\eta H) & 3\gamma^2\mu(1-\gamma\mu)(I-\eta H)\\-(\gamma-\eta)\left(3\gamma+6\eta-\gamma\mu(3\gamma+4\eta)\right)I & 3(1-\gamma\mu)\left(3\eta-\gamma\mu(\gamma+\eta)\right)I\end{bmatrix}.$$

Define the four blocks of $\mathcal{B}(\mu,\gamma,\eta,H)$ as $\mathcal{B}_{11}(\mu,\gamma,\eta,H)$, $\mathcal{B}_{12}(\mu,\gamma,\eta,H)$, $\mathcal{B}_{21}(\mu,\gamma,\eta)$, $\mathcal{B}_{22}(\mu,\gamma,\eta)$ (note that the lower two blocks do not involve $H$), namely

$$\mathcal{B}_{11}(\mu,\gamma,\eta,H) = \frac{3\gamma^2\mu(1-\gamma\mu)+\eta(3-\gamma\mu)(3-2\gamma\mu)}{(9-(6+\gamma\mu)\gamma\mu)\eta}(I-\eta H),$$

$$\mathcal{B}_{12}(\mu,\gamma,\eta,H) = \frac{3\gamma^2\mu(1-\gamma\mu)}{(9-(6+\gamma\mu)\gamma\mu)\eta}(I-\eta H),$$

$$\mathcal{B}_{21}(\mu,\gamma,\eta) = -\frac{(\gamma-\eta)\mu\left(3\gamma+6\eta-\gamma\mu(3\gamma+4\eta)\right)}{(9-(6+\gamma\mu)\gamma\mu)\eta}I,$$

$$\mathcal{B}_{22}(\mu,\gamma,\eta) = \frac{3(1-\gamma\mu)\left(3\eta-\gamma\mu(\gamma+\eta)\right)}{(9-(6+\gamma\mu)\gamma\mu)\eta}I.$$

**Case I:** $\eta < \gamma \le \sqrt{\frac{\eta}{\mu}}$. Since $\gamma\mu \le 1$, we know that the common denominator

$$(9 - (6 + \gamma\mu)\gamma\mu)\eta \ge 2\eta > 0.$$

Now we bound the operator norm of each block as follows.

**Bound for $\|\mathcal{B}_{11}\|$.** Since $3\gamma^2\mu(1-\gamma\mu) + \eta(3-\gamma\mu)(3-2\gamma\mu) \ge 0$, we have $\mathcal{B}_{11} \succeq 0$, and therefore

$$
\begin{aligned}
&\|\mathcal{B}_{11}(\mu,\gamma,\eta,H)\| \\
\le &\frac{3\gamma^2\mu(1-\gamma\mu) + \eta(3-\gamma\mu)(3-2\gamma\mu)}{(9-(6+\gamma\mu)\gamma\mu)\eta}(1-\eta\mu) \\
\le &\frac{3\gamma^2\mu(1-\gamma\mu) + \eta(3-\gamma\mu)(3-2\gamma\mu)}{(9-(6+\gamma\mu)\gamma\mu)\eta} \\
= &1 + \frac{3(\gamma-\eta)\gamma\mu(1-\gamma\mu)}{(9-(6+\gamma\mu)\gamma\mu)\eta} \\
\le &1 + \frac{3\gamma^2\mu}{\eta} \cdot \frac{1-\gamma\mu}{9-6\gamma\mu-\gamma^2\mu^2} && \text{(since } \gamma - \eta \le \gamma\text{)} \\
\le &1 + \frac{\gamma^2\mu}{3\eta}, && \text{(C.27)}
\end{aligned}
$$

where the last inequality is due to $\frac{1-\gamma\mu}{9-6\gamma\mu-\gamma^2\mu^2} \le \frac{1}{9}$ since $\gamma\mu \le 1$.

**Bound for $\|\mathcal{B}_{12}\|$.** Similarly we have

$$\|\mathcal{B}_{12}(\mu,\gamma,\eta,H)\| \le \frac{3\gamma^2\mu(1-\gamma\mu)}{(9-(6+\gamma\mu)\gamma\mu)\eta}(1-\eta\mu) \le \frac{3\gamma^2\mu}{\eta} \cdot \frac{1-\gamma\mu}{9-(6+\gamma\mu)\gamma\mu} \le \frac{\gamma^2\mu}{3\eta}, \quad \text{(C.28)}$$

where the last inequality is due to $\frac{1-\gamma\mu}{9-6\gamma\mu-\gamma^2\mu^2} \le \frac{1}{9}$ since $\gamma\mu \le 1$.

**Bound for $\|\mathcal{B}_{21}\|$.** Since $\gamma \ge \eta$, we have $(\gamma-\eta)\mu(3\gamma + 6\eta - \gamma\mu(3\gamma + 4\eta)) \ge 0$. Note that

$$
\begin{aligned}
&(\gamma - \eta)(3\gamma + 6\eta - \gamma\mu(3\gamma + 4\eta)) \\
= &3\gamma^2 + 3\gamma\eta - 6\eta^2 - \gamma\mu(3\gamma^2 + \gamma\eta - 4\eta^2) \\
= &4\gamma^2 - 3\gamma^3\mu - (\gamma^2 - 3\gamma\eta + 6\eta^2 + \gamma^2\mu\eta - 4\eta^2\gamma\mu),
\end{aligned}
$$

and

$$
\begin{aligned}
&\gamma^2 - 3\gamma\eta + 6\eta^2 + \gamma^2\mu\eta - 4\eta^2\gamma\mu \\
\ge &\gamma^2 - 3\gamma\eta + 6\eta^2 - 3\eta^2\gamma\mu && \text{(since } \eta \le \gamma\text{)} \\
\ge &\gamma^2 - 3\gamma\eta + 3\eta^2 && \text{(since } \gamma\mu \le 1\text{)} \\
\ge &0. && \text{(AM-GM inequality)}
\end{aligned}
$$

Consequently,

$$(\gamma - \eta)\mu(3\gamma + 6\eta - \gamma\mu(3\gamma + 4\eta)) \le 4\gamma^2\mu - 3\gamma^3\mu^2. \quad \text{(C.29)}$$

It follows that

$$
\begin{aligned}
\|\mathcal{B}_{21}(\mu,\gamma,\eta)\| = &\frac{\mu(\gamma-\eta)(3\gamma + 6\eta - \gamma\mu(3\gamma + 4\eta))}{(9-(6+\gamma\mu)\gamma\mu)\eta} \\
\le &\frac{4\gamma^2\mu - 3\gamma^3\mu^2}{(9-(6+\gamma\mu)\gamma\mu)\eta} && \text{(by Eq. (C.29))} \\
= &\frac{\gamma^2\mu}{\eta} \cdot \frac{4-3\gamma\mu}{9-6\gamma\mu-\gamma^2\mu^2} \le \frac{2\gamma^2\mu}{3\eta}. && \text{(C.30)}
\end{aligned}
$$

where the last inequality is due to $\frac{4-3\gamma\mu}{9-6\gamma\mu-\gamma^2\mu^2} \le \frac{2}{3}$ since $\gamma\mu \le 1$.

**Bound for $\mathcal{B}_{22}$.** Since $\gamma > \eta$ and $\gamma^2\mu \le \eta$, we have $3\eta - \gamma\mu(\gamma + \eta) \ge 3\eta - 2\gamma^2\mu \ge \eta$. Thus $\mathcal{B}_{22} \succeq 0$, which implies

$$\|\mathcal{B}_{22}(\mu, \gamma, \eta)\| = \frac{3(1 - \gamma\mu)\,(3\eta - \gamma\mu(\gamma + \eta))}{(9 - (6 + \gamma\mu)\gamma\mu)\eta} = 1 + \frac{\gamma\mu\,(-6\eta - 3\gamma + \gamma\mu(3\gamma + 4\eta))}{(9 - (6 + \gamma\mu)\gamma\mu)\eta} \le 1.$$

(C.31)

The operator norm of block matrix $\mathcal{B}$ can be bounded via its blocks via Lemma G.1 as

$$\begin{aligned}
&\mathcal{B}(\mu, \gamma, \eta, H) \\
&\le \max\left\{\|\mathcal{B}_{11}(\mu, \gamma, \eta, H)\|, \|\mathcal{B}_{22}(\mu, \gamma, \eta)\|\right\} + \max\left\{\|\mathcal{B}_{12}(\mu, \gamma, \eta, H)\|, \|\mathcal{B}_{21}(\mu, \gamma, \eta)\|\right\}
\end{aligned}$$

(Lemma G.1)

$$\le \max\left\{1 + \frac{\gamma^2\mu}{3\eta}, 1\right\} + \max\left\{\frac{\gamma^2\mu}{3\eta}, \frac{2\gamma^2\mu}{3\eta}\right\} \le 1 + \frac{\gamma^2\mu}{\eta}.$$

(Eqs. (C.27), (C.28), (C.30) and (C.31))

**Case II: $\gamma = \eta$.** In this case we have

$$\begin{aligned}
\|\mathcal{B}_{11}(\mu, \gamma, \eta, H)\| &\le 1 - \eta\mu, \\
\|\mathcal{B}_{12}(\mu, \gamma, \eta, H)\| &\le \frac{3\eta\mu - 6\eta^2\mu^2 + 3\eta^3\mu^3}{9 - 6\eta\mu - \eta^2\mu^2}, \\
\|\mathcal{B}_{21}(\mu, \gamma, \eta)\| &= 0, \\
\|\mathcal{B}_{22}(\mu, \gamma, \eta)\| &= \frac{9 - 15\eta\mu + 6\eta^2\mu^2}{9 - 6\eta\mu - \eta^2\mu^2} = 1 - \frac{9\eta\mu - 7\eta^2\mu^2}{9 - 6\eta\mu - \eta^2\mu^2}.
\end{aligned}$$

Similarly the operator norm of block matrix $\mathcal{B}$ can be bounded via its blocks via Lemma G.1 as

$$\begin{aligned}
&\mathcal{B}(\mu, \gamma, \eta, H) \\
&\le \max\left\{\|\mathcal{B}_{11}(\mu, \gamma, \eta, H)\|, \|\mathcal{B}_{22}(\mu, \gamma, \eta)\|\right\} + \max\left\{\|\mathcal{B}_{12}(\mu, \gamma, \eta, H)\|, \|\mathcal{B}_{21}(\mu, \gamma, \eta)\|\right\}
\end{aligned}$$

(Lemma G.1)

$$\le \max\left\{1 - \eta\mu + \frac{3\eta\mu - 6\eta^2\mu^2 + 3\eta^3\mu^3}{9 - 6\eta\mu - \eta^2\mu^2}, \frac{9 - 15\eta\mu + 6\eta^2\mu^2}{9 - 6\eta\mu - \eta^2\mu^2} + \frac{3\eta\mu - 6\eta^2\mu^2 + 3\eta^3\mu^3}{9 - 6\eta\mu - \eta^2\mu^2}\right\}$$

$$\le \max\left\{1 - \frac{6\eta\mu - 4\eta^3\mu^3}{9 - 6\eta\mu - \eta^2\mu^2}, 1 - \frac{6\eta\mu - \eta^2\mu^2 - 3\eta^3\mu^3}{9 - 6\eta\mu - \eta^2\mu^2}\right\} \le 1.$$

Summarizing the above two cases completes the proof of Proposition C.8. $\qquad\square$

### C.3.3  Proof of Proposition C.9

In this section we apply Propositions C.7 and C.8 to establish Proposition C.9.

*Proof of Proposition C.9.* If $t+1$ is a synchronized step, then the bound trivially holds since $\Delta_{t+1}^{\mathrm{ag}} = \Delta_{t+1} = 0$ due to synchronization.

From now on assume $t + 1$ is not a synchronized step, for which Proposition C.7 is applicable. Multiplying $\mathcal{X}(\gamma, \eta)^{-1}$ to the left on both sides of Proposition C.7 gives (we omit the details since the reasoning is the same as in the proof of Proposition B.10.

$$\mathcal{X}(\gamma, \eta)^{-1}\begin{bmatrix} \Delta_{t+1}^{\mathrm{ag}} \\ \Delta_{t+1} \end{bmatrix} = \mathcal{X}(\gamma, \eta)^{-1}\mathcal{A}(\mu, \gamma, \eta, H_t)\mathcal{X}(\gamma, \eta)^{-1}\left(\mathcal{X}(\gamma, \eta)\begin{bmatrix} \Delta_t^{\mathrm{ag}} \\ \Delta_t \end{bmatrix}\right) - \begin{bmatrix} \gamma I \\ 0 \end{bmatrix}\Delta_t^{\varepsilon}. \quad \text{(C.32)}$$

Before we proceed, we introduce a few more notations to simplify the discussion. Denote the shortcut $\mathcal{B}_t := \mathcal{X}(\gamma, \eta)^{-1}\mathcal{A}(\mu, \gamma, \eta, H_t)\mathcal{X}(\gamma, \eta)$, $\mathcal{X} = \mathcal{X}(\gamma, \eta)$, $\tilde{\Delta}_t = \mathcal{X}^{-1}\begin{bmatrix} \Delta_t^{\mathrm{ag}} \\ \Delta_t \end{bmatrix}$, and $\tilde{\Delta}_t^{\varepsilon} = \begin{bmatrix} \gamma I \\ 0 \end{bmatrix}\Delta_t^{\varepsilon}$.

Then Eq. (C.32) becomes $\tilde{\Delta}_{t+1} = \mathcal{B}_t \tilde{\Delta}_t - \tilde{\Delta}_t^\varepsilon$. Thus

$$\mathbb{E}\left[\|\tilde{\Delta}_{t+1}\|^4 | \mathcal{F}_t\right] = \mathbb{E}\left[\|\mathcal{B}_t \tilde{\Delta}_t - \tilde{\Delta}_t^\varepsilon\|^4 | \mathcal{F}_t\right] \qquad \text{(by Proposition C.7)}$$

$$= \mathbb{E}\left[\left(\|\mathcal{B}_t \tilde{\Delta}_t\|^2 + \|\tilde{\Delta}_t^\varepsilon\|^2 - 2\langle \mathcal{B}_t \tilde{\Delta}_t, \tilde{\Delta}_t^\varepsilon \rangle\right)^2\right]$$

$$= \|\mathcal{B}_t \tilde{\Delta}_t\|^4 + \mathbb{E}\left[\|\tilde{\Delta}_t^\varepsilon\|^4 | \mathcal{F}_t\right] + 4\mathbb{E}\left[\langle \mathcal{B}_t \tilde{\Delta}_t, \tilde{\Delta}_t^\varepsilon \rangle^2 | \mathcal{F}_t\right] + 2\|\mathcal{B}_t \tilde{\Delta}_t\|^2 \mathbb{E}\left[\|\tilde{\Delta}_t^\varepsilon\|^2 | \mathcal{F}_t\right]$$

$$\quad - 4\|\mathcal{B}_t \tilde{\Delta}_t\|^2 \mathbb{E}\left[\langle \mathcal{B}_t \tilde{\Delta}_t, \tilde{\Delta}_t^\varepsilon \rangle | \mathcal{F}_t\right] - 4\mathbb{E}\left[\|\tilde{\Delta}_t^\varepsilon\|^2 \langle \mathcal{B}_t \tilde{\Delta}_t, \tilde{\Delta}_t^\varepsilon \rangle | \mathcal{F}_t\right]$$

$$= \|\mathcal{B}_t \tilde{\Delta}_t\|^4 + \mathbb{E}\left[\|\tilde{\Delta}_t^\varepsilon\|^4 | \mathcal{F}_t\right] + 4\mathbb{E}\left[\langle \mathcal{B}_t \tilde{\Delta}_t, \tilde{\Delta}_t^\varepsilon \rangle^2 | \mathcal{F}_t\right] + 2\|\mathcal{B}_t \tilde{\Delta}_t\|^2 \mathbb{E}\left[\|\tilde{\Delta}_t^\varepsilon\|^2 | \mathcal{F}_t\right]$$

$$\quad - 4\mathbb{E}\left[\|\tilde{\Delta}_t^\varepsilon\|^2 \langle \mathcal{B}_t \tilde{\Delta}_t, \tilde{\Delta}_t^\varepsilon \rangle | \mathcal{F}_t\right] \qquad \text{(by independence and } \mathbb{E}[\tilde{\Delta}_t^\varepsilon | \mathcal{F}_t] = 0)$$

$$\leq \|\mathcal{B}_t \tilde{\Delta}_t\|^4 + \mathbb{E}\left[\|\tilde{\Delta}_t^\varepsilon\|^4 | \mathcal{F}_t\right] + 6\|\mathcal{B}_t \tilde{\Delta}_t\|^2 \mathbb{E}\left[\|\tilde{\Delta}_t^\varepsilon\|^2 | \mathcal{F}_t\right] + 4\|\mathcal{B}_t \tilde{\Delta}_t\| \mathbb{E}\left[\|\tilde{\Delta}_t^\varepsilon\|^3 | \mathcal{F}_t\right]$$

$$\qquad\qquad\qquad\qquad\qquad\qquad\qquad\qquad\qquad\qquad \text{(Cauchy-Schwarz inequality)}$$

$$\leq \|\mathcal{B}_t \tilde{\Delta}_t\|^4 + 5\mathbb{E}\left[\|\tilde{\Delta}_t^\varepsilon\|^4 | \mathcal{F}_t\right] + 7\|\mathcal{B}_t \tilde{\Delta}_t\|^2 \mathbb{E}\left[\|\tilde{\Delta}_t^\varepsilon\|^2 | \mathcal{F}_t\right] \qquad \text{(AM-GM inequality)}$$

$$\leq \|\mathcal{B}_t \tilde{\Delta}_t\|^4 + 40\gamma^4 \sigma^4 + 14\gamma^2 \sigma^2 \|\mathcal{B}_t \tilde{\Delta}_t\|^2 \qquad \text{(bounded 4}^{\text{th}}\text{ central moment via Lemma G.4)}$$

$$\leq \left(\|\mathcal{B}_t \tilde{\Delta}_t\|^2 + 7\gamma^2 \sigma^2\right)^2 \leq \left(\|\mathcal{B}_t\|^2 \|\tilde{\Delta}_t\|^2 + 7\gamma^2 \sigma^2\right)^2.$$

Applying Proposition C.8,

$$\sqrt{\mathbb{E}\left[\|\tilde{\Delta}_{t+1}\|^4 | \mathcal{F}_t\right]} \leq 7\gamma^2 \sigma^2 + \|\tilde{\Delta}_t\|^2 \cdot \begin{cases} \left(1 + \frac{\gamma^2 \mu}{\eta}\right)^2 & \text{if } \gamma \in \left(\eta, \sqrt{\frac{\eta}{\mu}}\right], \\ 1 & \text{if } \gamma = \eta. \end{cases}$$

Resetting the notations completes the proof. $\qquad\qquad\qquad\qquad\qquad\qquad\qquad\qquad\square$

### C.3.4 Proof of Proposition C.10

In this section we will prove Proposition C.10 in two steps via the following two claims. For both two claims $\mathcal{X}$ stands for the matrix-valued functions defined in Eq. (C.26).

**Claim C.11.** *In the same setting of Lemma C.3, the following inequality holds (for all possible $t$)*

$$\left\|\nabla F(\overline{w_t^{\text{md}}}) - \frac{1}{M}\sum_{m=1}^M \nabla F(w_t^{\text{md},m})\right\|^2 \leq \frac{Q^2}{4}\left\|\mathcal{X}(\gamma,\eta)^\intercal \begin{bmatrix} \frac{9-9\gamma\mu+2\gamma^2\mu^2}{9-6\gamma\mu-\gamma^2\mu^2}I \\ \frac{3\gamma\mu-3\gamma^2\mu^2}{9-6\gamma\mu-\gamma^2\mu^2}I \end{bmatrix}\right\|^4 \left\|\mathcal{X}(\gamma,\eta)^{-1}\begin{bmatrix}\Delta_t^{\text{ag}} \\ \Delta_t\end{bmatrix}\right\|^4.$$

**Claim C.12.** *Assume $\mu > 0$, $\gamma \in [\eta, \sqrt{\frac{\eta}{\mu}}]$, then* $\left\|\mathcal{X}(\gamma,\eta)^\intercal \begin{bmatrix} \frac{9-9\gamma\mu+2\gamma^2\mu^2}{9-6\gamma\mu-\gamma^2\mu^2}I \\ \frac{3\gamma\mu-3\gamma^2\mu^2}{9-6\gamma\mu-\gamma^2\mu^2}I \end{bmatrix}\right\| \leq \frac{\sqrt{17}\eta}{3\gamma}.$

*Proof of Proposition C.10.* Follow trivially with Claims B.13 and C.12 as

$$\left\|\nabla F(\overline{w_t^{\text{md}}}) - \frac{1}{M}\sum_{m=1}^M \nabla F(w_t^{\text{md},m})\right\|^2 \leq \frac{Q^2}{4}\left(\frac{\sqrt{17}\eta}{3\gamma}\right)^4 \left\|\mathcal{X}(\gamma,\eta)^{-1}\begin{bmatrix}\Delta_t^{\text{ag}} \\ \Delta_t\end{bmatrix}\right\|^4$$

$$= \frac{289\eta^4 Q^2}{324\gamma^4}\left\|\mathcal{X}(\gamma,\eta)^{-1}\begin{bmatrix}\Delta_t^{\text{ag}} \\ \Delta_t\end{bmatrix}\right\|^4.$$

$\qquad\qquad\qquad\qquad\qquad\qquad\qquad\qquad\qquad\qquad\qquad\qquad\qquad\qquad\qquad\qquad\qquad\square$

Now we finish the proof of these two claims.

*Proof of Claim C.11.* Helper Lemma G.3 shows that $\left\|\nabla F(\overline{w_t^{\mathrm{md}}}) - \frac{1}{M}\sum_{m=1}^M \nabla F(w_t^{\mathrm{md},m})\right\|^2$ can be bounded by $4^{\mathrm{th}}$-moment of difference:

$$\left\|\nabla F(\overline{w_t^{\mathrm{md}}}) - \frac{1}{M}\sum_{m=1}^M \nabla F(w_t^{\mathrm{md},m})\right\|^2 \leq \frac{Q^2}{4}\cdot\frac{1}{M}\sum_{m=1}^M \|w_t^{\mathrm{md},m} - \overline{w_t^{\mathrm{md}}}\|^4 \qquad \text{(Lemma G.3)}$$

$$\leq \frac{Q^2}{4}\|\Delta_t^{\mathrm{md}}\|^4 \qquad\qquad\qquad\qquad\qquad\qquad\quad \text{(convexity of } \|\cdot\|^4)$$

$$= \frac{Q^2}{4}\left\|\begin{bmatrix}(1-\beta^{-1})I\\ \beta^{-1}I\end{bmatrix}^{\mathsf{T}}\begin{bmatrix}\Delta_t^{\mathrm{ag}}\\ \Delta_t\end{bmatrix}\right\|^4 \qquad\qquad\qquad\quad \text{(definition of ``md'')}$$

$$\leq \frac{Q^2}{4}\left\|\mathcal{X}(\gamma,\eta)^{\mathsf{T}}\begin{bmatrix}(1-\beta^{-1})I\\ \beta^{-1}I\end{bmatrix}\right\|^4\cdot\left\|\mathcal{X}(\gamma,\eta)^{-1}\begin{bmatrix}\Delta_t^{\mathrm{ag}}\\ \Delta_t\end{bmatrix}\right\|^4. \qquad \text{(sub-multiplicativity)}$$

$$= \frac{Q^2}{4}\left\|\begin{bmatrix}\frac{9-9\gamma\mu+2\gamma^2\mu^2}{9-6\gamma\mu-\gamma^2\mu^2}I\\ \frac{3\gamma\mu-3\gamma^2\mu^2}{9-6\gamma\mu-\gamma^2\mu^2}I\end{bmatrix}\right\|^4\cdot\left\|\mathcal{X}(\gamma,\eta)^{-1}\begin{bmatrix}\Delta_t^{\mathrm{ag}}\\ \Delta_t\end{bmatrix}\right\|^4.$$

$\square$

*Proof of Claim C.12.* Direct calculation shows that

$$\mathcal{X}(\gamma,\eta)^{\mathsf{T}}\begin{bmatrix}\frac{9-9\gamma\mu+2\gamma^2\mu^2}{9-6\gamma\mu-\gamma^2\mu^2}I\\ \frac{3\gamma\mu-3\gamma^2\mu^2}{9-6\gamma\mu-\gamma^2\mu^2}I\end{bmatrix} = \begin{bmatrix}\frac{3\gamma^2\mu(1-\gamma\mu)+\eta(3-\gamma\mu)(3-2\gamma\mu)}{\gamma(9-6\gamma\mu-\gamma^2\mu^2)}I\\ \frac{3\gamma^2\mu(1-\gamma\mu)}{\gamma(9-6\gamma\mu-\gamma^2\mu^2)}I\end{bmatrix}.$$

Since $\gamma^2\mu \leq \eta$ and $\gamma\mu \leq 1$, we have

$$0 \leq \frac{3\gamma^2\mu(1-\gamma\mu)+\eta(3-\gamma\mu)(3-2\gamma\mu)}{\gamma(9-6\gamma\mu-\gamma^2\mu^2)} \leq \frac{\eta}{\gamma}\cdot\frac{12-12\gamma\mu+2\gamma^2\mu^2}{9-6\gamma\mu-\gamma^2\mu^2} \leq \frac{4\eta}{3\gamma},$$

and

$$0 \leq \frac{3\gamma^2\mu(1-\gamma\mu)}{\gamma(9-6\gamma\mu-\gamma^2\mu^2)} \leq \frac{\eta}{\gamma}\cdot\frac{3(1-\gamma\mu)}{9-6\gamma\mu-\gamma^2\mu^2} \leq \frac{\eta}{3\gamma}.$$

Consequently,

$$\left\|\begin{bmatrix}\frac{3\gamma^2\mu(1-\gamma\mu)+\eta(3-\gamma\mu)(3-2\gamma\mu)}{\gamma(9-6\gamma\mu-\gamma^2\mu^2)}I\\ \frac{3\gamma^2\mu(1-\gamma\mu)}{\gamma(9-6\gamma\mu-\gamma^2\mu^2)}I\end{bmatrix}\right\| \leq \sqrt{\left(\frac{4\eta}{3\gamma}\right)^2 + \left(\frac{\eta}{3\gamma}\right)^2} \leq \frac{\sqrt{17}\eta}{3\gamma}.$$

$\square$

## C.4 Convergence of FEDAC-II under Assumption 1: Complete version of Theorem 3.1(b)

### C.4.1 Main theorem and lemma

In this subsection we establish the convergence of FEDAC-II under Assumption 1. We will provide a complete, non-asymptotic version of Theorem 3.1(b) and provide the proof.

**Theorem C.13** (Convergence of FEDAC-II under Assumption 1, complete version of Theorem 3.1(b))**.** *Let $F$ be $\mu > 0$ strongly convex, and assume Assumption 1, then for*

$$\eta = \min\left\{\frac{1}{L}, \frac{9K}{\mu T^2}\log^2\left(\mathrm{e} + \min\left\{\frac{\mu M T\Phi_0}{\sigma^2} + \frac{\mu^3 T^4\Phi_0}{L^2 K^3\sigma^2}\right\}\right)\right\},$$

FEDAC-II *yields*

$$\mathbb{E}[\Phi_T] \leq \min\left\{\exp\left(-\frac{\mu T}{3L}\right), \exp\left(-\frac{\mu^{\frac{1}{2}}T}{3L^{\frac{1}{2}}K^{\frac{1}{2}}}\right)\right\}\Phi_0$$

$$+ \frac{4\sigma^2}{\mu M T}\log\left(\mathrm{e} + \frac{\mu M T\Phi_0}{\sigma^2}\right) + \frac{8101 L^2 K^3\sigma^2}{\mu^3 T^4}\log^4\left(\mathrm{e} + \frac{\mu^3 T^4\Phi_0}{L^2 K^3\sigma^2}\right),$$

*where $\Phi_t$ is the "centralized" potential function defined in Eq. (C.1).*

**Remark.** *The simplified version Theorem 3.1(b) in the main body can be obtained by replacing $K$ with $T/R$ and upper bound $\Phi_0$ by $LD_0^2$.*

Note that most of the results established towards Theorem C.1 can be recycled as long as it does not assume Assumption 2. In particular, we will reuse the perturbed iterate analysis Lemma C.2, and provide an alternative version of discrepancy overhead bounds, as shown in Lemma C.14. The only difference is that now we use $L$-smoothness to bound the discrepancy term.

**Lemma C.14** (Discrepancy overhead bounds). *Let $F$ be $\mu > 0$-strongly convex, and assume Assumption 1, then for $\alpha = \frac{3}{2\gamma\mu} - \frac{1}{2}$, $\beta = \frac{2\alpha^2 - 1}{\alpha - 1}$, $\gamma \in [\eta, \sqrt{\frac{\eta}{\mu}}]$, $\eta \in (0, \frac{1}{L}]$, FEDAC satisfies (for all $t$)*

$$\mathbb{E}\left[\left\|\nabla F(\overline{w_t^{\mathrm{md}}}) - \frac{1}{M}\sum_{m=1}^{M}\nabla F(w_t^{\mathrm{md},m})\right\|^2\right] \leq \begin{cases} 4\eta^2 L^2 K\sigma^2\left(1 + \frac{\gamma^2\mu}{\eta}\right)^{2K} & \text{if } \gamma \in \left(\eta, \sqrt{\frac{\eta}{\mu}}\right], \\ 4\eta^2 L^2 K\sigma^2 & \text{if } \gamma = \eta. \end{cases}$$

The proof of Lemma C.14 is deferred to Section C.4.2.

Now plug in the choice of $\gamma = \max\left\{\sqrt{\frac{\eta}{\mu K}}, \eta\right\}$ to Lemmas C.2 and C.14, which leads to the following lemma.

**Lemma C.15** (Convergence of FEDAC-II for general $\eta$ under Assumption 1). *Let $F$ be $\mu > 0$-strongly convex, and assume Assumption 1, then for any $\eta \in (0, \frac{1}{L}]$, FEDAC-II yields*

$$\mathbb{E}[\Phi_T] \leq \exp\left(-\frac{1}{3}\max\left\{\eta\mu, \sqrt{\frac{\eta\mu}{K}}\right\}T\right)\Phi_0 + \frac{\eta^{\frac{1}{2}}\sigma^2}{\mu^{\frac{1}{2}}MK^{\frac{1}{2}}} + \frac{100\eta^2 L^2 K\sigma^2}{\mu}. \tag{C.33}$$

*Proof of Lemma C.15.* Applying Lemma C.2 yields

$$\mathbb{E}[\Phi_T] \leq \exp\left(-\frac{1}{3}\max\left\{\eta\mu, \sqrt{\frac{\eta\mu}{K}}\right\}T\right)\Phi_0 + \min\left\{\frac{3\eta L\sigma^2}{2\mu M}, \frac{3\eta^{\frac{3}{2}}LK^{\frac{1}{2}}\sigma^2}{2\mu^{\frac{1}{2}}M}\right\}$$

$$+ \max\left\{\frac{\eta\sigma^2}{2M}, \frac{\eta^{\frac{1}{2}}\sigma^2}{2\mu^{\frac{1}{2}}MK^{\frac{1}{2}}}\right\} + \frac{3}{\mu}\max_{0\leq t<T}\mathbb{E}\left[\left\|\nabla F(\overline{w_t^{\mathrm{md}}}) - \frac{1}{M}\sum_{m=1}^{M}\nabla F(w_t^{\mathrm{md},m})\right\|^2\right].$$

Applying Lemma C.14 yields (for all $t$)

$$\frac{3}{\mu}\mathbb{E}\left[\left\|\nabla F(\overline{w_t^{\mathrm{md}}}) - \frac{1}{M}\sum_{m=1}^{M}\nabla F(w_t^{\mathrm{md},m})\right\|^2\right] \leq \begin{cases} 12\mu^{-1}\eta^2 L^2 K\sigma^2\left(1 + \frac{1}{K}\right)^{2K} & \text{if } \gamma = \sqrt{\frac{\eta}{\mu K}}, \\ 12\mu^{-1}\eta^2 L^2 K\sigma^2 & \text{if } \gamma = \eta \end{cases}$$

$$\leq 12e^2\mu^{-1}\eta^2 L^2 K\sigma^2.$$

Note that

$$\min\left\{\frac{3\eta L\sigma^2}{2\mu M}, \frac{3\eta^{\frac{3}{2}}LK^{\frac{1}{2}}\sigma^2}{2\mu^{\frac{1}{2}}M}\right\} + \max\left\{\frac{\eta\sigma^2}{2M}, \frac{\eta^{\frac{1}{2}}\sigma^2}{2\mu^{\frac{1}{2}}MK^{\frac{1}{2}}}\right\}$$

$$\leq \frac{3\eta^{\frac{3}{2}}LK^{\frac{1}{2}}\sigma^2}{2\mu^{\frac{1}{2}}M} + \frac{\eta\sigma^2}{2M} + \frac{\eta^{\frac{1}{2}}\sigma^2}{2\mu^{\frac{1}{2}}MK^{\frac{1}{2}}}$$

$$\leq \frac{7\eta^{\frac{3}{2}}LK^{\frac{1}{2}}\sigma^2}{4\mu^{\frac{1}{2}}M} + \frac{3\eta^{\frac{1}{2}}\sigma^2}{4\mu^{\frac{1}{2}}MK^{\frac{1}{2}}}. \qquad \text{(by AM-GM inequality, and } \mu \leq L)$$

By Young's inequality,

$$\frac{7\eta^{\frac{3}{2}}LK^{\frac{1}{2}}\sigma^2}{4\mu^{\frac{1}{2}}M} \leq \left(\frac{3}{4}\frac{\eta^{\frac{1}{2}}\sigma^2}{\mu^{\frac{1}{2}}MK^{\frac{1}{2}}}\right)^{\frac{1}{3}}\left(3\cdot\frac{\eta^2 L^{\frac{3}{2}}K\sigma^2}{\mu^{\frac{1}{2}}M}\right)^{\frac{2}{3}} \qquad \text{(since } \frac{7}{4} \leq \left(\frac{3}{4}\right)^{\frac{1}{3}}(3)^{\frac{2}{3}})$$

$$\leq \frac{1}{4}\cdot\frac{\eta^{\frac{1}{2}}\sigma^2}{\mu^{\frac{1}{2}}MK^{\frac{1}{2}}} + 2\cdot\frac{\eta^2 L^{\frac{3}{2}}K\sigma^2}{\mu^{\frac{1}{2}}M} \qquad \text{(by Young's inequality)}$$

$$\leq \frac{\eta^{\frac{1}{2}}\sigma^2}{4\mu^{\frac{1}{2}}MK^{\frac{1}{2}}} + \frac{2\eta^2 L^2 K\sigma^2}{\mu}. \qquad \text{(since } L \geq \mu \text{ and } M \geq 1)$$

Combining the above inequalities gives

$$\mathbb{E}[\Phi_T] \leq \exp\left(-\frac{1}{3}\max\left\{\eta\mu, \sqrt{\frac{\eta\mu}{K}}\right\}T\right)\Phi_0 + \frac{\eta^{\frac{1}{2}}\sigma^2}{\mu^{\frac{1}{2}}MK^{\frac{1}{2}}} + \frac{(12\mathrm{e}^2+2)\eta^2L^2K\sigma^2}{\mu}.$$

The proof then follows by the estimate $12\mathrm{e}^2 + 2 < 100$. $\qquad\square$

Theorem C.13 then follows by plugging in the appropriate $\eta$ to Lemma C.15.

*Proof of Theorem C.13.* To simplify the notation, we denote the decreasing term in Eq. (C.33) in Lemma C.15 as $\varphi_\downarrow(\eta)$ and the increasing term as $\varphi_\uparrow(\eta)$, namely

$$\varphi_\downarrow(\eta) := \exp\left(-\frac{1}{3}\max\left\{\eta\mu, \sqrt{\frac{\eta\mu}{K}}\right\}T\right)\Phi_0, \quad \varphi_\uparrow(\eta) := \frac{\eta^{\frac{1}{2}}\sigma^2}{\mu^{\frac{1}{2}}MK^{\frac{1}{2}}} + \frac{100\eta^2L^2K\sigma^2}{\mu}.$$

Let

$$\eta_0 := \frac{9K}{\mu T^2}\log^2\left(\mathrm{e} + \min\left\{\frac{\mu MT\Phi_0}{\sigma^2} + \frac{\mu^3T^4\Phi_0}{L^2K^3\sigma^2}\right\}\right), \quad \text{then } \eta = \min\left\{\frac{1}{L}, \eta_0\right\}.$$

Therefore

$$\varphi_\downarrow(\eta) \leq \min\left\{\exp\left(-\frac{\mu T}{3L}\right), \exp\left(-\frac{\mu^{\frac{1}{2}}T}{3L^{\frac{1}{2}}K^{\frac{1}{2}}}\right)\right\}\Phi_0 + \frac{\sigma^2}{\mu MT} + \frac{L^2K^3\sigma^2}{\mu^3T^4}.$$

and

$$\varphi_\uparrow(\eta) \leq \varphi_\uparrow(\eta_0) \leq \frac{3\sigma^2}{\mu MT}\log\left(\mathrm{e} + \frac{\mu MT\Phi_0}{\sigma^2}\right) + \frac{8100L^2K^3\sigma^2}{\mu^3T^4}\log^4\left(\mathrm{e} + \frac{\mu^3T^4\Phi_0}{L^2K^3\sigma^2}\right).$$

Consequently,

$$\begin{aligned}
\mathbb{E}[\Phi_T] \leq &\varphi_\downarrow\left(\frac{1}{L}\right) + \varphi_\downarrow(\eta_0) + \varphi_\uparrow(\eta_0) \leq \min\left\{\exp\left(-\frac{\mu T}{3L}\right), \exp\left(-\frac{\mu^{\frac{1}{2}}T}{3L^{\frac{1}{2}}K^{\frac{1}{2}}}\right)\right\}\Phi_0 \\
&+ \frac{4\sigma^2}{\mu MT}\log\left(\mathrm{e} + \frac{\mu MT\Phi_0}{\sigma^2}\right) + \frac{8101L^2K^3\sigma^2}{\mu^3T^4}\log^4\left(\mathrm{e} + \frac{\mu^3T^4\Phi_0}{L^2K^3\sigma^2}\right).
\end{aligned}$$

$\qquad\square$

### C.4.2  Proof of Lemma C.14

We first introduce the supporting propositions for Lemma C.14. We omit most of the proof details since the analysis is largely shared.

The following proposition is parallel to Proposition C.9, where the difference is that the present proposition analyzes the 2$^\text{nd}$-order stability instead of 4$^\text{th}$-order.

**Proposition C.16.** *In the same setting of Lemma C.14, the following inequality holds (for all possible $t$)*

$$\mathbb{E}\left[\left\|\mathcal{X}(\gamma,\eta)^{-1}\begin{bmatrix}\Delta_{t+1}^{\mathrm{ag}}\\\Delta_{t+1}\end{bmatrix}\right\|^2\middle|\mathcal{F}_t\right] \leq 2\gamma^2\sigma^2 + \left\|\mathcal{X}(\gamma,\eta)^{-1}\begin{bmatrix}\Delta_t^{\mathrm{ag}}\\\Delta_t\end{bmatrix}\right\|^2 \cdot \begin{cases}\left(1+\frac{\gamma^2\mu}{\eta}\right)^2 & \text{if } \gamma\in\left(\eta, \sqrt{\frac{\eta}{\mu}}\right],\\1 & \text{if }\gamma=\eta,\end{cases}$$

*where $\mathcal{X}$ is the matrix-valued function defined in Eq. (C.26).*

*Proof of Proposition C.16.* Apply the uniform norm bound Proposition C.8, and the rest of the analysis is the same as Proposition B.10. $\qquad\square$

The following proposition is parallel to Proposition C.10, where the difference is that the present proposition uses $L$-(2$^\text{nd}$-order)-smoothness to bound the LHS quantity.

**Proposition C.17.** *In the same setting of Lemma C.14, the following inequality holds (for all possible t)*

$$\left\| \nabla F(\overline{w_t^{\text{md}}}) - \frac{1}{M} \sum_{m=1}^{M} \nabla F(w_t^{\text{md},m}) \right\|^2 \leq \frac{17\eta^2 L^2}{9\gamma^2} \left\| \mathcal{X}(\gamma,\eta)^{-1} \begin{bmatrix} \Delta_t^{\text{ag}} \\ \Delta_t \end{bmatrix} \right\|^2,$$

*where $\mathcal{X}$ is the matrix-valued function defined in Eq.* (C.26).

*Proof of Proposition C.17.* By $L$-smoothness (Assumption 1(b)),

$$\left\| \nabla F(\overline{w_t^{\text{md}}}) - \frac{1}{M} \sum_{m=1}^{M} \nabla F(w_t^{\text{md},m}) \right\|^2 \leq L^2 \|\Delta_t^{\text{md}}\|^2.$$

By definition of "md", sub-multiplicativity, and Claim C.12,

$$\|\Delta_t^{\text{md}}\|^2 = \left\| \mathcal{X}(\gamma,\eta)^{\mathsf{T}} \begin{bmatrix} \frac{9-9\gamma\mu+2\gamma^2\mu^2}{9-6\gamma\mu-\gamma^2\mu^2}I \\ \frac{3\gamma\mu-3\gamma^2\mu^2}{9-6\gamma\mu-\gamma^2\mu^2}I \end{bmatrix} \right\|^2 \left\| \mathcal{X}(\gamma,\eta)^{-1} \begin{bmatrix} \Delta_t^{\text{ag}} \\ \Delta_t \end{bmatrix} \right\|^2 \leq \frac{17\eta^2}{9\gamma^2} \left\| \mathcal{X}(\gamma,\eta)^{-1} \begin{bmatrix} \Delta_t^{\text{ag}} \\ \Delta_t \end{bmatrix} \right\|^2.$$

$\square$

Lemma C.14 then follows by telescoping Proposition C.16 and plugging in Proposition C.17.

*Proof of Lemma C.14.* Let $t_0$ be the latest synchronized step prior to $t$, then telescoping Proposition C.16 from $t_0$ to $t$ (note that $\Delta_{t_0} = \Delta_{t_0} = 0$)

$$\mathbb{E}\left[ \left\| \mathcal{X}(\gamma,\eta)^{-1} \begin{bmatrix} \Delta_t^{\text{ag}} \\ \Delta_t \end{bmatrix} \right\|^2 \bigg| \mathcal{F}_{t_0} \right] \leq 2\gamma^2\sigma^2 K \cdot \begin{cases} \left(1 + \frac{\gamma^2\mu}{\eta}\right)^{2K} & \text{if } \gamma \in \left(\eta, \sqrt{\frac{\eta}{\mu}}\right], \\ 1 & \text{if } \gamma = \eta. \end{cases}$$

Thus, by Proposition C.17,

$$\mathbb{E}\left[ \left\| \nabla F(\overline{w_t^{\text{md}}}) - \frac{1}{M} \sum_{m=1}^{M} \nabla F(w_t^{\text{md},m}) \right\|^2 \right] \leq \frac{34}{9}\eta^2\sigma^2 K \cdot \begin{cases} \left(1 + \frac{\gamma^2\mu}{\eta}\right)^{2K} & \text{if } \gamma \in \left(\eta, \sqrt{\frac{\eta}{\mu}}\right], \\ 1 & \text{if } \gamma = \eta. \end{cases}$$

The Lemma C.14 then follows by bounding $\frac{34}{9}$ with 4. $\square$

# D Analysis of FEDAVG under Assumption 2

In this section we study the convergence of FEDAVG under Assumption 2. We provide a complete, non-asymptotic version of Theorem 3.4 and provide the proof. We formally define FEDAVG in Algorithm 2 for reference.

Formally we use $\mathcal{F}_t$ to denote the $\sigma$-algebra generated by $\{w_\tau^m\}_{\tau \leq t, m \in [M]}$. Since FEDAVG is Markovian, conditioning on $\mathcal{F}_t$ is equivalent to conditioning on $\{w_t^m\}_{m \in [M]}$.

---

**Algorithm 2** Federated Averaging (a.k.a. Local SGD, Parallel SGD)

---

1: **procedure** FEDAVG($\eta$)
2:     Initialize $= w_0^m = w_0$ for all $m \in [M]$
3:     **for** $t = 0, \ldots, T-1$ **do**
4:         **for** every worker $m \in [M]$ **in parallel do**
5:             $g_t^m \leftarrow \nabla f(w_t^m; \xi_t^m)$                    ▷ Query gradient at $w_t^m$
6:             $v_{t+1}^m \leftarrow w_t^m - \eta \cdot g_t^m$            ▷ Compute next iterate candidate $v_{t+1}^m$
7:             **if** sync **then**
8:                 $w_{t+1}^m \leftarrow \frac{1}{M} \sum_{m=1}^{M} v_{t+1}^m$                    ▷ Average and broadcast
9:             **else**
10:                $w_{t+1}^m \leftarrow v_{t+1}^m$                    ▷ Candidates assigned to be the next iterates

---

## D.1 Main theorem and lemma: Complete version of Theorem 3.4

**Theorem D.1.** *Let $F$ be $\mu > 0$-strongly convex, and assume Assumption 2, then for*

$$\eta := \min\left\{ \frac{1}{4L}, \frac{2}{\mu T} \log\left( e + \min\left\{ \frac{\mu^2 M T^2 D_0^2}{\sigma^2}, \frac{\mu^6 T^5 D_0^2}{Q^2 K^2 \sigma^4} \right\} \right) \right\},$$

FEDAVG *yields*

$$\mathbb{E}\left[ F\left( \sum_{t=0}^{T-1} \frac{\rho_t}{S_T} \overline{w}_t \right) \right] - F^* + \frac{\mu}{2} \mathbb{E}[\|\overline{w}_T - w^*\|^2]$$

$$\leq \exp\left( -\frac{\mu T}{8L} \right) 4L D_0^2 + \frac{3\sigma^2}{\mu M T} \log\left( e + \frac{\mu^2 M T^2 D_0^2}{\sigma^2} \right) + \frac{3073 Q^2 K^2 \sigma^4}{\mu^5 T^4} \log^4\left( e + \frac{\mu^6 T^5 D_0^2}{Q^2 K^2 \sigma^4} \right).$$

*where $\rho_t := (1 - \frac{1}{2}\eta\mu)^{T-t-1}$, $S_T := \sum_{t=0}^{T-1} \rho_t$, and $D_0 = \|\overline{w}_0 - w^*\|$.*

The proof of Theorem D.1 is based on the following two lemmas regarding the convergence and 4th-order stability of FEDAVG. The averaging technique applied here is similar to [Stich, 2019b].

**Lemma D.2** (Perturbed iterate analysis for FEDAVG under Assumption 2)**.** *Let $F$ be $\mu > 0$-strongly convex, and assume Assumption 2, then for $\eta \in (0, \frac{1}{4L}]$, FEDAVG satisfies*

$$\mathbb{E}\left[ F\left( \sum_{t=0}^{T-1} \frac{\rho_t}{S_T} \overline{w}_t \right) \right] - F^* + \frac{\mu}{2} \mathbb{E}[\|\overline{w}_T - w^*\|^2]$$

$$\leq \frac{1}{\eta} \exp\left( -\frac{1}{2}\eta\mu T \right) D_0^2 + \frac{1}{M}\eta\sigma^2 + \frac{Q^2}{\mu} \left( \max_{0 \leq t < T} \frac{1}{M} \sum_{m=1}^{M} \mathbb{E}\left[ \|\overline{w}_t - w_t^m\|^4 \right] \right).$$

*where $\rho_t, S_T$ are defined in the statement of Theorem D.1.*

The proof of Lemma D.2 is deferred to Section D.2.

**Lemma D.3** (4th-order discrepancy overhead bound for FEDAVG)**.** *In the same settings of Lemma D.2, FEDAVG satisfies (for any $t$)*

$$\mathbb{E}\left[ \frac{1}{M} \sum_{m=1}^{M} \|\overline{w}_t - w_t^m\|^4 \right] \leq 192\eta^4 K^2 \sigma^4.$$

The proof of Lemma D.3 is deferred to Section D.3.

Combining Lemmas D.2 and D.3 gives

**Lemma D.4** (Convergence of FEDAVG under Assumption 2 for general $\eta$)**.** *In the same settings of Lemma D.2, FEDAVG yields*

$$\mathbb{E}\left[ F\left( \sum_{t=0}^{T-1} \frac{\rho_t}{S_T} \overline{w}_t \right) \right] - F^* + \frac{\mu}{2} \mathbb{E}[\|\overline{w}_T - w^*\|^2] \leq \frac{1}{\eta} \exp\left( -\frac{1}{2}\eta\mu T \right) D_0^2 + \frac{1}{M}\eta\sigma^2 + \frac{192\eta^4 Q^2 K^2 \sigma^4}{\mu}.$$
$$\text{(D.1)}$$

*Proof of Lemma D.4.* Immediate from Lemmas D.2 and D.3. □

Theorem D.1 then follows by plugging an appropriate $\eta$ to Lemma D.4.

*Proof of Theorem D.1.* To simplify the notation, denote the terms on the RHS of Eq. (D.1) as

$$\varphi_\downarrow(\eta) := \frac{1}{\eta} \exp\left( -\frac{1}{2}\eta\mu T \right) D_0^2, \qquad \varphi_\uparrow(\eta) := \frac{1}{M}\eta\sigma^2 + \frac{192\eta^4 Q^2 K^2 \sigma^4}{\mu}.$$

Let

$$\eta_0 := \frac{2}{\mu T} \log\left( e + \min\left\{ \frac{\mu^2 M T^2 D_0^2}{\sigma^2}, \frac{\mu^6 T^5 D_0^2}{Q^2 K^2 \sigma^4} \right\} \right), \qquad \text{then } \eta = \min\left\{ \frac{1}{4L}, \eta_0 \right\}.$$

Therefore $\varphi_\downarrow(\eta) \le \varphi_\downarrow(\frac{1}{4L}) + \varphi_\downarrow(\eta_0)$, where

$$\varphi_\downarrow\left(\frac{1}{4L}\right) = \exp\left(-\frac{\mu T}{8L}\right) 4LD_0^2, \tag{D.2}$$

and

$$\varphi_\downarrow(\eta_0) \le \frac{\mu T}{2} D_0^2 \cdot \left(\min\left\{\frac{\mu^2 MT^2 D_0^2}{\sigma^2}, \frac{\mu^6 T^5 D_0^2}{Q^2 K^2 \sigma^4}\right\}\right)^{-1} \le \frac{\sigma^2}{2\mu MT} + \frac{Q^2 K^2 \sigma^4}{2\mu^5 T^4}. \tag{D.3}$$

On the other hand

$$\varphi_\uparrow(\eta) \le \varphi_\uparrow(\eta_0) \le \frac{2\sigma^2}{\mu MT} \log\left(\mathrm{e} + \frac{\mu^2 MT^2 D_0^2}{\sigma^2}\right) + \frac{3072 Q^2 K^2 \sigma^4}{\mu^5 T^4} \log^4\left(\mathrm{e} + \frac{\mu^6 T^5 D_0^2}{Q^2 K^2 \sigma^4}\right). \tag{D.4}$$

Combining Lemma D.4 and Eqs. (D.2), (D.3) and (D.4) gives

$$\mathbb{E}\left[F\left(\sum_{t=0}^{T-1} \frac{\rho_t}{S_T}\overline{w_t}\right)\right] - F^* + \frac{\mu}{2}\mathbb{E}[\|\overline{w_T} - w^*\|^2]$$
$$\le \exp\left(-\frac{\mu T}{8L}\right) 4LD_0^2 + \frac{3\sigma^2}{\mu MT} \log\left(\mathrm{e} + \frac{\mu^2 MT^2 D_0^2}{\sigma^2}\right) + \frac{3073 Q^2 K^2 \sigma^4}{\mu^5 T^4} \log^4\left(\mathrm{e} + \frac{\mu^6 T^5 D_0^2}{Q^2 K^2 \sigma^4}\right).$$

$$\square$$

### D.2 Perturbed iterative analysis for FEDAVG: Proof of Lemma D.2

We first state and proof the following proposition on one-step analysis.

**Proposition D.5.** *Under the same assumption of Lemma D.2, for all t, the following inequality holds*

$$\mathbb{E}\left[\|\overline{w_{t+1}} - w^*\|^2|\mathcal{F}_t\right] \le \left(1 - \frac{1}{2}\eta\mu\right) \|\overline{w_t} - w^*\|^2 - \eta(F(\overline{w_t}) - F^*) + \frac{\eta Q^2}{\mu M}\sum_{m=1}^{M}\|\overline{w_t} - w_t^m\|^4 + \frac{\eta^2\sigma^2}{M}.$$

*Proof of Proposition D.5.* By definition of the FEDAVG procedure (see Algorithm 2), for all $m \in [M]$, $v_{t+1}^m = w_t^m - \eta\nabla f(w_t^m; \xi_t^m)$. Taking average over $m = 1, \dots, M$ gives

$$\overline{w_{t+1}} - w^* = w_t - \eta \cdot \frac{1}{M}\sum_{m=1}^{M}\nabla f(w_t^m; \xi_t^m) - w^*.$$

Taking conditional expectation, by bounded variance Assumption 1(c),

$$\mathbb{E}\left[\|\overline{w_{t+1}} - w^*\|^2|\mathcal{F}_t\right] = \left\|w_t - \eta \cdot \frac{1}{M}\sum_{m=1}^{M}\nabla F(w_t^m) - w^*\right\|^2 + \frac{1}{M}\eta^2\sigma^2. \tag{D.5}$$

Now we analyze the $\left\| w_t - \eta \cdot \frac{1}{M} \sum_{m=1}^{M} \nabla F(w_t^m) - w^* \right\|^2$ term as follows

$$
\left\| w_t - \eta \cdot \frac{1}{M} \sum_{m=1}^{M} \nabla F(w_t^m) - w^* \right\|^2
$$

$$
= \left\| \overline{w_t} - \eta \cdot \nabla F(\overline{w_t}) - w^* + \eta \left( \nabla F(\overline{w_t}) - \frac{1}{M} \sum_{m=1}^{M} \nabla F(w_t^m) \right) \right\|^2
$$

$$
\leq \left( 1 + \frac{1}{2}\eta\mu \right) \| \overline{w_t} - \eta \nabla F(\overline{w_t}) - w^* \|^2 + \eta^2 \left( 1 + \frac{2}{\eta\mu} \right) \left\| \nabla F(\overline{w_t}) - \frac{1}{M} \sum_{m=1}^{M} \nabla F(w_t^m) \right\|^2
$$
$$
\text{(apply Lemma G.2 with } \zeta = \tfrac{1}{2}\eta\mu)
$$

$$
\leq \left( 1 + \frac{1}{2}\eta\mu \right) \| \overline{w_t} - \eta \nabla F(\overline{w_t}) - w^* \|^2 + \eta^2 \left( 1 + \frac{2}{\eta\mu} \right) \frac{Q^2}{4M} \sum_{m=1}^{M} \| \overline{w_t} - w_t^m \|^4
$$
$$
\text{(by Lemma G.3)}
$$

$$
\leq \left( 1 + \frac{1}{2}\eta\mu \right) \| \overline{w_t} - \eta \nabla F(\overline{w_t}) - w^* \|^2 + \frac{\eta Q^2}{\mu M} \sum_{m=1}^{M} \| \overline{w_t} - w_t^m \|^4. \qquad (D.6)
$$

where the last inequality is due to $1 + \frac{2}{\eta\mu} \leq \frac{4}{\eta\mu}$ since $\eta\mu \leq \eta L \leq \frac{1}{4}$.

The first term of the RHS of Eq. (D.6) is bounded as

$$
\| \overline{w_t} - \eta \nabla F(\overline{w_t}) - w^* \|^2
$$
$$
= \| \overline{w_t} - w^* \|^2 - 2\eta \langle \nabla F(\overline{w_t}), \overline{w_t} - w^* \rangle + \eta^2 \| \nabla F(\overline{w_t}) \|^2 \qquad \text{(expansion of squared norm)}
$$
$$
\leq \| \overline{w_t} - w^* \|^2 - \eta \left( \mu \| \overline{w_t} - w^* \|^2 - 2(F(\overline{w_t}) - F^*) \right) + \eta^2 \cdot (2L(F(\overline{w_t}) - F^*))
$$
$$
\text{($\mu$-strongly convexity and $L$-smoothness by Assumption 1)}
$$
$$
= (1 - \eta\mu) \| \overline{w_t} - w^* \|^2 - 2\eta(1 - \eta L)(F(\overline{w_t}) - F^*)
$$
$$
\leq (1 - \eta\mu) \| \overline{w_t} - w^* \|^2 - \eta(F(\overline{w_t}) - F^*). \qquad \text{(since } \eta \leq \tfrac{1}{2L})
$$

Multiplying $(1 + \frac{1}{2}\eta\mu)$ on both sides gives (note that $(1 + \frac{1}{2}\eta\mu)(1 - \eta\mu) \leq (1 - \frac{1}{2}\eta\mu)$)

$$
\left( 1 + \frac{1}{2}\eta\mu \right) \| \overline{w_t} - \eta_t \nabla F(\overline{w_t}) - w^* \|^2
$$

$$
\leq \left( 1 + \frac{1}{2}\eta\mu \right) (1 - \eta\mu) \| \overline{w_t} - w^* \|^2 - \eta \left( 1 + \frac{1}{2}\eta\mu \right) (F(\overline{w_t}) - F^*)
$$

$$
\leq \left( 1 - \frac{1}{2}\eta\mu \right) \| \overline{w_t} - w^* \|^2 - \eta \left( F(\overline{w_t}) - F^* \right). \qquad (D.7)
$$

Combining Eqs. (D.5), (D.6) and (D.7) completes the proof of Proposition D.5. $\qquad \square$

With Proposition D.5 at hand we are ready to prove Lemma D.2. The telescoping techniques applied here are similar to [Stich, 2019b].

*Proof of Lemma D.2.* Telescoping Proposition D.5 yields

$$
\mathbb{E}\left[ \| \overline{w_T} - w^* \|^2 \right] + \eta \sum_{t=0}^{T-1} \left( 1 - \frac{1}{2}\eta\mu \right)^{T-t-1} (\mathbb{E}[F(\overline{w_t})] - F^*)
$$

$$
\leq \left( 1 - \frac{1}{2}\eta\mu \right)^T \| \overline{w_0} - w^* \|^2 + \sum_{t=0}^{T-1} \left( 1 - \frac{1}{2}\eta\mu \right)^{T-t-1} \left( \frac{1}{M}\eta^2\sigma^2 + \frac{\eta Q^2}{\mu M} \sum_{m=1}^{M} \mathbb{E}\left[ \| \overline{w_t} - w_t^m \|^4 \right] \right)
$$

$$
\leq \left( 1 - \frac{1}{2}\eta\mu \right)^T \| \overline{w_0} - w^* \|^2 + S_T \left( \frac{1}{M}\eta^2\sigma^2 + \frac{\eta Q^2}{\mu} \max_{0 \leq t < T} \frac{1}{M} \sum_{m=1}^{M} \mathbb{E}\left[ \| \overline{w_t} - w_t^m \|^4 \right] \right).
$$

Multiplying $\frac{1}{\eta S_T}$ on both sides and rearranging,

$$\sum_{t=0}^{T-1} \frac{\rho_t}{S_T}\left(\mathbb{E}[F(\overline{w_t})] - F^*\right) + \frac{1}{\eta S_T}\mathbb{E}[\|\overline{w_T} - w^*\|^2]$$

$$\leq \frac{(1-\frac{1}{2}\eta\mu)^T}{\eta S_T}\|\overline{w_0} - w^*\|^2 + \frac{1}{M}\eta\sigma^2 + \frac{Q^2}{\mu}\left(\max_{0\leq t<T}\frac{1}{M}\sum_{m=1}^M \mathbb{E}\left[\|\overline{w_t} - w_t^m\|^4\right]\right). \tag{D.8}$$

Note that $S_T := \sum_{t=0}^{T-1}\rho_t = \frac{1-(1-\frac{1}{2}\eta\mu)^T}{\frac{1}{2}\eta\mu}$, we have

$$\frac{1}{\eta S_T} = \frac{\mu}{2\left(1 - (1-\frac{1}{2}\eta\mu)^T\right)} \geq \frac{\mu}{2}, \tag{D.9}$$

and

$$\frac{(1-\frac{1}{2}\eta\mu)^T}{\eta S_T} = \frac{\mu(1-\frac{1}{2}\eta\mu)^T}{2\left(1 - (1-\frac{1}{2}\eta\mu)^T\right)} \leq \frac{\mu(1-\frac{1}{2}\eta\mu)^T}{\eta\mu} \leq \frac{1}{\eta}\exp\left(-\frac{1}{2}\eta\mu T\right). \tag{D.10}$$

Also by convexity

$$\sum_{t=0}^{T-1}\frac{\rho_t}{S_T}\left(\mathbb{E}[F(\overline{w_t})] - F^*\right) \geq \mathbb{E}\left[F\left(\sum_{t=0}^{T-1}\frac{\rho_t}{S_T}\overline{w_t}\right)\right] - F^*. \tag{D.11}$$

Plugging Eqs. (D.9), (D.10) and (D.11) to Eq. (D.8) gives

$$\mathbb{E}\left[F\left(\sum_{t=0}^{T-1}\frac{\rho_t}{S_T}\overline{w_t}\right)\right] - F^* + \frac{\mu}{2}\mathbb{E}[\|\overline{w_T} - w^*\|^2]$$

$$\leq \frac{1}{\eta}\exp\left(-\frac{1}{2}\eta\mu T\right)\|\overline{w_0} - w^*\|^2 + \frac{1}{M}\eta\sigma^2 + \frac{Q^2}{\mu}\left(\max_{0\leq t<T}\frac{1}{M}\sum_{m=1}^M \mathbb{E}\left[\|\overline{w_t} - w_t^m\|^4\right]\right).$$

$\square$

## D.3 Discrepancy overhead bound for FEDAVG: Proof of Lemma D.3

In this subsection we will prove Lemma D.3 regarding the 4th order stability of FEDAVG. We introduce a few more notations to simplify the discussions. Let $m_1, m_2 \in [M]$ be two arbitrary distinct workers. For any timestep $t$, let $\Delta_t := w_t^{m_1} - w_t^{m_2}$, and $\Delta_t^\varepsilon := \varepsilon_t^{m_1} - \varepsilon_t^{m_2}$ where $\varepsilon_t^m = \nabla f(w_t^m;\xi_t^m) - \nabla F(w_t^m)$ be the bias of the gradient oracle of the $m$-th worker evaluated at $w_t$. Let $\Delta_t^\nabla := \nabla F(w_t^{m_1}) - \nabla F(w_t^{m_2})$.

We first state and prove the following proposition on one-step 4th-order stability. The proof is analogous to the 4th-order convergence analysis of FEDAVG in [Dieuleveut and Patel, 2019].

**Proposition D.6.** *In the same setting of Lemma D.3, for all $t$,*

$$\sqrt{\mathbb{E}\|\Delta_{t+1}\|^4} \leq \sqrt{\mathbb{E}\|\Delta_t\|^4} + \sqrt{192}\eta^2\sigma^2.$$

*Proof of Proposition D.6.* If $t+1$ is a synchronized step, then the result follows trivially. We assume from now on that $t+1$ is not a synchronized step, then

$$\mathbb{E}[\|\Delta_{t+1}\|^4|\mathcal{F}_t] = \mathbb{E}\left[\|\Delta_t - \eta(\Delta_t^\nabla + \Delta_t^\varepsilon)\|^4|\mathcal{F}_t\right]$$

$$= \mathbb{E}\left[\left(\|\Delta_t\|^2 - 2\eta\langle\Delta_t, \Delta_t^\nabla + \Delta_t^\varepsilon\rangle + \eta^2\|\Delta_t^\nabla + \Delta_t^\varepsilon\|^2\right)^2\Big|\mathcal{F}_t\right]$$

$$= \mathbb{E}\|\Delta_t\|^4 - 4\eta\|\Delta_t\|^2\langle\Delta_t, \Delta_t^\nabla\rangle + 4\eta^2\,\mathbb{E}\left[\langle\Delta_t, \Delta_t^\nabla + \Delta_t^\varepsilon\rangle^2|\mathcal{F}_t\right] + 2\eta^2\|\Delta_t\|^2\,\mathbb{E}\left[\|\Delta_t^\nabla + \Delta_t^\varepsilon\|^2|\mathcal{F}_t\right]$$

$$\quad - 4\eta^3\,\mathbb{E}\left[\langle\Delta_t, \Delta_t^\nabla + \Delta_t^\varepsilon\rangle\cdot\|\Delta_t^\nabla + \Delta_t^\varepsilon\|^2|\mathcal{F}_t\right] + \eta^4\,\mathbb{E}\left[\|\Delta_t^\nabla + \Delta_t^\varepsilon\|^4|\mathcal{F}_t\right]$$

$$\leq \mathbb{E}\|\Delta_t\|^4 - 4\eta\|\Delta_t\|^2\langle\Delta_t, \Delta_t^\nabla\rangle + 6\eta^2\|\Delta_t\|^2\,\mathbb{E}\left[\|\Delta_t^\nabla + \Delta_t^\varepsilon\|^2|\mathcal{F}_t\right]$$

$$\quad + 4\eta^3\|\Delta_t\|\,\mathbb{E}\left[\|\Delta_t^\nabla + \Delta_t^\varepsilon\|^3|\mathcal{F}_t\right] + \eta^4\,\mathbb{E}\left[\|\Delta_t^\nabla + \Delta_t^\varepsilon\|^4|\mathcal{F}_t\right] \quad \text{(Cauchy-Schwarz inequality)}$$

$$\leq \mathbb{E}\|\Delta_t\|^4 - 4\eta\|\Delta_t\|^2\langle\Delta_t, \Delta_t^\nabla\rangle + 8\eta^2\|\Delta_t\|^2\,\mathbb{E}\left[\|\Delta_t^\nabla + \Delta_t^\varepsilon\|^2|\mathcal{F}_t\right] + 3\eta^4\,\mathbb{E}\left[\|\Delta_t^\nabla + \Delta_t^\varepsilon\|^4|\mathcal{F}_t\right],$$

$$\tag{D.12}$$

where the last inequality is due to

$$4\eta^3\|\Delta_t\| \mathbb{E}\left[\|\Delta_t^\nabla + \Delta_t^\varepsilon\|^3|\mathcal{F}_t\right] \leq 2\eta^2\|\Delta_t\|^2 \mathbb{E}\left[\|\Delta_t^\nabla + \Delta_t^\varepsilon\|^2|\mathcal{F}_t\right] + 2\eta^4 \mathbb{E}\left[\|\Delta_t^\nabla + \Delta_t^\varepsilon\|^4|\mathcal{F}_t\right]$$

by AM-GM inequality.

Note that by $L$-smoothness and convexity, we have the following inequality by standard convex analysis (*cf.*, Theorem 2.1.5 of [Nesterov, 2018]),

$$\|\Delta_t^\nabla\|^2 = \|\nabla F(w_t^{m_1}) - \nabla F(w_t^{m_2})\|^2 \leq L\left\langle w_t^{m_1} - w_t^{m_2}, \nabla F(w_t^{m_1}) - \nabla F(w_t^{m_2})\right\rangle = L\langle\Delta_t, \Delta_t^\nabla\rangle. \tag{D.13}$$

Consequently

$$\mathbb{E}\left[\|\Delta_t^\nabla + \Delta_t^\varepsilon\|^2|\mathcal{F}_t\right] = \|\Delta_t^\nabla\|^2 + \mathbb{E}\left[\|\Delta_t^\varepsilon\|^2|\mathcal{F}_t\right] \leq \|\Delta_t^\nabla\|^2 + 2\sigma^2 \leq L\langle\Delta_t, \Delta_t^\nabla\rangle + 2\sigma^2.$$

Similarly

$$\begin{aligned}
\mathbb{E}\left[\|\Delta_t^\nabla + \Delta_t^\varepsilon\|^4|\mathcal{F}_t\right] &\leq 8\|\Delta_t^\nabla\|^4 + 8\mathbb{E}\left[\|\Delta_t^\varepsilon\|^4|\mathcal{F}_t\right] &\text{(AM-GM inequality)}\\
&\leq 8\|\Delta_t^\nabla\|^4 + 64\sigma^4 &\text{(by Lemma G.4)}\\
&\leq 8L^2\|\Delta_t^2\|^2\|\Delta_t^\nabla\|^2 + 64\sigma^4 &\text{(by $L$-smoothness)}\\
&\leq 8L^3\|\Delta_t^2\|^2\langle\Delta_t, \Delta_t^\nabla\rangle + 64\sigma^4. &\text{(by Eq. (D.13))}
\end{aligned}$$

Plugging the above two bounds to Eq. (D.12) gives

$$\mathbb{E}[\|\Delta_{t+1}\|^4|\mathcal{F}_t] \leq \|\Delta_t\|^4 - 4\eta(1 - 2\eta L - 6\eta^3 L^3)\|\Delta_t\|^2\langle\Delta_t, \Delta_t^\nabla\rangle + 16\eta^2\|\Delta_t\|^2\sigma^2 + 192\eta^4\sigma^4. \tag{D.14}$$

Since $\eta L \leq \frac{1}{4}$ we have $(1 - 2\eta L - 6\eta^3 L^3) > 0$. By convexity $\langle\Delta_t, \Delta_t^\nabla\rangle \geq 0$. Hence the second term on the RHS of Eq. (D.14) is non-positive. We conclude that

$$\mathbb{E}[\|\Delta_{t+1}\|^4|\mathcal{F}_t] \leq \|\Delta_t\|^4 + 16\eta^2\sigma^2\|\Delta_t\|^2 + 192\eta^4\sigma^4.$$

Taking expectation gives

$$\begin{aligned}
\mathbb{E}[\|\Delta_{t+1}\|^4] &\leq \mathbb{E}[\|\Delta_t\|^4] + 16\eta^2\sigma^2 \mathbb{E}[\|\Delta_t\|^2] + 192\eta^4\sigma^4\\
&\leq \mathbb{E}[\|\Delta_t\|^4] + 16\eta^2\sigma^2\sqrt{\mathbb{E}[\|\Delta_t\|^4]} + 192\eta^4\sigma^4 = \left(\sqrt{\mathbb{E}\|\Delta_t\|^4} + \sqrt{192}\eta^2\sigma^2\right)^2.
\end{aligned}$$

Taking square root on both sides completes the proof. $\qquad\square$

With Proposition D.6 at hand we are ready to prove Lemma D.3.

*Proof of Lemma D.3.* Let $t_0$ be the latest synchronized prior to $t$, then telescoping Proposition D.6 yields (note that $\Delta_{t_0} = 0$)

$$\sqrt{\mathbb{E}\|\Delta_t\|^4} \leq \sqrt{192}\eta^2\sigma^2(t - t_0) \leq \sqrt{192}\eta^2 K\sigma^2,$$

where the last inequality is because $K$ is the synchronization gap. Thus

$$\frac{1}{M}\sum_{m=1}^M \mathbb{E}\left[\|\overline{w_t} - w_t^m\|^4\right] \leq \mathbb{E}[\|\Delta_t\|^4] \leq 192\eta^4 K^2\sigma^4,$$

where the first "$\leq$" is due to Jensen's inequality. $\qquad\square$

# E    Analysis of FEDAC for general convex objectives

## E.1    Main theorems

In this section we study the convergence of FEDAC for general convex ($\mu = 0$) objectives. Let $F$ be a general convex function, the main idea is to apply FEDAC to the $\ell_2$-augmented $\tilde{F}_\lambda(w)$ defined as

$$\tilde{F}_\lambda(w) := F(w) + \frac{1}{2}\lambda\|w - w_0\|^2, \tag{E.1}$$

where $w_0$ is the initial guess. Let $w_\lambda^*$ be the optimum of $\tilde{F}_\lambda(w)$ and define $\tilde{F}_\lambda^* := \tilde{F}_\lambda(w_\lambda^*)$.

One can verify that if $F$ satisfies Assumption 1 with general convexity ($\mu = 0$) and $L$-smoothness, then $\tilde{F}_\lambda$ satisfies Assumption 1 with smoothness $L + \lambda$ and strong-convexity $\lambda$ (variance does not change). If $F$ satisfies Assumption 2, then $\tilde{F}_\lambda$ also satisfies Assumption 2 with the same $Q$-3$^\text{rd}$-order-smoothness (4$^\text{th}$-order central moment does not change).

Now we state the convergence theorems. Note that the bounds in Table 2 can be obtained by replacing $K = T/R$. Recall $\|D_0 := \|w_0 - w^*\|$.

**Theorem E.1** (Convergence of FEDAC-I for general convex objective, under Assumption 1). *Assume Assumption 1 where $F$ is general convex. Then for any $T \geq 24$,[14] applying FEDAC-I to $\tilde{F}_\lambda$ (E.1) with*

$$\lambda = \max\left\{ \frac{\sigma}{M^{\frac{1}{2}}T^{\frac{1}{2}}D_0}, \frac{L^{\frac{1}{3}}K^{\frac{2}{3}}\sigma^{\frac{2}{3}}}{TD_0^{\frac{2}{3}}}, \frac{2LK}{T^2}\log^2\left(e^2 + \frac{T^2}{K}\right) \right\},$$

*and hyperparameter*

$$\eta = \min\left\{ \frac{1}{L+\lambda}, \frac{K}{\lambda T^2}\log^2\left(e + \min\left\{ \frac{\lambda LMTD_0^2}{\sigma^2}, \frac{\lambda^2 T^3 D_0^2}{K^2\sigma^2} \right\}\right), \frac{L^{\frac{1}{3}}K^{\frac{1}{3}}D_0^{\frac{2}{3}}}{\lambda^{\frac{2}{3}}T\sigma^{\frac{2}{3}}}, \frac{L^{\frac{1}{4}}K^{\frac{1}{4}}D_0^{\frac{1}{2}}}{\lambda^{\frac{3}{4}}T\sigma^{\frac{1}{2}}} \right\}$$

*yields*

$$\mathbb{E}\left[F(\overline{w_T^{\text{ag}}}) - F^*\right] \leq \frac{2LKD_0^2}{T^2}\log^2\left(e^2 + \frac{T^2}{K}\right) + \frac{2\sigma D_0}{M^{\frac{1}{2}}T^{\frac{1}{2}}}\log^2\left(e^2 + \frac{LM^{\frac{1}{2}}T^{\frac{1}{2}}D_0}{\sigma}\right)$$
$$+ \frac{1005 L^{\frac{1}{3}}K^{\frac{2}{3}}\sigma^{\frac{2}{3}}D_0^{\frac{4}{3}}}{T}\log^4\left(e^4 + \frac{L^{\frac{2}{3}}TD_0^{\frac{2}{3}}}{K^{\frac{2}{3}}\sigma^{\frac{2}{3}}}\right).$$

The proof of Theorem E.1 is deferred to Section E.2.

**Theorem E.2** (Convergence of FEDAC-II for general convex objective, under Assumption 1). *Assume Assumption 2 where $F$ is general convex. Then for any $T \geq 10^3$, applying FEDAC-II to $\tilde{F}_\lambda$ (E.1) with*

$$\lambda = \max\left\{ \frac{\sigma}{M^{\frac{1}{2}}T^{\frac{1}{2}}D_0}, \frac{L^{\frac{1}{2}}K^{\frac{3}{4}}\sigma^{\frac{1}{2}}}{TD_0^{\frac{1}{2}}}, \frac{18LK}{T^2}\log^2\left(e^2 + \frac{T^2}{K}\right) \right\},$$

*and hyperparameter*

$$\eta = \min\left\{ \frac{1}{L+\lambda}, \frac{9K}{\lambda T^2}\log^2\left(e + \min\left\{ \frac{\lambda LMTD_0^2}{\sigma^2}, \frac{\lambda^3 T^4 D_0^2}{LK^3\sigma^2} \right\}\right), \frac{L^{\frac{1}{3}}D_0^{\frac{2}{3}}}{\lambda^{\frac{2}{3}}T^{\frac{2}{3}}\sigma^{\frac{2}{3}}} \right\}$$

*yields*

$$\mathbb{E}\left[F(\overline{w_T^{\text{ag}}}) - F^*\right] \leq \frac{10LKD_0^2}{T^2}\log^2\left(e^2 + \frac{T^2}{K}\right) + \frac{5\sigma D_0}{M^{\frac{1}{2}}T^{\frac{1}{2}}}\log\left(e + \frac{LM^{\frac{1}{2}}T^{\frac{1}{2}}D_0}{\sigma}\right)$$
$$+ \frac{16411 L^{\frac{1}{2}}K^{\frac{3}{4}}\sigma^{\frac{1}{2}}D_0^{\frac{3}{2}}}{T}\log^4\left(e^4 + \frac{L^{\frac{1}{2}}TD_0^{\frac{1}{2}}}{K^{\frac{3}{4}}\sigma^{\frac{1}{2}}}\right).$$

The proof of Theorem E.2 is deferred to Section E.3.

**Theorem E.3** (Convergence of FEDAC-II for general convex objective, under Assumption 2). *Assume Assumption 2 where $F$ is general convex. Then for any $T \geq 10^3$, applying FEDAC-II to $\tilde{F}_\lambda$ (E.1) with*

$$\lambda = \max\left\{ \frac{\sigma}{M^{\frac{1}{2}}T^{\frac{1}{2}}D_0}, \frac{L^{\frac{1}{3}}K^{\frac{2}{3}}\sigma^{\frac{2}{3}}}{M^{\frac{1}{3}}TD_0^{\frac{2}{3}}}, \frac{Q^{\frac{1}{3}}K\sigma^{\frac{2}{3}}}{T^{\frac{4}{3}}D_0^{\frac{1}{3}}}, \frac{18LK}{T^2}\log^2\left(e^2 + \frac{T^2}{K}\right) \right\},$$

*and hyperparameter*

$$\eta = \min\left\{\frac{1}{L+\lambda}, \frac{9K}{\lambda T^2}\log^2\left(e + \min\left\{\frac{\lambda LMTD_0^2}{\sigma^2}, \frac{\lambda^2 MT^3 D_0^2}{K^2\sigma^2}, \frac{\lambda^5 LT^8 D_0^2}{Q^2 K^6 \sigma^4}\right\}\right), \frac{L^{\frac{1}{3}}K^{\frac{1}{3}}M^{\frac{1}{3}}D_0^{\frac{2}{3}}}{\lambda^{\frac{2}{3}}T\sigma^{\frac{2}{3}}}\right\}$$

*yields*

$$\mathbb{E}\left[F(\overline{w_T^{\mathrm{ag}}}) - F^*\right] \le \frac{10LKD_0^2}{T^2}\log^2\left(e^2 + \frac{T^2}{K}\right) + \frac{5\sigma D_0}{M^{\frac{1}{2}}T^{\frac{1}{2}}}\log\left(e + \frac{LM^{\frac{1}{2}}T^{\frac{1}{2}}D_0}{\sigma}\right)$$

$$+ \frac{139L^{\frac{1}{3}}K^{\frac{2}{3}}\sigma^{\frac{2}{3}}D_0^{\frac{4}{3}}}{M^{\frac{1}{3}}T}\log^3\left(e^3 + \frac{L^{\frac{2}{3}}M^{\frac{1}{3}}TD_0^{\frac{2}{3}}}{K^{\frac{2}{3}}\sigma^{\frac{2}{3}}}\right) + \frac{e^{19}Q^{\frac{1}{3}}K\sigma^{\frac{2}{3}}D_0^{\frac{5}{3}}}{T^{\frac{4}{3}}}\log^8\left(e^8 + \frac{LT^{\frac{4}{3}}D_0^{\frac{1}{3}}}{Q^{\frac{1}{3}}K\sigma^{\frac{2}{3}}}\right).$$

The proof of Theorem E.3 is deferred to Section E.4.

For comparison, we also establish the convergence of FEDAVG for general convex objective under Assumption 2.

**Theorem E.4** (Convergence of FEDAVG for general convex objective, under Assumption 2)**.** *Assume Assumption 2 where $F$ is general convex, then for any $T \ge 100$, applying FEDAVG to $\tilde{F}_\lambda$ (E.1) with*

$$\lambda := \max\left\{\frac{\sigma}{M^{\frac{1}{2}}T^{\frac{1}{2}}D_0}, \frac{Q^{\frac{1}{3}}K^{\frac{1}{3}}\sigma^{\frac{2}{3}}}{T^{\frac{2}{3}}D_0^{\frac{1}{3}}}, \frac{16L}{T}\log(e + T)\right\},$$

*and hyperparameter $\eta$*

$$\eta := \min\left\{\frac{1}{4(L+\lambda)}, \frac{2}{\lambda T}\log\left(e + \min\left\{\frac{\lambda^2 MT^2 D_0^2}{\sigma^2}, \frac{\lambda^6 T^5 D_0^2}{Q^2 K^2 \sigma^4}\right\}\right)\right\}$$

*yields*

$$\mathbb{E}\left[F\left(\sum_{t=0}^{T-1}\frac{\rho_t}{S_T}\overline{w_t}\right) - F^*\right] \le \frac{50LD_0^2}{T}\log(e+T) + \frac{6\sigma D_0}{M^{\frac{1}{2}}T^{\frac{1}{2}}}\log\left(e^2 + T\right) + \frac{3076Q^{\frac{1}{3}}K^{\frac{1}{3}}\sigma^{\frac{2}{3}}D_0^{\frac{5}{3}}}{T^{\frac{2}{3}}}\log^4\left(e^5 + T\right)$$

*where $\rho_t := (1 - \frac{1}{2}\eta\lambda)^{T-t-1}$, $S_T := \sum_{t=0}^{T-1}\rho_t$.*

The proof of Theorem E.4 is deferred to Section E.5.

### E.2 Proof of Theorem E.1 on FEDAC-I for general-convex objectives under Assumption 1

We first introduce the supporting lemmas for Theorem E.1.

**Lemma E.5.** *Assume Assumption 1 where $F$ is general convex, then for any $\lambda > 0$, for any $\eta \le \frac{1}{L+\lambda}$, applying FEDAC-I to $\tilde{F}_\lambda$ gives*

$$\mathbb{E}\left[F(\overline{w_T^{\mathrm{ag}}}) - F^*\right] \le \frac{1}{2}\lambda D_0^2 + \frac{1}{2}LD_0^2 \exp\left(-\sqrt{\frac{\eta\lambda}{K}}T\right) + \frac{\eta^{\frac{1}{2}}\sigma^2}{2\lambda^{\frac{1}{2}}MK^{\frac{1}{2}}} + \frac{\eta\sigma^2}{2M}$$

$$+ \frac{390\eta^{\frac{3}{2}}LK^{\frac{1}{2}}\sigma^2}{\lambda^{\frac{1}{2}}} + 7\eta^2 LK\sigma^2 + 390\eta^{\frac{3}{2}}\lambda^{\frac{1}{2}}K^{\frac{1}{2}}\sigma^2 + 7\eta^2\lambda K\sigma^2. \quad \text{(E.2)}$$

The proof of Lemma E.5 is deferred to Section E.2.1. Now we plug in $\eta$.

**Lemma E.6.** *Assume Assumption 1 where $F$ is general convex, then for any $\lambda > 0$, for*

$$\eta = \min\left\{\frac{1}{L+\lambda}, \frac{K}{\lambda T^2}\log^2\left(e + \min\left\{\frac{\lambda LMTD_0^2}{\sigma^2}, \frac{\lambda^2 T^3 D_0^2}{K^2\sigma^2}\right\}\right), \frac{L^{\frac{1}{3}}K^{\frac{1}{3}}D_0^{\frac{2}{3}}}{\lambda^{\frac{2}{3}}T\sigma^{\frac{2}{3}}}, \frac{L^{\frac{1}{4}}K^{\frac{1}{4}}D_0^{\frac{1}{2}}}{\lambda^{\frac{3}{4}}T\sigma^{\frac{1}{2}}}\right\},$$

*applying* FEDAC-I *to* $\tilde{F}_\lambda$ *gives*

$$\mathbb{E}\left[F(\overline{w_T^{\mathrm{ag}}}) - F^*\right] \leq \frac{1}{2}\lambda D_0^2 + \frac{3\sigma^2}{2\lambda MT} \log^2\left(\mathrm{e}^2 + \frac{\lambda LMTD_0^2}{\sigma^2}\right)$$

$$+ \frac{592LK^2\sigma^2}{\lambda^2 T^3} \log^4\left(\mathrm{e}^4 + \frac{\lambda^2 T^3 D_0^2}{K^2\sigma^2}\right)$$

$$+ \frac{412L^{\frac{1}{2}}K\sigma D_0}{\lambda^{\frac{1}{2}}T^{\frac{3}{2}}} + \frac{1}{2}LD_0^2 \exp\left(-\sqrt{\frac{1}{(1 + L/\lambda)K}}T\right). \qquad \text{(E.3)}$$

*Proof of Lemma E.6.* To simplify the notation, we name the terms of RHS of Eq. (E.2) as

$$\varphi_0(\eta) := \frac{1}{2}LD_0^2 \exp\left(-\sqrt{\frac{\eta\lambda}{K}}T\right),$$

$$\varphi_1(\eta) := \frac{\eta^{\frac{1}{2}}\sigma^2}{2\lambda^{\frac{1}{2}}MK^{\frac{1}{2}}}, \qquad\qquad \varphi_2(\eta) := \frac{\eta\sigma^2}{2M},$$

$$\varphi_3(\eta) := \frac{390\eta^{\frac{3}{2}}LK^{\frac{1}{2}}\sigma^2}{\lambda^{\frac{1}{2}}}, \qquad\qquad \varphi_4(\eta) := 7\eta^2 LK\sigma^2,$$

$$\varphi_5(\eta) := 390\eta^{\frac{3}{2}}\lambda^{\frac{1}{2}}K^{\frac{1}{2}}\sigma^2, \qquad\qquad \varphi_6(\eta) := 7\eta^2\lambda K\sigma^2.$$

Define

$$\eta_1 := \frac{K}{\lambda T^2} \log^2\left(\mathrm{e}^2 + \min\left\{\frac{\lambda LMTD_0^2}{\sigma^2}, \frac{\lambda^2 T^3 D_0^2}{K^2\sigma^2}\right\}\right), \quad \eta_2 := \frac{L^{\frac{1}{3}}K^{\frac{1}{3}}D_0^{\frac{2}{3}}}{\lambda^{\frac{2}{3}}T\sigma^{\frac{2}{3}}}, \quad \eta_3 := \frac{L^{\frac{1}{4}}K^{\frac{1}{4}}D_0^{\frac{1}{2}}}{\lambda^{\frac{3}{4}}T\sigma^{\frac{1}{2}}}.$$

then $\eta = \min\left\{\eta_1, \eta_2, \eta_3, \frac{1}{L+\lambda}\right\}$. Now we bound $\varphi_1(\eta), \ldots, \varphi_6(\eta)$ term by term.

$$\varphi_1(\eta) \leq \varphi_1(\eta_1) \leq \frac{\sigma^2}{2\lambda MT} \log\left(\mathrm{e} + \frac{\lambda LMTD_0^2}{\sigma^2}\right),$$

$$\varphi_2(\eta) \leq \varphi_2(\eta_1) \leq \frac{K\sigma^2}{2\lambda MT^2} \log^2\left(\mathrm{e} + \frac{\lambda LMTD_0^2}{\sigma^2}\right) \leq \frac{\sigma^2}{2\lambda MT} \log^2\left(\mathrm{e} + \frac{\lambda LMTD_0^2}{\sigma^2}\right),$$
$$\text{(since } K \leq T)$$

$$\varphi_3(\eta) \leq \varphi_3(\eta_1) \leq \frac{390LK^2\sigma^2}{\lambda^2 T^3} \log^3\left(\mathrm{e} + \frac{\lambda^2 T^3 D_0^2}{K^2\sigma^2}\right),$$

$$\varphi_4(\eta) \leq \varphi_4(\eta_1) \leq \frac{7LK^3\sigma^2}{\lambda^2 T^4} \log^4\left(\mathrm{e} + \frac{\lambda^2 T^3 D_0^2}{K^2\sigma^2}\right) \leq \frac{7LK^2\sigma^2}{\lambda^2 T^3} \log^4\left(\mathrm{e} + \frac{\lambda^2 T^3 D_0^2}{K^2\sigma^2}\right),$$
$$\text{(since } K \leq T)$$

$$\varphi_5(\eta) \leq \varphi_5(\eta_2) = \frac{390L^{\frac{1}{2}}KD_0\sigma}{\lambda^{\frac{1}{2}}T^{\frac{3}{2}}},$$

$$\varphi_6(\eta) \leq \varphi_6(\eta_3) \leq 7\eta_3^2\lambda K\sigma^2 = \frac{7L^{\frac{1}{2}}K^{\frac{3}{2}}D_0\sigma}{\lambda^{\frac{1}{2}}T^2} \leq \frac{7L^{\frac{1}{2}}KD_0\sigma}{\lambda^{\frac{1}{2}}T^{\frac{3}{2}}}. \qquad \text{(since } K \leq T)$$

In summary

$$\sum_{i=1}^{6}\varphi_i(\eta) \leq \frac{\sigma^2}{\lambda MT} \log^2\left(\mathrm{e}^2 + \frac{\lambda LMTD_0^2}{\sigma^2}\right) + \frac{397LK^2\sigma^2}{\lambda^2 T^3} \log^4\left(\mathrm{e}^4 + \frac{\lambda^2 T^3 D_0^2}{K^2\sigma^2}\right) + \frac{397L^{\frac{1}{2}}KD_0\sigma}{\lambda^{\frac{1}{2}}T^{\frac{3}{2}}}.$$
$$\text{(E.4)}$$

On the other hand $\varphi_0(\eta) \leq \varphi_0(\eta_1) + \varphi_0(\eta_2) + \varphi_0(\eta_3) + \varphi_0(\frac{1}{L+\lambda})$, where

$$\varphi_0(\eta_1) = \frac{1}{2}LD_0^2 \left(\mathrm{e}^2 + \min\left\{\frac{\lambda LMTD_0^2}{\sigma^2}, \frac{\lambda^2 T^3 D_0^2}{K^2\sigma^2}\right\}\right)^{-1} \leq \frac{\sigma^2}{2\lambda MT} + \frac{195LK^2\sigma^2}{\lambda^2 T^3},$$

$$\varphi_0(\eta_2) \leq \frac{3!}{2}LD_0^2 \left(\sqrt{\frac{\eta_2\lambda}{K}}T\right)^{-3} = \frac{3LK^{\frac{3}{2}}D_0^2}{\eta_2^{\frac{3}{2}}\lambda^{\frac{3}{2}}T^3} = \frac{3L^{\frac{1}{2}}KD_0\sigma}{\lambda^{\frac{1}{2}}T^{\frac{3}{2}}},$$

$$\varphi_0(\eta_3) \leq \frac{4!}{2}LD_0^2 \left(\sqrt{\frac{\eta_3\lambda}{K}}T\right)^{-4} = \frac{12LK^2 D_0^2}{\eta_3^2\lambda^2 T^4} = \frac{12L^{\frac{1}{2}}K^{\frac{3}{2}}\sigma D_0}{\lambda^{\frac{1}{2}}T^2} \leq \frac{12L^{\frac{1}{2}}KD_0\sigma}{\lambda^{\frac{1}{2}}T^{\frac{3}{2}}}.$$

In summary

$$\varphi_0(\eta) \leq \frac{1}{2}LD_0^2 \exp\left(-\sqrt{\frac{\lambda}{(L+\lambda)K}}T\right) + \frac{\sigma^2}{2\lambda MT} + \frac{195LK^2\sigma^2}{\lambda^2 T^3} + \frac{15L^{\frac{1}{2}}KD_0\sigma}{\lambda^{\frac{1}{2}}T^{\frac{3}{2}}}. \tag{E.5}$$

Combining Lemma E.5 and Eqs. (E.4) and (E.5) gives

$$\mathbb{E}\left[F(\overline{w_T^{\mathrm{ag}}}) - F^*\right] \leq \sum_{i=0}^{6}\varphi_i(\eta) + \frac{1}{2}\lambda D_0^2$$

$$\leq \frac{1}{2}\lambda D_0^2 + \frac{3\sigma^2}{2\lambda MT}\log^2\left(\mathrm{e}^2 + \frac{\lambda LMTD_0^2}{\sigma^2}\right) + \frac{592LK^2\sigma^2}{\lambda^2 T^3}\log^4\left(\mathrm{e}^4 + \frac{\lambda^2 T^3 D_0^2}{K^2\sigma^2}\right)$$

$$+ \frac{412L^{\frac{1}{2}}K\sigma D_0}{\lambda^{\frac{1}{2}}T^{\frac{3}{2}}} + \frac{1}{2}LD_0^2 \exp\left(-\sqrt{\frac{1}{(1+L/\lambda)K}}T\right).$$

$\square$

The main Theorem E.1 then follows by plugging in the appropriate $\eta$.

*Proof of Theorem E.1.* To simplify the notation, we name the terms on the RHS of Eq. (E.3) as

$$\psi_0(\lambda) := \frac{1}{2}\lambda D_0^2, \qquad\qquad \psi_1(\lambda) := \frac{3\sigma^2}{2\lambda MT}\log^2\left(\mathrm{e}^2 + \frac{\lambda LMTD_0^2}{\sigma^2}\right),$$

$$\psi_2(\lambda) := \frac{592LK^2\sigma^2}{\lambda^2 T^3}\log^4\left(\mathrm{e}^4 + \frac{\lambda^2 T^3 D_0^2}{K^2\sigma^2}\right), \quad \psi_3(\lambda) := \frac{412L^{\frac{1}{2}}KD_0\sigma}{\lambda^{\frac{1}{2}}T^{\frac{3}{2}}},$$

$$\psi_4(\lambda) := \frac{1}{2}LD_0^2 \exp\left(-\sqrt{\frac{1}{(1+L/\lambda)K}}T\right).$$

Let

$$\lambda_1 := \frac{\sigma}{M^{\frac{1}{2}}T^{\frac{1}{2}}D_0}, \quad \lambda_2 := \frac{L^{\frac{1}{3}}K^{\frac{2}{3}}\sigma^{\frac{2}{3}}}{TD_0^{\frac{2}{3}}}, \quad \lambda_3 := \frac{2KL}{T^2}\log^2\left(\mathrm{e}^2 + \frac{T^2}{K}\right),$$

then $\lambda := \max\{\lambda_1, \lambda_2, \lambda_3\}$. By helper Lemma G.5, $\psi_1$ and $\psi_2$ are monotonically decreasing w.r.t $\lambda$ for $\lambda > 0$. $\psi_3$ is trivially decreasing. Thus

$$\psi_1(\lambda) \leq \psi_1(\lambda_1) \leq \frac{3\sigma D_0}{2M^{\frac{1}{2}}T^{\frac{1}{2}}}\log^2\left(\mathrm{e}^2 + \frac{LM^{\frac{1}{2}}T^{\frac{1}{2}}D_0}{\sigma}\right), \tag{E.6}$$

$$\psi_2(\lambda) \leq \psi_2(\lambda_2) \leq \frac{592L^{\frac{1}{3}}K^{\frac{2}{3}}\sigma^{\frac{2}{3}}D_0^{\frac{4}{3}}}{T}\log^4\left(\mathrm{e}^4 + \frac{L^{\frac{2}{3}}TD_0^{\frac{2}{3}}}{K^{\frac{2}{3}}\sigma^{\frac{2}{3}}}\right), \tag{E.7}$$

$$\psi_3(\lambda) \leq \psi_3(\lambda_2) = \frac{412L^{\frac{1}{3}}K^{\frac{2}{3}}\sigma^{\frac{2}{3}}D_0^{\frac{4}{3}}}{T}. \tag{E.8}$$

Now we analyze $\psi_4(\lambda_3)$. Note first that $\frac{\lambda_3}{L} = \frac{2K}{T^2}\log^2\left(e^2 + \frac{T^2}{K}\right)$. Since $T \geq 24$ we have $\frac{T^2}{K} \geq 24$. By helper Lemma G.5, $x^{-1}\log^2(e^2 + x)$ is monotonically decreasing over $(0, +\infty)$, thus

$$\frac{\lambda_3}{L} = \frac{2K}{T^2}\log^2\left(e^2 + \frac{T^2}{K}\right) \leq \frac{1}{12}\log^2(e^2 + 24) < 1.$$

Hence

$$1 + \frac{L}{\lambda_3} \leq \frac{2L}{\lambda_3} = \frac{T^2}{K}\log^{-2}\left(e^2 + \frac{T^2}{K}\right).$$

We conclude that

$$\psi_4(\lambda) \leq \psi_4(\lambda_3) = \frac{1}{2}LD_0^2\exp\left(-\sqrt{\frac{1}{(1+L/\lambda_3)K}}T\right) \leq \frac{1}{2}LD_0^2\left(e^2 + \frac{T^2}{K}\right)^{-1} \leq \frac{LKD_0^2}{2T^2}.$$
$$\tag{E.9}$$

Finally note that

$$\psi_0(\lambda) \leq \frac{1}{2}\lambda_1 D_0^2 + \frac{1}{2}\lambda_2 D_0^2 + \frac{1}{2}\lambda_3 D_0^2 = \frac{\sigma D_0}{2M^{\frac{1}{2}}T^{\frac{1}{2}}} + \frac{L^{\frac{1}{3}}K^{\frac{2}{3}}\sigma^{\frac{2}{3}}D_0^{\frac{4}{3}}}{2T} + \frac{LKD_0^2}{T^2}\log^2\left(e^2 + \frac{T^2}{K}\right).$$
$$\tag{E.10}$$

Combining Lemma E.6 and Eqs. (E.6), (E.7), (E.8), (E.9) and (E.10) gives

$$\mathbb{E}\left[F(\overline{w_T^{\mathrm{ag}}}) - F^*\right] \leq \sum_{i=0}^{4}\psi_i(\lambda)$$

$$\leq \frac{2LKD_0^2}{T^2}\log^2\left(e^2 + \frac{T^2}{K}\right) + \frac{2\sigma D_0}{M^{\frac{1}{2}}T^{\frac{1}{2}}}\log^2\left(e^2 + \frac{LM^{\frac{1}{2}}T^{\frac{1}{2}}D_0}{\sigma}\right)$$

$$+ \frac{1005L^{\frac{1}{3}}K^{\frac{2}{3}}\sigma^{\frac{2}{3}}D_0^{\frac{4}{3}}}{T}\log^4\left(e^4 + \frac{L^{\frac{2}{3}}TD_0^{\frac{2}{3}}}{K^{\frac{2}{3}}\sigma^{\frac{2}{3}}}\right).$$

$$\square$$

### E.2.1 Proof of Lemma E.5

We first introduce a supporting proposition for Lemma E.5.

**Proposition E.7.** *Assume $F$ is general convex and $L$-smooth, and let $\Psi_t$ be the decentralized potential Eq. (B.1) for $\tilde{F}_\lambda$, namely*

$$\Psi_t := \frac{1}{M}\sum_{m=1}^{M}\left(\tilde{F}_\lambda(w_t^{\mathrm{ag},m}) - \tilde{F}_\lambda^*\right) + \frac{1}{2}\lambda\|\overline{w_T} - w_\lambda^*\|^2.$$

*Then*

$$\Psi_T \geq F(\overline{w_T^{\mathrm{ag}}}) - F^* - \frac{1}{2}\lambda D_0^2, \qquad \Psi_0 \leq \frac{1}{2}L\|w_0 - w^*\|^2.$$

*Proof of Proposition E.7.* Since $w_\lambda^*$ optimizes $\tilde{F}_\lambda(w)$ we have $\tilde{F}_\lambda(w_\lambda^*) \leq \tilde{F}_\lambda(w^*)$ (recall $w^*$ is defined as the optimum of the un-augmented objective $F$), and thus

$$\tilde{F}_\lambda^* = F(w_\lambda^*) + \frac{1}{2}\lambda\|w_\lambda^* - w_0\|^2 \leq F(w^*) + \frac{1}{2}\lambda\|w^* - w_0\|^2.$$
$$\tag{E.11}$$

Consequently, $\Psi_T$ is lower bounded as

$$
\begin{aligned}
\Psi_T &= \frac{1}{M} \sum_{m=1}^{M} \left( \tilde{F}_\lambda(w_T^{\mathrm{ag},m}) - \tilde{F}_\lambda^* \right) + \frac{1}{2}\lambda \|\overline{w_T} - w_\lambda^*\|^2 \geq \frac{1}{M} \sum_{m=1}^{M} \left( \tilde{F}_\lambda(w_T^{\mathrm{ag},m}) - \tilde{F}_\lambda^* \right) \\
&= \frac{1}{M} \sum_{m=1}^{M} \left[ \left( F(w_T^{\mathrm{ag},m}) + \frac{1}{2}\lambda \|w_T^{\mathrm{ag},m} - w_0\|^2 \right) - \tilde{F}_\lambda^* \right] \\
&\geq \frac{1}{M} \sum_{m=1}^{M} \left[ F(w_T^{\mathrm{ag},m}) - F^* + \frac{1}{2}\lambda \left( \|w_T^{\mathrm{ag},m} - w_0\|^2 - \|w^* - w_0\|^2 \right) \right] \quad \text{(by Eq. (E.11))} \\
&\geq \frac{1}{M} \sum_{m=1}^{M} \left( F(w_T^{\mathrm{ag},m}) - F^* \right) - \frac{1}{2}\lambda \|w^* - w_0\|^2 \\
&\geq F(\overline{w_T^{\mathrm{ag}}}) - F^* - \frac{1}{2}\lambda \|w^* - w_0\|^2 \quad \text{(by convexity)} \\
&= F(\overline{w_T^{\mathrm{ag}}}) - F^* - \frac{1}{2}\lambda D_0^2.
\end{aligned}
$$

The initial potential $\Psi_0$ is upper bounded as

$$
\begin{aligned}
\Psi_0 &= \tilde{F}_\lambda(w_0) - \tilde{F}_\lambda^* + \frac{1}{2}\lambda \|w_\lambda^* - w_0\|^2 \\
&= F(w_0) - \left( F(w_\lambda^*) + \frac{1}{2}\lambda \|w_\lambda^* - w_0\|^2 \right) + \frac{1}{2}\lambda \|w_\lambda^* - w_0\|^2 \quad \text{(by definition of } \tilde{F}_\lambda \text{ (E.1))} \\
&= F(w_0) - F(w_\lambda^*) \leq F(w_0) - F^* \quad \text{(by optimality } F(w_\lambda^*) \geq F^*) \\
&\leq \frac{1}{2}L\|w_0 - w^*\|^2 = \frac{1}{2}LD_0^2. \quad \text{(by } L\text{-smoothness of } F)
\end{aligned}
$$

$\square$

Lemma E.5 then follows by applying Lemma B.4 and Proposition E.7.

*Proof of Lemma E.5.* By Lemma B.4 on the convergence of FEDAC-I, for any $\eta \in (0, \frac{1}{L+\lambda})$,

$$
\mathbb{E}\left[\Psi_T\right] \leq \exp\left(-\sqrt{\frac{\eta\lambda}{K}}T\right)\Psi_0 + \frac{\eta^{\frac{1}{2}}\sigma^2}{2\lambda^{\frac{1}{2}}MK^{\frac{1}{2}}} + \frac{\eta\sigma^2}{2M} + \frac{390\eta^{\frac{3}{2}}(L+\lambda)K^{\frac{1}{2}}\sigma^2}{\lambda^{\frac{1}{2}}} + 7\eta^2(L+\lambda)K\sigma^2.
$$

Applying Proposition E.7 gives

$$
\begin{aligned}
\mathbb{E}\left[F(\overline{w_T^{\mathrm{ag}}}) - F^*\right] \leq &\frac{1}{2}LD_0^2 \exp\left(-\sqrt{\frac{\eta\lambda}{K}}T\right) + \frac{1}{2}\lambda D_0^2 + \frac{\eta^{\frac{1}{2}}\sigma^2}{2\lambda^{\frac{1}{2}}MK^{\frac{1}{2}}} + \frac{\eta\sigma^2}{2M} \\
&+ \frac{390\eta^{\frac{3}{2}}LK^{\frac{1}{2}}\sigma^2}{\lambda^{\frac{1}{2}}} + 7\eta^2 LK\sigma^2 + 390\eta^{\frac{3}{2}}\lambda^{\frac{1}{2}}K^{\frac{1}{2}}\sigma^2 + 7\eta^2\lambda K\sigma^2.
\end{aligned}
$$

$\square$

### E.3 Proof of Theorem E.2 on FEDAC-II for general-convex objectives under Assumption 1

We omit some technical details since the proof is similar to Theorem E.1. We first introduce the supporting lemma for Theorem E.2.

**Lemma E.8.** *Assume Assumption 1 where $F$ is general convex, then for any $\lambda > 0$, for any $\eta \leq \frac{1}{L+\lambda}$, applying FEDAC-II to $\tilde{F}_\lambda$ gives*

$$
\mathbb{E}\left[F(\overline{w_T^{\mathrm{ag}}}) - F^*\right] \leq \frac{1}{2}\lambda D_0^2 + \frac{1}{2}LD_0^2 \exp\left(-\sqrt{\frac{\eta\lambda T^2}{9K}}\right) + \frac{\eta^{\frac{1}{2}}\sigma^2}{\lambda^{\frac{1}{2}}MK^{\frac{1}{2}}} + \frac{200\eta^2 L^2 K\sigma^2}{\lambda} + 200\eta^2\lambda K\sigma^2.
$$

(E.12)

The proof of Lemma E.8 is deferred to Section E.3.1.

**Lemma E.9.** *Assume Assumption 1 where $F$ is general convex, then for any $\lambda > 0$, for*

$$\eta = \min \left\{ \frac{1}{L+\lambda}, \frac{9K}{\lambda T^2} \log^2 \left( e + \min \left\{ \frac{\lambda L M T D_0^2}{\sigma^2}, \frac{\lambda^3 T^4 D_0^2}{L K^3 \sigma^2} \right\} \right), \frac{L^{\frac{1}{3}} D_0^{\frac{2}{3}}}{\lambda^{\frac{2}{3}} T^{\frac{2}{3}} \sigma^{\frac{2}{3}}}, \right\}$$

*applying* FEDAC-II *to $\tilde{F}_\lambda$ gives*

$$\mathbb{E} \left[ F(\overline{w_T^{\text{ag}}}) - F^* \right] \le \frac{1}{2} \lambda D_0^2 + \frac{1}{2} L D_0^2 \exp \left( -\sqrt{\frac{T^2}{9(1+L/\lambda)K}} \right) + \frac{209 L^{\frac{2}{3}} K D_0^{\frac{4}{3}} \sigma^{\frac{2}{3}}}{\lambda^{\frac{1}{3}} T^{\frac{4}{3}}}$$

$$+ \frac{4\sigma^2}{\lambda M T} \log \left( e + \frac{\lambda L M T D_0^2}{\sigma^2} \right) + \frac{16201 L^2 K^3 \sigma^2}{\lambda^3 T^4} \log^4 \left( e^4 + \frac{\lambda^3 T^4 D_0^2}{L K^3 \sigma^2} \right). \tag{E.13}$$

*Proof of Lemma E.9.* To simplify the notation, define the terms on the RHS of Eq. (E.12) as

$$\varphi_0(\eta) := \frac{1}{2} L D_0^2 \exp \left( -\sqrt{\frac{\eta \lambda T^2}{9K}} \right), \quad \varphi_1(\eta) := \frac{\eta^{\frac{1}{2}} \sigma^2}{\lambda^{\frac{1}{2}} M K^{\frac{1}{2}}},$$

$$\varphi_2(\eta) := \frac{200 \eta^2 L^2 K \sigma^2}{\lambda}, \qquad \varphi_3(\eta) := 200 \eta^2 \lambda K \sigma^2.$$

Define

$$\eta_1 := \frac{9K}{\lambda T^2} \log^2 \left( e + \min \left\{ \frac{\lambda L M T D_0^2}{\sigma^2}, \frac{\lambda^3 T^4 D_0^2}{L K^3 \sigma^2} \right\} \right), \qquad \eta_2 := \frac{L^{\frac{1}{3}} D_0^{\frac{2}{3}}}{\lambda^{\frac{2}{3}} T^{\frac{2}{3}} \sigma^{\frac{2}{3}}},$$

Then $\eta = \min \{\eta_1, \eta_2\}$. Since $\varphi_1, \varphi_2, \varphi_3$ are increasing we have

$$\varphi_1(\eta) \le \varphi_1(\eta_1) \le \frac{3\sigma^2}{\lambda M T} \log \left( e + \frac{\lambda L M T D_0^2}{\sigma^2} \right),$$

$$\varphi_2(\eta) \le \varphi_2(\eta_1) \le \frac{16200 L^2 K^3 \sigma^2}{\lambda^3 T^4} \log^4 \left( e + \frac{\lambda^3 T^4 D_0^2}{L K^3 \sigma^2} \right),$$

$$\varphi_3(\eta) \le \varphi_3(\eta_2) \le \frac{200 L^{\frac{2}{3}} K D_0^{\frac{4}{3}} \sigma^{\frac{2}{3}}}{\lambda^{\frac{1}{3}} T^{\frac{4}{3}}}.$$

On the other hand, since $\varphi_0$ is decreasing we have $\varphi_0(\eta) \le \varphi_0(\eta_1) + \varphi_0(\eta_2) + \varphi_0(\frac{1}{L+\lambda})$, where

$$\varphi_0(\eta_1) \le \frac{\sigma^2}{2\lambda M T} + \frac{L^2 K^3 \sigma^2}{2\lambda^3 T^4},$$

$$\varphi_0(\eta_2) \le \frac{2!}{2} L D_0^2 \left( \sqrt{\frac{\eta_2 \lambda T^2}{9K}} \right)^{-2} = \frac{9K L D_0^2}{\eta_2 \lambda T^2} = \frac{9 L^{\frac{2}{3}} K D_0^{\frac{4}{3}} \sigma^{\frac{2}{3}}}{\lambda^{\frac{1}{3}} T^{\frac{4}{3}}}.$$

Combining the above bounds completes the proof. □

Theorem E.2 then follows by plugging in an appropriate $\lambda$.

*Proof of Theorem E.2.* To simplify the notation, define the terms on the RHS of Eq. (E.13) as

$$\psi_0(\lambda) := \frac{1}{2} \lambda D_0^2, \qquad\qquad\qquad \psi_1(\lambda) := \frac{1}{2} L D_0^2 \exp \left( -\sqrt{\frac{T^2}{9(1+L/\lambda)K}} \right),$$

$$\psi_2(\lambda) := \frac{209 L^{\frac{2}{3}} K D_0^{\frac{4}{3}} \sigma^{\frac{2}{3}}}{\lambda^{\frac{1}{3}} T^{\frac{4}{3}}}, \qquad\qquad \psi_3(\lambda) := \frac{4\sigma^2}{\lambda M T} \log \left( e + \frac{\lambda L M T D_0^2}{\sigma^2} \right),$$

$$\psi_4(\lambda) := \frac{16201 L^2 K^3 \sigma^2}{\lambda^3 T^4} \log^4 \left( e^4 + \frac{\lambda^3 T^4 D_0^2}{L K^3 \sigma^2} \right).$$

Define

$$\lambda_1 := \frac{\sigma}{M^{\frac{1}{2}}T^{\frac{1}{2}}D_0}, \quad \lambda_2 := \frac{L^{\frac{1}{2}}K^{\frac{3}{4}}\sigma^{\frac{1}{2}}}{D_0^{\frac{1}{2}}T}, \quad \lambda_3 := \frac{18LK}{T^2}\log^2\left(e^2 + \frac{T^2}{K}\right).$$

Then $\lambda = \max\{\lambda_1, \lambda_2, \lambda_3\}$. By helper Lemma G.5 $\psi_3, \psi_4$ are decreasing; $\psi_2$ is trivially decreasing, thus

$$\psi_2(\lambda) \leq \psi_2(\lambda_2) = \frac{209L^{\frac{1}{2}}K^{\frac{3}{4}}D_0^{\frac{3}{2}}\sigma^{\frac{1}{2}}}{T},$$

$$\psi_3(\lambda) \leq \psi_3(\lambda_1) = \frac{4\sigma D_0}{M^{\frac{1}{2}}T^{\frac{1}{2}}}\log\left(e + \frac{LM^{\frac{1}{2}}T^{\frac{1}{2}}D_0}{\sigma}\right),$$

$$\psi_4(\lambda) \leq \psi_4(\lambda_2) = \frac{16201L^{\frac{1}{2}}K^{\frac{3}{4}}D_0^{\frac{3}{2}}\sigma^{\frac{1}{2}}}{T}\log^4\left(e^4 + \frac{L^{\frac{1}{2}}TD_0^{\frac{1}{2}}}{K^{\frac{3}{4}}\sigma^{\frac{1}{2}}}\right).$$

For $\psi_1(\lambda)$ since $T \geq 1000$ we have $\frac{T^2}{K} \geq 1000$, thus

$$\frac{\lambda_3}{L} = \frac{18K}{T^2}\log^2\left(e^2 + \frac{T^2}{K}\right) \leq \frac{18}{1000}\log^2\left(e^2 + 1000\right) < 1.$$

Thus $1 + \frac{L}{\lambda_3} \leq \frac{2L}{\lambda_3}$, and therefore

$$\psi_1(\lambda) \leq \psi_1(\lambda_3) = \frac{1}{2}LD_0^2\left(e^2 + \frac{T^2}{K}\right)^{-1} \leq \frac{LKD_0^2}{2T^2}.$$

Finally

$$\psi_0(\lambda) \leq \sum_{i=1}^3 \psi_0(\lambda_i) \leq \frac{\sigma D_0}{2M^{\frac{1}{2}}T^{\frac{1}{2}}} + \frac{L^{\frac{1}{2}}K^{\frac{3}{4}}D_0^{\frac{3}{2}}\sigma^{\frac{1}{2}}}{2T} + \frac{9LKD_0^2}{T^2}\log^2\left(e^2 + \frac{T^2}{K}\right).$$

Consequently,

$$\sum_{i=0}^4 \psi(\lambda) \leq \frac{10LKD_0^2}{T^2}\log^2\left(e^2 + \frac{T^2}{K}\right) + \frac{5\sigma D_0}{M^{\frac{1}{2}}T^{\frac{1}{2}}}\log\left(e + \frac{LM^{\frac{1}{2}}T^{\frac{1}{2}}D_0}{\sigma}\right)$$
$$+ \frac{16411L^{\frac{1}{2}}K^{\frac{3}{4}}D_0^{\frac{3}{2}}\sigma^{\frac{1}{2}}}{T}\log^4\left(e^4 + \frac{L^{\frac{1}{2}}TD_0^{\frac{1}{2}}}{K^{\frac{3}{4}}\sigma^{\frac{1}{2}}}\right),$$

completing the proof. $\qquad\square$

### E.3.1 Proof of Lemma E.8

Lemma E.8 is parallel to Lemma E.5 where the main difference is the following supporting proposition.

**Proposition E.10.** *Assume $F$ is general convex and $L$-smooth, and let $\Phi_t$ be the centralized potential Eq. (C.1) for $\tilde{F}_\lambda$ (with strong convexity estimate $\mu = \lambda$), namely*

$$\Phi_t := \left(\tilde{F}_\lambda(\overline{w_t^{\mathrm{ag}}}) - \tilde{F}_\lambda^*\right) + \frac{1}{6}\lambda\|\overline{w_T} - w_\lambda^*\|^2.$$

*Then*

$$\Phi_T \geq F(\overline{w_T^{\mathrm{ag}}}) - F^* - \frac{1}{2}\lambda D_0^2, \qquad \Phi_0 \leq \frac{1}{2}L\|w_0 - w^*\|^2.$$

*Proof of Proposition E.10.* The proof is almost identical to Proposition E.7. $\qquad\square$

*Proof of Lemma E.8.* Follows by applying Lemma C.15 and plugging in the bound of Proposition E.10. The rest of proof is the same as Lemma E.5 which we omit the details. $\qquad\square$

### E.4 Proof of Theorem E.3 on FEDAC-II for general-convex objectives under Assumption 2

We omit some of the proof details since the proof is similar to Theorem E.1. We first introduce the supporting lemma for Theorem E.3.

**Lemma E.11.** *Assume Assumption 2 where $F$ is general convex, then for any $\lambda > 0$, for any $\eta \le \frac{1}{L+\lambda}$, applying FEDAC-II to $\tilde{F}_\lambda$ gives*

$$\mathbb{E}\left[F(\overline{w_T^{\mathrm{ag}}}) - F^*\right] \le \frac{1}{2}\lambda D_0^2 + \frac{1}{2}LD_0^2 \exp\left(-\sqrt{\frac{\eta\lambda T^2}{9K}}\right)$$

$$+ \frac{\eta^{\frac{1}{2}}\sigma^2}{\lambda^{\frac{1}{2}}MK^{\frac{1}{2}}} + \frac{2\eta^{\frac{3}{2}}LK^{\frac{1}{2}}\sigma^2}{\lambda^{\frac{1}{2}}M} + \frac{2\eta^{\frac{3}{2}}\lambda^{\frac{1}{2}}K^{\frac{1}{2}}\sigma^2}{M} + \frac{\mathrm{e}^9\eta^4Q^2K^2\sigma^4}{\lambda}. \qquad (\mathrm{E}.14)$$

*Proof of Lemma E.11.* Follows by Lemma C.4 and Proposition E.10. The proof is similar to Lemma E.5 so we omit the details. □

**Lemma E.12.** *Assume Assumption 2 where $F$ is general convex, then for any $\lambda > 0$, for*

$$\eta = \min\left\{\frac{1}{L+\lambda}, \frac{9K}{\lambda T^2}\log^2\left(\mathrm{e} + \min\left\{\frac{\lambda LMTD_0^2}{\sigma^2}, \frac{\lambda^2 MT^3 D_0^2}{K^2\sigma^2}, \frac{\lambda^5 LT^8 D_0^2}{Q^2 K^6 \sigma^4}\right\}\right), \frac{L^{\frac{1}{3}}K^{\frac{1}{3}}M^{\frac{1}{3}}D_0^{\frac{2}{3}}}{\lambda^{\frac{2}{3}}T\sigma^{\frac{2}{3}}}\right\},$$

*applying FEDAC-II to $\tilde{F}_\lambda$ gives*

$$\mathbb{E}\left[F(\overline{w_T^{\mathrm{ag}}}) - F^*\right] \le \frac{1}{2}\lambda D_0^2 + \frac{1}{2}LD_0^2 \exp\left(-\sqrt{\frac{T^2}{9(1+L/\lambda)K}}\right) + \frac{4\sigma^2}{\lambda MT}\log\left(\mathrm{e} + \frac{\lambda LMTD_0^2}{\sigma^2}\right)$$

$$+ \frac{55LK^2\sigma^2}{\lambda^2 MT^3}\log^3\left(\mathrm{e}^3 + \frac{\lambda^2 MT^3 D_0^2}{K^2\sigma^2}\right) + \frac{83L^{\frac{1}{2}}KD_0\sigma}{\lambda^{\frac{1}{2}}M^{\frac{1}{2}}T^{\frac{3}{2}}} + \frac{\mathrm{e}^{18}Q^2 K^6 \sigma^4}{\lambda^5 T^8}\log^8\left(\mathrm{e}^8 + \frac{\lambda^5 LT^8 D_0^2}{Q^2 K^6 \sigma^4}\right). \qquad (\mathrm{E}.15)$$

*Proof of Lemma E.12.* To simplify the notation, define the terms on the RHS of Eq. (E.14) as

$$\varphi_0(\eta) := \frac{1}{2}LD_0^2 \exp\left(-\sqrt{\frac{\eta\lambda T^2}{9K}}\right), \quad \varphi_1(\eta) := \frac{\eta^{\frac{1}{2}}\sigma^2}{\lambda^{\frac{1}{2}}MK^{\frac{1}{2}}}, \quad \varphi_2(\eta) := \frac{2\eta^{\frac{3}{2}}LK^{\frac{1}{2}}\sigma^2}{\lambda^{\frac{1}{2}}M},$$

$$\varphi_3(\eta) := \frac{2\eta^{\frac{3}{2}}\lambda^{\frac{1}{2}}K^{\frac{1}{2}}\sigma^2}{M}, \qquad \varphi_4(\eta) := \frac{\mathrm{e}^9\eta^4Q^2K^2\sigma^4}{\lambda}.$$

Define

$$\eta_1 := \frac{9K}{\lambda T^2}\log^2\left(\mathrm{e} + \min\left\{\frac{\lambda LMTD_0^2}{\sigma^2}, \frac{\lambda^2 MT^3 D_0^2}{K^2\sigma^2}, \frac{\lambda^5 LT^8 D_0^2}{Q^2 K^6 \sigma^4}\right\}\right), \quad \eta_2 := \frac{L^{\frac{1}{3}}K^{\frac{1}{3}}M^{\frac{1}{3}}D_0^{\frac{2}{3}}}{\lambda^{\frac{2}{3}}T\sigma^{\frac{2}{3}}}.$$

Then $\eta = \min\{\eta_1, \eta_2\}$. Since $\varphi_1, \ldots, \varphi_4$ are increasing we have

$$\varphi_1(\eta) \le \varphi_1(\eta_1) \le \frac{3\sigma^2}{\lambda MT}\log\left(\mathrm{e} + \frac{\lambda LMTD_0^2}{\sigma^2}\right),$$

$$\varphi_2(\eta) \le \varphi_2(\eta_1) \le \frac{54LK^2\sigma^2}{\lambda^2 MT^3}\log^3\left(\mathrm{e} + \frac{\lambda^2 MT^3 D_0^2}{K^2\sigma^2}\right),$$

$$\varphi_3(\eta) \le \varphi_3(\eta_2) = \frac{2L^{\frac{1}{2}}KD_0\sigma}{\lambda^{\frac{1}{2}}M^{\frac{1}{2}}T^{\frac{3}{2}}},$$

$$\varphi_4(\eta) \le \varphi_4(\eta_1) \le \frac{9^4\mathrm{e}^9 Q^2 K^6 \sigma^4}{\lambda^5 T^8}\log^8\left(\mathrm{e} + \frac{\lambda^5 LT^8 D_0^2}{Q^2 K^6 \sigma^4}\right).$$

On the other hand $\varphi_0(\eta) \le \varphi_0(\eta_1) + \varphi_0(\eta_2) + \varphi_0(\frac{1}{L+\lambda})$, where

$$\varphi_0(\eta_1) \le \frac{\sigma^2}{2\lambda MT} + \frac{LK^2\sigma^2}{2\lambda^2 MT^3} + \frac{Q^2 K^6 \sigma^4}{2\lambda^5 T^8},$$

$$\varphi_0(\eta_2) \le \frac{3!}{2}LD_0^2\left(\sqrt{\frac{\eta_2\lambda T^2}{9K}}\right)^{-3} = \frac{81LK^{\frac{3}{2}}D_0^2}{\eta_2^{\frac{3}{2}}\lambda^{\frac{3}{2}}T^3} = \frac{81L^{\frac{1}{2}}KD_0\sigma}{\lambda^{\frac{1}{2}}M^{\frac{1}{2}}T^{\frac{3}{2}}}.$$

Combining the above bounds completes the proof. □

Theorem E.3 then follows by plugging in an appropriate $\lambda$.

*Proof of Theorem E.3.* To simplify the notation, define the terms on the RHS of Eq. (E.15) as

$$\psi_0(\lambda) := \frac{1}{2}\lambda D_0^2, \qquad\qquad \psi_1(\lambda) := \frac{1}{2}LD_0^2 \exp\left(-\sqrt{\frac{T^2}{9(1+L/\lambda)K}}\right),$$

$$\psi_2(\lambda) := \frac{4\sigma^2}{\lambda MT}\log\left(\mathrm{e}+\frac{\lambda LMTD_0^2}{\sigma^2}\right), \psi_3(\lambda) := \frac{55LK^2\sigma^2}{\lambda^2 MT^3}\log^3\left(\mathrm{e}^3+\frac{\lambda^2 MT^3 D_0^2}{K^2\sigma^2}\right),$$

$$\psi_4(\lambda) := \frac{83L^{\frac{1}{2}}KD_0\sigma}{\lambda^{\frac{1}{2}}M^{\frac{1}{2}}T^{\frac{3}{2}}}, \qquad\qquad \psi_5(\lambda) := \frac{\mathrm{e}^{18}Q^2K^6\sigma^4}{\lambda^5 T^8}\log^8\left(\mathrm{e}^8+\frac{\lambda^5 LT^8 D_0^2}{Q^2 K^6\sigma^4}\right).$$

Define

$$\lambda_1 := \frac{\sigma}{M^{\frac{1}{2}}T^{\frac{1}{2}}D_0}, \quad \lambda_2 := \frac{L^{\frac{1}{3}}K^{\frac{2}{3}}\sigma^{\frac{2}{3}}}{M^{\frac{1}{3}}TD_0^{\frac{2}{3}}}, \quad \lambda_3 := \frac{Q^{\frac{1}{3}}K\sigma^{\frac{2}{3}}}{D_0^{\frac{2}{3}}T^{\frac{4}{3}}}, \quad \lambda_4 := \frac{18LK}{T^2}\log^2\left(\mathrm{e}^2+\frac{T^2}{K}\right).$$

Then $\lambda = \max\{\lambda_1,\lambda_2,\lambda_3\}$. By Lemma G.5, $\psi_2, \psi_3, \psi_5$ are increasing. $\psi_4$ is trivially decreasing, thus

$$\psi_2(\lambda) \le \psi_2(\lambda_1) = \frac{4\sigma D_0}{M^{\frac{1}{2}}T^{\frac{1}{2}}}\log\left(\mathrm{e}+\frac{LM^{\frac{1}{2}}T^{\frac{1}{2}}D_0}{\sigma}\right),$$

$$\psi_3(\lambda) \le \psi_3(\lambda_2) = \frac{55L^{\frac{1}{3}}K^{\frac{2}{3}}D_0^{\frac{4}{3}}\sigma^{\frac{2}{3}}}{M^{\frac{1}{3}}T}\log^3\left(\mathrm{e}^3+\frac{L^{\frac{2}{3}}M^{\frac{1}{3}}TD_0^{\frac{2}{3}}}{K^{\frac{2}{3}}\sigma^{\frac{2}{3}}}\right),$$

$$\psi_4(\lambda) \le \psi_4(\lambda_2) = \frac{83L^{\frac{1}{3}}K^{\frac{2}{3}}D_0^{\frac{4}{3}}\sigma^{\frac{2}{3}}}{M^{\frac{1}{3}}T},$$

$$\psi_5(\lambda) \le \psi_5(\lambda_3) = \frac{\mathrm{e}^{18}Q^{\frac{1}{3}}KD_0^{\frac{5}{3}}\sigma^{\frac{2}{3}}}{T^{\frac{4}{3}}}\log^8\left(\mathrm{e}^8+\frac{LT^{\frac{4}{3}}D_0^{\frac{1}{3}}}{Q^{\frac{1}{3}}K\sigma^{\frac{2}{3}}}\right).$$

For $\psi_1(\lambda)$ since $T \ge 1000$ we have $\frac{T^2}{K} \ge 1000$, thus

$$\frac{\lambda_3}{L} = \frac{18K}{T^2}\log^2\left(\mathrm{e}^2+\frac{T^2}{K}\right) \le \frac{18}{1000}\log^2\left(\mathrm{e}^2+1000\right) < 1.$$

Thus $1 + \frac{L}{\lambda_3} \le \frac{2L}{\lambda_3}$, and therefore

$$\psi_1(\lambda) \le \psi_1(\lambda_3) = \frac{1}{2}LD_0^2\left(\mathrm{e}^2+\frac{T^2}{K}\right)^{-1} \le \frac{LKD_0^2}{2T^2}.$$

Finally

$$\psi_0(\lambda) \le \sum_{i=1}^{4}\psi_0(\lambda_i) \le \frac{\sigma D_0}{2M^{\frac{1}{2}}T^{\frac{1}{2}}} + \frac{L^{\frac{1}{3}}K^{\frac{2}{3}}D_0^{\frac{4}{3}}\sigma^{\frac{2}{3}}}{2M^{\frac{1}{3}}T} + \frac{Q^{\frac{1}{3}}KD_0^{\frac{5}{3}}\sigma^{\frac{2}{3}}}{2T^{\frac{4}{3}}} + \frac{9LKD_0^2}{T^2}\log^2\left(\mathrm{e}^2+\frac{T^2}{K}\right).$$

Consequently,

$$\sum_{i=0}^{4}\psi(\lambda) \le \frac{10LKD_0^2}{T^2}\log^2\left(\mathrm{e}^2+\frac{T^2}{K}\right) + \frac{5\sigma D_0}{M^{\frac{1}{2}}T^{\frac{1}{2}}}\log\left(\mathrm{e}+\frac{LM^{\frac{1}{2}}T^{\frac{1}{2}}D_0}{\sigma}\right)$$

$$+ \frac{139L^{\frac{1}{3}}K^{\frac{2}{3}}\sigma^{\frac{2}{3}}D_0^{\frac{4}{3}}}{M^{\frac{1}{3}}T}\log^3\left(\mathrm{e}^3+\frac{L^{\frac{2}{3}}M^{\frac{1}{3}}TD_0^{\frac{2}{3}}}{K^{\frac{2}{3}}\sigma^{\frac{2}{3}}}\right) + \frac{\mathrm{e}^{19}Q^{\frac{1}{3}}K\sigma^{\frac{2}{3}}D_0^{\frac{5}{3}}}{T^{\frac{4}{3}}}\log^8\left(\mathrm{e}^8+\frac{LT^{\frac{4}{3}}D_0^{\frac{1}{3}}}{Q^{\frac{1}{3}}K\sigma^{\frac{2}{3}}}\right).$$

$\square$

### E.5 Proof of Theorem E.4 on FEDAVG for general-convex objectives under Assumption 2

We omit some of the proof details since the proof is similar to Theorem E.1. We first introduce the supporting lemma for Theorem E.4.

**Lemma E.13.** *Assume Assumption 2 where $F$ is general convex, then for any $\lambda > 0$, for*

$$\eta := \min\left\{\frac{1}{4(L+\lambda)}, \frac{2}{\lambda T}\log\left(e + \min\left\{\frac{\lambda^2 M T^2 D_0^2}{\sigma^2}, \frac{\lambda^6 T^5 D_0^2}{Q^2 K^2 \sigma^4}\right\}\right)\right\},$$

*applying FEDAVG to $\tilde{F}_\lambda$ gives*

$$\mathbb{E}\left[F\left(\sum_{t=0}^{T-1}\frac{\rho_t}{S_T}\overline{w}_t\right) - F^*\right] \leq 3\lambda D_0^2 + 2L D_0^2 \exp\left(-\frac{\lambda T}{8(L+\lambda)}\right)$$

$$+ \frac{3\sigma^2}{\lambda M T}\log\left(e^2 + \frac{\lambda^2 M T^2 D_0^2}{\sigma^2}\right) + \frac{3073 Q^2 K^2 \sigma^4}{\lambda^5 T^4}\log^4\left(e^5 + \frac{\lambda^6 T^5 D_0^2}{Q^2 K^2 \sigma^4}\right), \qquad \text{(E.16)}$$

*where $\rho_t := (1 - \frac{1}{2}\eta\lambda)^{T-t-1}$, $S_T := \sum_{t=0}^{T-1}\rho_t$, and $D_0 = \|\overline{w}_0 - w^*\|$.*

*Proof of Lemma E.13.* Apply Theorem D.1. The rest of analysis is similar to Lemmas E.5 and E.6.
□

*Proof of Theorem E.4.* To simplify the notation, define the RHS of Eq. (E.16) as

$$\psi_0(\lambda) := 3\lambda D_0^2, \qquad\qquad \psi_1(\lambda) := 2L D_0^2 \exp\left(-\frac{T}{8(1+(L/\lambda))}\right),$$

$$\psi_2(\lambda) := \frac{3\sigma^2}{\lambda M T}\log\left(e^2 + \frac{\lambda^2 M T^2 D_0^2}{\sigma^2}\right), \psi_3(\lambda) := \frac{3073 Q^2 K^2 \sigma^4}{\lambda^5 T^4}\log^4\left(e^5 + \frac{\lambda^6 T^5 D_0^2}{Q^2 K^2 \sigma^4}\right).$$

Define

$$\lambda_1 := \frac{\sigma}{M^{\frac{1}{2}}T^{\frac{1}{2}}D_0}, \quad \lambda_2 := \frac{Q^{\frac{1}{3}}K^{\frac{1}{3}}\sigma^{\frac{2}{3}}}{T^{\frac{2}{3}}D_0^{\frac{1}{3}}}, \quad \lambda_3 := \frac{16L}{T}\log(e+T).$$

Then $\lambda = \max\{\lambda_1, \lambda_2, \lambda_3\}$. We have (by helper Lemma G.5 $\psi_2, \psi_3$ are decreasing)

$$\psi_2(\lambda) \leq \psi_2(\lambda_1) \leq \frac{3\sigma D_0}{M^{\frac{1}{2}}T^{\frac{1}{2}}}\log\left(e^2 + T\right),$$

$$\psi_3(\lambda) \leq \psi_3(\lambda_2) \leq \frac{3073 Q^{\frac{1}{3}}K^{\frac{1}{3}}\sigma^{\frac{2}{3}}D_0^{\frac{5}{3}}}{T^{\frac{2}{3}}}\log^4\left(e^5 + T\right).$$

Since $T \geq 100$ we have (by helper Lemma G.5, $x^{-1}\log(e+x)$ is decreasing)

$$\frac{\lambda_3}{L} = \frac{16}{T}\log(e+T) \leq \frac{16}{100}\log(e+100) < 1,$$

and thus

$$\psi_1(\lambda) \leq \psi_1(\lambda_3) \leq 2L D_0^2 \exp\left(-\frac{T}{16(L/\lambda_3)}\right) = 2L D_0^2(e+T)^{-1} \leq \frac{2L D_0^2}{T}.$$

Finally

$$\psi_0(\lambda) \leq \sum_{i=1}^{3}\psi_0(\lambda_i) = \frac{3\sigma D_0}{M^{\frac{1}{2}}T^{\frac{1}{2}}} + \frac{3Q^{\frac{1}{3}}K^{\frac{1}{3}}\sigma^{\frac{2}{3}}D_0^{\frac{5}{3}}}{T^{\frac{2}{3}}} + \frac{48L D_0^2}{T}\log(e+T).$$

Accordingly

$$\sum_{i=0}^{3}\psi_i(\lambda) \leq \frac{50L D_0^2}{T}\log(e+T) + \frac{6\sigma D_0}{M^{\frac{1}{2}}T^{\frac{1}{2}}}\log\left(e^2 + T\right) + \frac{3076 Q^{\frac{1}{3}}K^{\frac{1}{3}}\sigma^{\frac{2}{3}}D_0^{\frac{5}{3}}}{T^{\frac{2}{3}}}\log^4\left(e^5 + T\right).$$

□

# F Initial-value instability of standard accelerated gradient descent

## F.1 Main theorem and lemmas

In this section we show that standard accelerated gradient descent [Nesterov, 2018] may not be initial-value stable even for strongly convex and smooth objectives in the sense that the initial infinitesimal difference may grow exponentially fast. This provides an evidence on the necessity of acceleration-stability tradeoff.

We formally define the standard deterministic AGD in Algorithm 3 for $L$-smooth and $\mu$-strongly-convex objective $F$ [Nesterov, 2018].

---

**Algorithm 3** Nesterov's Accelerated Gradient Descent Method (AGD)

---
1: **procedure** AGD($w_0^{\mathrm{ag}}, w_0, L, \mu$)
2:      $\kappa \leftarrow L/\mu$
3:      **for** $t = 0, \ldots, T-1$ **do**
4:          $w_t^{\mathrm{md}} \leftarrow \frac{1}{\sqrt{\kappa}+1} w_t + \frac{\sqrt{\kappa}}{\sqrt{\kappa}+1} w_t^{\mathrm{ag}}$
5:          $w_{t+1}^{\mathrm{ag}} \leftarrow w_t^{\mathrm{md}} - \frac{1}{L} \nabla F(w_t^{\mathrm{md}})$
6:          $w_{t+1} \leftarrow \left(1 - \frac{1}{\sqrt{\kappa}}\right) w_t + \frac{1}{\sqrt{\kappa}} w_t^{\mathrm{md}} - \sqrt{\frac{1}{L\mu}} \nabla F(w_t^{\mathrm{md}})$

---

Now we introduce the formal theorem on the initial-value instability.

**Theorem F.1** (Initial-value instability of deterministic standard AGD, complete version of Theorem 4.2)**.** *For any $L, \mu > 0$ such that $L/\mu \geq 25$, and for any $K \geq 1$, there exists a 1D objective $F$ that is $L$-smooth and $\mu$-strongly-convex, and an $\varepsilon_0 > 0$, such that for any positive $\varepsilon < \varepsilon_0$, there exists $w_0, u_0, w_0^{\mathrm{ag}}, u_0^{\mathrm{ag}}$ such that $|w_0 - u_0| \leq \varepsilon$, $|w_0^{\mathrm{ag}} - u_0^{\mathrm{ag}}| \leq \varepsilon$, but the sequence $\{w_t^{\mathrm{ag}}, w_t^{\mathrm{md}}, w_t\}_{t=0}^{3K}$ output by AGD($w_0^{\mathrm{ag}}, w_0, L, \mu$) and sequence $\{u_t^{\mathrm{ag}}, u_t^{\mathrm{md}}, u_t\}_{t=0}^{3K}$ output by AGD($u_0^{\mathrm{ag}}, u_0, L, \mu$) satisfies*

$$|w_{3K} - u_{3K}| \geq \frac{1}{2} \varepsilon (1.02)^K, \qquad |w_{3K}^{\mathrm{ag}} - u_{3K}^{\mathrm{ag}}| \geq \varepsilon (1.02)^K.$$

**Remark.** *It is worth mentioning that the instability theorem **does not contradicts the convergence** of AGD [Nesterov, 2018]. The convergence of AGD suggests that $w_t^{\mathrm{ag}}$, $w_t$, $u_t^{\mathrm{ag}}$, and $u_t$ will all converge to the same point $w^*$ as $t \to \infty$, which implies $\lim_{t\to\infty} \|w_t^{\mathrm{ag}} - u_t^{\mathrm{ag}}\| = \|w_t - u_t\| = 0$. However, the convergence theorem does not imply the stability with respect to the initialization — it does not exclude the possibility that the difference between two instances (possibly with very close initialization) first expand and only shrink until they both approach $w^*$. Our Theorem 4.2 suggests this possibility: for any finite steps, no matter how small the (positive) initial difference is, it is possible that the difference will grow exponentially fast. This is fundamentally different from the Gradient Descent (for convex objectives), for which the difference between two instances does not expand for standard choice of learning rate $\eta = \frac{1}{L}$ (where $L$ is the smoothness).*

We first introduce the supporting lemmas for Theorem 4.2. Lemma F.2 shows the existence of an objective $F$ and a trajectory of AGD on $F$ such that $F''(w_t^{\mathrm{md}}) = L$ (including also the neighborhood) once every three steps and $F''(w_t^{\mathrm{md}}) = \mu$ otherwise. The proof of Lemma F.2 is deferred to Section F.2.

**Lemma F.2.** *For any $L > \mu > 0$, and for any $K \geq 1$, there exists a 1D objective $F$ that is $L$-smooth and $\mu$-strongly convex, a neighborhood bound $\delta > 0$, and initial points $w_0$ and $w_0^{\mathrm{ag}}$ such that the sequence $\{w_t^{\mathrm{ag}}, w_t^{\mathrm{md}}, w_t\}_{t=0}^{3K-1}$ output by AGD($w_0^{\mathrm{ag}}, w_0, L, \mu$) satisfies for any $t = 0, \ldots, 3K-1$,*

$$\text{if } t \bmod 3 \neq 1, \text{ then } F''(w) \equiv \mu, \text{ for all } w \in [w_t^{\mathrm{md}} - \delta, w_t^{\mathrm{md}} + \delta],$$

$$\text{if } t \bmod 3 = 1, \text{ then } F''(w) \equiv L, \text{ for all } w \in [w_t^{\mathrm{md}} - \delta, w_t^{\mathrm{md}} + \delta].$$

The following Lemma F.3 analyzes the growth of the difference of two instances of AGD. The proof is very similar to the analysis of FEDAC.

**Lemma F.3.** *Let $F$ be a $L$-smooth and $\mu > 0$-strongly convex 1D function. Let $(w_{t+1}^{\mathrm{ag}}, w_{t+1})$, $(u_{t+1}^{\mathrm{ag}}, u_{t+1})$ be generated by applying one step of AGD on $F$ with hyperparameter $(L, \mu)$ from*

$(w_t^{\mathrm{ag}}, w_t)$ and $(u_t^{\mathrm{ag}}, u_t)$, respectively. Then there exists a $\zeta_t$ within the interval between $w_t^{\mathrm{md}}$ and $u_t^{\mathrm{md}}$, such that

$$\begin{bmatrix} w_{t+1}^{\mathrm{ag}} - u_{t+1}^{\mathrm{ag}} \\ w_{t+1} - u_{t+1} \end{bmatrix} = \begin{bmatrix} \frac{\sqrt{\kappa}}{\sqrt{\kappa}+1}\left(1 - \frac{1}{L}F''(\zeta_t)\right) & \frac{1}{\sqrt{\kappa}+1}\left(1 - \frac{1}{L}F''(\zeta_t)\right) \\ \frac{1}{\sqrt{\kappa}+1}\left(1 - \frac{1}{\mu}F''(\zeta_t)\right) & \frac{\sqrt{\kappa}}{\sqrt{\kappa}+1}\left(1 - \frac{1}{L}F''(\zeta_t)\right) \end{bmatrix} \begin{bmatrix} w_t^{\mathrm{ag}} - u_t^{\mathrm{ag}} \\ w_t - u_t \end{bmatrix}.$$

*Proof of Lemma F.3.* This is a special case of Claim B.12 with no noise. $\qquad\square$

With Lemmas F.2 and F.3 at hand we are ready to prove Theorem F.1. The proof follows by constructing an auxiliary trajectory for around the one given by Lemma F.2.

*Proof of Theorem F.1.* First apply Lemma F.2. Let $F$ be the objective, $(w_0^{\mathrm{ag}}, w_0)$ be the initial point and $\delta$ be the neighborhood bound given by Lemma F.2. Since $\{w_t^{\mathrm{ag}}, w_t^{\mathrm{md}}, w_t\}_{t=0}^{3K-1}$ is a continuous function with respect to the initial point $(w_0^{\mathrm{ag}}, w_0)$, there exists a $\varepsilon_0$ such that for any $(v_0^{\mathrm{ag}}, v_0)$ such that $|v_0^{\mathrm{ag}} - w_0^{\mathrm{ag}}| \le \varepsilon_0$ and $|v_0 - w_0| \le \varepsilon_0$, trajectory $\{v_t^{\mathrm{ag}}, v_t^{\mathrm{md}}, v_t\}_{t=0}^{3K}$ output by AGD $(v_0^{\mathrm{ag}}, v_0, L, \mu)$ satisfies $\max_{0 \le t < 3K} |v_t^{\mathrm{md}} - w_t^{\mathrm{md}}| \le \delta$.

Thus, by Lemma F.3, for any $t = 0, \ldots, 3K - 1$,

$$\begin{bmatrix} w_{t+1}^{\mathrm{ag}} - v_{t+1}^{\mathrm{ag}} \\ w_{t+1} - v_{t+1} \end{bmatrix} = \begin{bmatrix} 1 - \frac{1}{\sqrt{\kappa}} & \frac{1}{\kappa}(\sqrt{\kappa} - 1) \\ 0 & 1 - \frac{1}{\sqrt{\kappa}} \end{bmatrix} \begin{bmatrix} w_t^{\mathrm{ag}} - v_t^{\mathrm{ag}} \\ w_t - v_t \end{bmatrix}, \quad \text{if } t \bmod 3 \ne 1;$$

$$\begin{bmatrix} w_{t+1}^{\mathrm{ag}} - v_{t+1}^{\mathrm{ag}} \\ w_{t+1} - v_{t+1} \end{bmatrix} = \begin{bmatrix} 0 & 0 \\ 1 - \sqrt{\kappa} & 0 \end{bmatrix} \begin{bmatrix} w_t^{\mathrm{ag}} - v_t^{\mathrm{ag}} \\ w_t - v_t \end{bmatrix}, \quad \text{if } t \bmod 3 = 1.$$

Hence for any $k = 0, \ldots, K - 1$,

$$\begin{bmatrix} w_{3(k+1)}^{\mathrm{ag}} - v_{3(k+1)}^{\mathrm{ag}} \\ w_{3(k+1)} - v_{3(k+1)} \end{bmatrix} = -\begin{bmatrix} \frac{1}{\kappa^{\frac{3}{2}}}(\sqrt{\kappa} - 1)^3 & \frac{1}{\kappa^2}(\sqrt{\kappa} - 1)^3 \\ \frac{1}{\kappa}(\sqrt{\kappa} - 1)^3 & \frac{1}{\kappa^{\frac{3}{2}}}(\sqrt{\kappa} - 1)^3 \end{bmatrix} \begin{bmatrix} w_{3k}^{\mathrm{ag}} - v_{3k}^{\mathrm{ag}} \\ w_{3k} - v_{3k} \end{bmatrix}$$

$$= -2\left(1 - \frac{1}{\sqrt{\kappa}}\right)^3 \begin{bmatrix} \frac{1}{2} & \frac{1}{2\sqrt{\kappa}} \\ \frac{1}{2}\sqrt{\kappa} & \frac{1}{2} \end{bmatrix} \begin{bmatrix} w_{3k}^{\mathrm{ag}} - v_{3k}^{\mathrm{ag}} \\ w_{3k} - v_{3k} \end{bmatrix}.$$

Note that $\begin{bmatrix} \frac{1}{2} & \frac{1}{2\sqrt{\kappa}} \\ \frac{1}{2}\sqrt{\kappa} & \frac{1}{2} \end{bmatrix}$ is idempotent, *i.e.*, $\begin{bmatrix} \frac{1}{2} & \frac{1}{2\sqrt{\kappa}} \\ \frac{1}{2}\sqrt{\kappa} & \frac{1}{2} \end{bmatrix}^K = \begin{bmatrix} \frac{1}{2} & \frac{1}{2\sqrt{\kappa}} \\ \frac{1}{2}\sqrt{\kappa} & \frac{1}{2} \end{bmatrix}$. Thus

$$\begin{bmatrix} w_{3K}^{\mathrm{ag}} - v_{3K}^{\mathrm{ag}} \\ w_{3K} - v_{3K} \end{bmatrix} = \left(-2\left(1 - \frac{1}{\sqrt{\kappa}}\right)^3\right)^K \begin{bmatrix} \frac{1}{2} & \frac{1}{2\sqrt{\kappa}} \\ \frac{1}{2}\sqrt{\kappa} & \frac{1}{2} \end{bmatrix} \begin{bmatrix} w_0^{\mathrm{ag}} - v_0^{\mathrm{ag}} \\ w_0 - v_0 \end{bmatrix}.$$

Thus for any given $\varepsilon \le \varepsilon_0$, put $u_0^{\mathrm{ag}} = w_0^{\mathrm{ag}} - \varepsilon$, and $u_0 = w_0 - \varepsilon$, we have

$$\begin{bmatrix} w_{3K}^{\mathrm{ag}} - u_{3K}^{\mathrm{ag}} \\ w_{3K} - u_{3K} \end{bmatrix} = \frac{1}{2}\varepsilon\left(-2\left(1 - \frac{1}{\sqrt{\kappa}}\right)^3\right)^K \begin{bmatrix} 1 + \frac{1}{\sqrt{\kappa}} \\ \sqrt{\kappa} + 1 \end{bmatrix}.$$

For $\kappa \ge 25$ we have $\left|2\left(1 - \frac{1}{\sqrt{\kappa}}\right)^3\right| > 1.02$. Therefore

$$|w_{3K}^{\mathrm{ag}} - u_{3K}^{\mathrm{ag}}| \ge \frac{1}{2}(1.02)^K \cdot \varepsilon, \quad |w_{3K} - u_{3K}| \ge (1.02)^K \cdot \varepsilon,$$

completing the proof. $\qquad\square$

As a sanity check, the proof framework above for instability does not apply to the convergence of AGD. For instability, we only need to locally change the curvature to "separate" two instances. This trick does not break the convergence proof where the progress depends on the global curvature. We refer readers to Lessard et al. [2016] for the relative discussion.

## F.2 Proof of Lemma F.2

In this section we prove Lemma F.2 on the existence of objective $F$ and the trajectory with specific curvature at certain intervals. The high-level rationale is that Lemma F.2 only specifies local curvatures of $F$, and therefore we can modify an objective at certain local points to make Lemma F.2 satisfied. Here we provide a constructive approach by incrementally updating $F$.

We inductively prove the following claim.

**Claim F.4.** *For any $k = 0, \ldots, K$, there exists a function $H_k$ valued in $[\mu, L]$, a neighborhood bound $\delta_k > 0$, and a pair of initial points $(w_0^{\mathrm{ag}}, w_0)$, such that for objective $F_k(w) := \int_0^w \int_0^y H_k(x)\mathrm{d}x\mathrm{d}y$, the sequence output by* AGD $(w_0^{\mathrm{ag}}, w_0, L, \mu)$ *on $F_k$ satisfies $|w_{t_1}^{\mathrm{md}} - w_{t_2}^{\mathrm{md}}| \geq 2\delta_k$ if $t_1 \neq t_2$, and for any $t = 0, \ldots, 3K - 1$,*

> *if $t \bmod 3 \neq 1$ or $t \geq 3k$, then $F''(w) \equiv H_k(w) \equiv \mu$ for all $w \in [w_t^{\mathrm{md}} - \delta_k, w_t^{\mathrm{md}} + \delta_k]$;* (F.1)
>
> *if $t \bmod 3 = 1$ and $t < 3k$, then $F''(w) \equiv H_k(w) \equiv L$ for all $w \in [w_t^{\mathrm{md}} - \delta_k, w_t^{\mathrm{md}} + \delta_k]$.* (F.2)

To simplify the notation, we refer to Eqs. (F.1) and (F.2) as "curvature conditions" and denote $\mathcal{U}(x; r) := \{y : |y - x| < r\}$, and $\bar{\mathcal{U}}(x; r) := \{y : |y - x| \leq r\}$.

*Inductive proof of Claim F.4.* For $k = 0$, we can put $H_0(w) \equiv \mu$ (then $F_k(w) = \frac{1}{2}\mu w^2$) and select any arbitrary initial points $(w_0^{\mathrm{ag}}, w_0)$ as long as $w_{t_1}^{\mathrm{md}} \neq w_{t_2}^{\mathrm{md}}$ for $t_1 \neq t_2$, which is trivially possible.

Suppose Claim F.4 holds for $k$, now we construct $H_{k+1}$ and $\delta_{k+1}$. Let $\{w_{t,k}^{\mathrm{ag}}, w_{t,k}^{\mathrm{md}}, w_{t,k}\}_{t=0}^{3K-1}$ be the trajectory output by AGD $(w_0^{\mathrm{ag}}, w_0, L, \mu)$ on $F_k$. For some positive $\varepsilon_k < \frac{1}{2}\delta_k$ to be determined, consider

$$\tilde{H}_{k+1}(w) = H_k(w) + (L - \mu)\mathbf{1}\left[w \in \bar{\mathcal{U}}(w_{3k+1,k}^{\mathrm{md}}; \varepsilon_k)\right], \quad \tilde{F}_{k+1}(w) = \int_0^w \int_0^y \tilde{H}_{k+1}(x)\mathrm{d}x\mathrm{d}y.$$

Let $\{\tilde{w}_{t,k+1}^{\mathrm{ag}}, \tilde{w}_{t,k+1}^{\mathrm{md}}, \tilde{w}_{t,k+1}\}_{t=0}^{3K-1}$ be the trajectory output by AGD $(w_0^{\mathrm{ag}}, w_0, L, \mu)$ on $\tilde{F}_{k+1}$. Since the trajectory is continuous with respect to $\varepsilon_k$, there exists a $\bar{\varepsilon} < \frac{1}{2}\delta_k$ such that for any $\varepsilon_k < \bar{\varepsilon}$ (which we assume from now on), it is the case that $|\tilde{w}_{t,k+1}^{\mathrm{md}} - w_{t,k}^{\mathrm{md}}| \leq \frac{1}{2}\delta_k$ for all $t \leq 3k + 1$. Then let

$$H_{k+1}(w) = H_k(w) + (L - \mu)\mathbf{1}\left[w \in \bar{\mathcal{U}}(\tilde{w}_{3k+1,k+1}^{\mathrm{md}}; \varepsilon_k)\right], \quad F_{k+1}(w) = \int_0^w \int_0^y H_{k+1}(x)\mathrm{d}x\mathrm{d}y.$$

and let $\{w_{t,k+1}^{\mathrm{ag}}, w_{t,k+1}^{\mathrm{md}}, w_{t,k+1}\}_{t=0}^{3K-1}$ be the trajectory output by AGD $(w_0^{\mathrm{ag}}, w_0, L, \mu)$ on $F_{k+1}$.

Consequently,

  (a) By construction of $H_{k+1}$ and $\tilde{H}_{k+1}$, we have $H_{k+1}(w) = \tilde{H}_{k+1}(w) = H_k(w)$ and $\nabla F_{k+1}(w) = \nabla \tilde{F}_{k+1}(w)$ for all $w \notin \bar{U}(w_{3k+1,k}^{\mathrm{md}}; \delta_k)$.

  (b) Since $\tilde{w}_{t,k+1}^{\mathrm{md}} \notin \bar{U}(w_{3k+1,k}^{\mathrm{md}}; \delta_k)$, by (a), we can inductively show that $\tilde{w}_{t,k+1}^{\mathrm{md}} = w_{t,k+1}^{\mathrm{md}}$ for $t < 3k + 1$, namely the trajectories for $F_{k+1}$ and $\tilde{F}_{k+1}$ are identical up to timestep $t < 3k + 1$.

  (c) Since $|\tilde{w}_{t,k+1}^{\mathrm{md}} - w_{t,k}^{\mathrm{md}}| \leq \frac{1}{2}\delta_k$, by (b), we further have $|w_{t,k+1}^{\mathrm{md}} - w_{t,k}^{\mathrm{md}}| \leq \frac{1}{2}\delta_k$ for $t < 3k + 1$. Thus, by (a), the curvature conditions will be satisfied for $w_{t,k+1}^{\mathrm{md}}$ and $H_{k+1}$ up to $t < 3k + 1$ and any neighborhood bound $\delta_{k+1} < \frac{1}{2}\delta_k$ since $H_{k+1} \equiv H_k$ for $w \notin \bar{U}(w_{3k+1,k}^{\mathrm{md}}; \delta_k)$.

  (d) By (b), we have $w_{3k+1,k+1}^{\mathrm{md}} = \tilde{w}_{3k+1,k+1}^{\mathrm{md}}$ since all previous gradients evaluated are identical for $F_{k+1}$ and $\tilde{F}_{k+1}$. Thus, by construction of $H_{k+1}$ the curvature conditions hold for $w_{3k+1,k+1}^{\mathrm{md}}$ and $H_{k+1}$.

  (e) Similarly, for sufficiently small $\varepsilon_k$, we have $|w_{t,k+1}^{\mathrm{md}} - w_{t,k}^{\mathrm{md}}| \leq \frac{1}{2}\delta_k$ for $t > 3k + 1$, and the curvature conditions also hold for $t > 3k + 1$.

Summarizing (c), (d), and (e) completes the induction. $\square$

*Proof of Lemma F.2.* Follows by applying Claim F.4. $\square$

# G Helper Lemmas

In this section we include some generic helper lemmas. Most of the results are standard and we provide the proof for completeness.

**Lemma G.1.** *Let* $A = \begin{bmatrix} A_{11} & A_{12} \\ A_{21} & A_{22} \end{bmatrix}$ *be an arbitrary* $2d \times 2d$ *block matrix, where* $A_{11}, A_{12}, A_{21}, A_{22}$ *are* $d \times d$ *matrix blocks. Then the operator norm of A is bounded by*

$$\|A\| \leq \max\left\{\|A_{11}\|, \|A_{22}\|\right\} + \left\{\|A_{12}\|, \|A_{21}\|\right\}.$$

*Proof of Lemma G.1.* Let $A_{ij} = U_{ij}\Sigma_{ij}V_{ij}^T$ be the SVD decomposition of matrix $A_{ij}$, for $i = 1, 2$, and $j = 1, 2$. Then

$$\begin{bmatrix} A_{11} & \\ & A_{22} \end{bmatrix} = \begin{bmatrix} U_{11}\Sigma_{11}V_{11}^{\mathsf{T}} & \\ & U_{22}\Sigma_{22}V_{22}^{\mathsf{T}} \end{bmatrix} = \begin{bmatrix} U_{11} & \\ & U_{22} \end{bmatrix}\begin{bmatrix} \Sigma_{11} & \\ & \Sigma_{22} \end{bmatrix}\begin{bmatrix} V_{11} & \\ & V_{22} \end{bmatrix}^{\mathsf{T}},$$

thus

$$\left\|\begin{bmatrix} A_{11} & \\ & A_{22} \end{bmatrix}\right\| = \left\|\begin{bmatrix} \Sigma_{11} & \\ & \Sigma_{22} \end{bmatrix}\right\| = \max\left\{\|\Sigma_{11}\|, \|\Sigma_{22}\|\right\} = \max\left\{\|A_{11}\|, \|A_{22}\|\right\}.$$

Similarly

$$\begin{bmatrix} & A_{12} \\ A_{21} & \end{bmatrix} = \begin{bmatrix} & U_{12}\Sigma_{12}V_{12}^{\mathsf{T}} \\ U_{21}\Sigma_{21}V_{21}^{\mathsf{T}} & \end{bmatrix} = \begin{bmatrix} & U_{12} \\ U_{21} & \end{bmatrix}\begin{bmatrix} \Sigma_{21} & \\ & \Sigma_{12} \end{bmatrix}\begin{bmatrix} & V_{21} \\ V_{12} & \end{bmatrix}^{\mathsf{T}},$$

thus

$$\left\|\begin{bmatrix} & A_{12} \\ A_{21} & \end{bmatrix}\right\| = \left\|\begin{bmatrix} \Sigma_{21} & \\ & \Sigma_{12} \end{bmatrix}\right\| = \max\left\{\|\Sigma_{12}\|, \|\Sigma_{21}\|\right\} = \max\left\{\|A_{12}\|, \|A_{21}\|\right\}.$$

Consequently, by the subadditivity of the operator norm,

$$\|A\| \leq \left\|\begin{bmatrix} A_{11} & \\ & A_{22} \end{bmatrix}\right\| + \left\|\begin{bmatrix} & A_{12} \\ A_{21} & \end{bmatrix}\right\| \leq \max\left\{\|A_{11}\|, \|A_{22}\|\right\} + \max\left\{\|A_{12}\|, \|A_{21}\|\right\}.$$

$\square$

**Lemma G.2.** *Let* $x, y \in \mathbb{R}^d$*, then for any* $\zeta > 0$*, the following inequality holds*

$$\|x + y\|^2 \leq (1 + \zeta)\|x\|^2 + (1 + \zeta^{-1})\|y\|^2.$$

*Proof of Lemma G.2.* First note that $\|x + y\|^2 = \|x\|^2 + \|y\|^2 + 2\langle x, y\rangle$, then the proof follows by $2\langle x, y\rangle \leq \zeta\|x\|^2 + \zeta^{-1}\|y\|^2$ due to Cauchy-Schwartz inequality. $\square$

**Lemma G.3.** *Let* $F\colon \mathbb{R}^d \to \mathbb{R}$ *be an arbitrary twice-continuous-differentiable function that is* $Q$*-3rd-order-smooth. Then for any* $w^1, \ldots, w^M \in \mathbb{R}^d$*, the following inequality holds*

$$\left\|\nabla F(\overline{w}) - \frac{1}{M}\sum_{m=1}^{M}\nabla F(w^m)\right\|^2 \leq \frac{Q^2}{4M}\sum_{m=1}^{M}\|w^m - \overline{w}\|^4,$$

*where* $\overline{w} := \frac{1}{M}\sum_{m=1}^{M}w^m$.

*Proof of Lemma G.3.*

$$\left\| \frac{1}{M} \sum_{m=1}^{M} \nabla F(w^m) - \nabla F(\overline{w}) \right\|^2$$

$$= \left\| \frac{1}{M} \sum_{m=1}^{M} \left( \nabla F(w^m) - \nabla F(\overline{w}) - \nabla^2 F(\overline{w})(w^m - \overline{w}) \right) \right\|^2 \quad \text{(since } \frac{1}{M} \sum_{m=1}^{M} w^m - \overline{w} = 0)$$

$$\leq \frac{1}{M} \sum_{m=1}^{M} \left\| \nabla F(w^m) - \nabla F(\overline{w}) - \nabla^2 F(\overline{w})(w^m - \overline{w}) \right\|^2 \quad \text{(Jensen's inequality)}$$

$$\leq \frac{Q^2}{4M} \sum_{m=1}^{M} \| w^m - \overline{w} \|^4. \quad \text{(}Q\text{-3rd-order-smoothness)}$$

$\square$

**Lemma G.4.** *Let $X$ and $Y$ be two i.i.d. $\mathbb{R}^d$-valued random vectors, and assume $\mathbb{E}\,X = 0$, $\mathbb{E}\,\|X\|^4 \leq \sigma^4$. Then*
$$\mathbb{E}\,\|X + Y\|^2 \leq 2\sigma^2, \quad \mathbb{E}\,\|X + Y\|^3 \leq 4\sigma^3, \quad \mathbb{E}\,\|X + Y\|^4 \leq 8\sigma^4.$$

*Proof of Lemma G.4.* The first inequality is due to $\mathbb{E}\,\|X + Y\|^2 = \mathbb{E}\,\|X\|^2 + \mathbb{E}\,\|Y\|^2 = 2\sigma^2$ where $\mathbb{E}\,\|X\|^2 \leq \sigma^2$ follows by applying Hölder's inequality to the assumption $\mathbb{E}\,\|X\|^4 \leq \sigma^4$.

The 4$^{\text{th}}$ moment is bounded as

$$\mathbb{E}\,\|X + Y\|^4 = \mathbb{E}\left[ \|X\|^2 + \|Y\|^2 + 2\langle X, Y \rangle \right]^2$$

$$= \mathbb{E}\left[ \|X\|^4 + \|Y\|^4 + 2\|X\|^2\|Y\|^2 + 4\langle X, Y \rangle^2 + 4\|X\|^2\langle X, Y \rangle + 4\|Y\|^2\langle X, Y \rangle \right]$$

$$= \mathbb{E}\left[ \|X\|^4 + \|Y\|^4 + 2\|X\|^2\|Y\|^2 + 4\langle X, Y \rangle^2 \right] \quad \text{(by independence and mean-zero assumption)}$$

$$\leq \mathbb{E}\left[ 4\|X\|^4 + 4\|Y\|^4 \right] \leq 8\sigma^4. \quad \text{(Cauchy-Schwarz inequality)}$$

The 3$^{\text{rd}}$ moment is bounded via Cauchy-Schwarz inequality since

$$\mathbb{E}\,\|X + Y\|^3 \leq \sqrt{\mathbb{E}\,\|X + Y\|^2 \, \mathbb{E}\,\|X + Y\|^4} \leq 4\sigma^3.$$

$\square$

**Lemma G.5.** *Let $\varphi(x) := \frac{1}{x^q} \log^p(a + bx)$, where $a, p, q \geq 1$, $b > 0$ are constants. Then suppose $a \geq \exp(p/q)$, it is the case that $\varphi(x)$ is monotonically decreasing over $(0, +\infty)$.*

*Proof of Lemma G.5.* Without loss of generality assume $b = 1$, otherwise we put $\psi(x) = \varphi(x/b)$ then $\psi$ has the same form (up to constants) with $b = 1$. Taking derivative for $\varphi(x) = x^{-q} \log^p(a + x)$ gives

$$\varphi'(x) = \frac{px^{-q} \log^{p-1}(a + x)}{a + x} - qx^{-q-1} \log^p(a + x)$$

$$= \frac{x^{-q-1} \log^{p-1}(a + x)}{a + x} \left( px - q(a + x) \log(a + x) \right).$$

Since $a \geq 1$ and $x > 0$ we always have $\frac{x^{-q-1} \log^{p-1}(a+x)}{a+x} \geq 0$. Suppose $a \geq \exp(p/q)$ then

$$px - q(a + x) \log(a + x) < px - qx \log(a) \leq px - qx \cdot \frac{p}{q} \leq 0.$$

Hence $\varphi'(x) < 0$ and thus $\varphi(x)$ is monotonically decreasing. $\square$