[Reviews · NeurIPS 2020]

Review 1

Summary and Contributions: The paper proposes a new version of Local-SGD/Federated Averaging algorithm -- Federated Accelerated SGD (FedAc). In particular, the algorithm solves a smooth convex expectation minimization problem in a distributed/federated fashion: M workers in parallel can access the stochastic gradients of the objective function and periodically communicate with a parameter-server. FedAc is a combination of AC-SA method from (Ghadimi and Lan, 2012) and Federated Averaging. Authors propose a first analysis of this method for generally strongly convex functions (in the convex case this method was analyzed in (Woodworth et al., 2020), but only for quadratic objectives) under the assumption that the variance of the stochastic gradients is uniformly bounded. The derived bounds outperform the state-of-the-art result for federated methods in this setting, and these rates are close to the accelerated ones. Moreover, authors show how their bounds improve under the additional assumption that the Hessian is Lipschitz continuous, and the 4-th central moment of the stochastic gradient is bounded and also extended known results for Local-SGD (FedAvg) to this case. Finally, using the l_2-penalization technique, which is a well-known trick, authors extend the obtained results to the case of generally convex functions without strong convexity. ==========After Reading the Response============= I have read the response and other reviews. I want to thank the authors for their response. Despite some weaknesses of the paper like big logarithmical factors in the bounds, the analysis of the convex case through the lens of l_2-penalization, and the assumption about the homogeneity of the data, I find the main contribution of the paper significant enough to be accepted to NeurIPS. Therefore, I want to keep my initial score (7) and thank the authors for the interesting paper.

Strengths: 1. The first (almost) accelerated method with infrequent communications in the case of homogeneous data and uniformly bounded variance of the stochastic gradient. This is a significant contribution since it makes a first step towards developing optimal federated methods. However, the obtained bound for the proposed method is worse than expected from accelerated method at least in exponentially decaying term: the extra squared root of K (number of local steps between two consequent communications) is needed to obtain acceleration in the usual sense. For example, in the case when the objective function is quadratic one can apply the restarts technique for Local-AC-SA and using Corollary 1 from (Woodworth et al., 2020) one can get the exponentially decaying term of order exp(-\sqrt{\mu/L}T). 2. Clear comparison with the state-of-the-art results in these settings with explicit convergence rates.

Weaknesses: 1. Convex case (without strong convexity) is considered through the lens of l_2-penalization. This trick allows to reduce the convex case to the strongly convex case and apply the result for the strongly convex case. However, this technique requires a particular choice of penalization parameter. That is, to use this method in practice, one should additionally tune the penalization parameter. Furthermore, this technique gives extra logarithmical factors in the complexity bounds. 2. Homogeneous data. Typically, in this case, the analysis becomes simpler, and federated methods perform very well. In some sense, it was not surprising that acceleration works well in this setting. However, a much more interesting case is the case of heterogeneous data. 3. The uniformly bounded variance of the stochastic gradients. Despite the fact that this is assumption was widely used in the stochastic optimization literature, it is quite restrictive. It would be interesting to investigate the milder assumption on noise.

Correctness: Unfortunatelly, I didn't have enough time to check all the proofs, but the results seem to be reasonable.

Clarity: The paper uses accurate claims and many aspects of the work are shown explicitly which is very important for theoretical paper. This makes the work easy to follow.

Relation to Prior Work: The most relevant literature is clearly discussed. However, I was surprised that in lines 97-98 authors do not mention the following works on quantization: 1. Dan Alistarh, Demjan Grubic, Jerry Li, Ryota Tomioka, and Milan Vojnovic. QSGD: Communication-efficient SGD via gradient quantization and encoding. In Advances in Neural Information Processing Systems, pages 1709–1720, 2017. 2. Wei Wen, Cong Xu, Feng Yan, Chunpeng Wu, Yandan Wang, Yiran Chen, and Hai Li. Terngrad: Ternary gradients to reduce communication in distributed deep learning. In Advances in Neural Information Processing Systems, pages 1509–1519, 2017. 3. Mishchenko, K., Gorbunov, E., Takáč, M. and Richtárik, P., 2019. Distributed learning with compressed gradient differences. arXiv preprint arXiv:1901.09269.

Reproducibility: Yes

Additional Feedback: 1. line 39: it would be nice to notice that (Woodworth et al., 2020) analyzed Local-AC-SA for quadratic problems as well. 2. For logistic regression one can evaluate smoothness constant L for each individual function in the sum and for the total loss as well. Therefore, one can choose \lambda to be explictly proportional to L. It will clarify what is the actual condition number of the problem. 3. Why do you test only FedAc-I in the experimental part? Have you tried to conduct experiments with Fed-Ac-II? Does it really give better communication complexity than FedAc-I in your experiments?


Review 2

Summary and Contributions: This paper explores a principled acceleration for federated learning. The theoretical results show that acceleration can significantly improve the communication efficiency in the federated learning context. The supplementary gives complete analysis.

Strengths: I appreciate this work for the combination of acceleration into the federated learning context. I think the improvement for the synchronization R is very significant theoretically despite many large constants and polylog factors. In the appendix, the authors show a systematic analysis.

Weaknesses: This is a convex paper, but is full of the style of nonconvex ones. In nonconvex optimization, we are tolerated for large constants and polylog factors, but in the convex setting, it is often the case that the analysis can be very elegant and tight. In my experience, convex analys is often so tight that the effect of the constants in the convergence bound can be found in experiments. As a researcher mainly in convex area, I think the bounds have many undesired terms such as exp(18) and polylog^4, etc. I strongly believe such terms are not essential. Meanwhile, in the main body, if the authors can not cancel these strange terms, I recommend the authors do not use tilde(O) to hide any log factors or large constants, which I believe is not the style for convex analysis. Again, for convex optimizationk, I should emphasize that log factors are significant. If we do not consider log factors, studying the general convex setting is meaningless as the log suboptimal result can always be derived from the strongly convex setting. I think the treatment for the general convex setting by regularization is not very careful. This paper considers the stochastic optimization setting with infinite data assumption. However, as the experiment shows the data is large scale but is often finite. Can author also compare with distributed finite sum solvers in experiments?

Correctness: I am not quite sure about the correctness of Theorems 2 and 3. For quadratic function, Q = 0, so the term about the stochastic variance is independent from R. Is it correct?

Clarity: I recommend authors not to hide logfactors in the main body.

Relation to Prior Work: Clearly discussed.

Reproducibility: Yes

Additional Feedback:


Review 3

Summary and Contributions: This paper proposed an accelerated variant of FedAvg, where the acceleration technique is applied to the updates of local variables. The convergence property of this algorithm is analyzed, which suggests the convergence rate and the required number of synchronization are better than FedAvg. Experiments also proved its efficiency.

Strengths: The theory in this work is sound, and the theoretical bound is better than existing related methods.

Weaknesses: Though this paper developed theoretical bounds better than existing methods, the insights of these bounds are lacked, e.g., why this bound looks like this, what each term intuitively implies, which part of improvement is contributed by acceleration technique. Just showing "our bound is better than others" is not good enough and may limit the impact of the paper.

Correctness: Yes.

Clarity: The algorithm and experiment parts are easy to read. But more explanation and discussion are needed for the theoretical results.

Relation to Prior Work: Yes.

Reproducibility: Yes

Additional Feedback: I think it is better to compare the theoretical bounds with more existing works, such as accelerated distributed SGD. I am curious how much communication costs can be saved by communicating every K iterations instead of every iteration. In line 10 of Algorithm 1, the meaning of $m$ is ambiguous. it is better to use another symbol in the summations. ==========After Authors' Rebuttal============= I have read authors' feedback, and decided to raise my score.


Review 4

Summary and Contributions: This work propose a new Accelerated Stochastic Gradient Descent for Federated learning setting. The algorithm is based on ASGD from Ghadimi and Lan (2012) and has been proved theoretically to achieve a significant lower synchronization rounds over FEDAVG for strongly convex and convex setting. Numerical experiments are provided to verify the theoretical results.

Strengths: As far as I know, this is the first provable acceleration of FED-AVG. Since federated learning is more and more popular this year, I believe NeuRIPS community will be interested. The technique it uses is of independent interest.

Weaknesses: The convergence analysis is still limited to convex functions but the authors conjecture that Fed-AC can be generalized to non-convex cases. Does that mean we can implement accelerated sgd in nonconvex settings such as that in (Ghadimi and Lan 2013) to Federated learning setting and gain performance improvement? Please provide more comments and details. The authors feedback has addressed my concern. Ghadimi and Lan 2013: Accelerated Gradient Methods for Nonconvex Nonlinear and Stochastic Programming

Correctness: The analysis looks sound to me.

Clarity: Yeah, very clear.

Relation to Prior Work: Yes. The authors provide very detailed literature review.

Reproducibility: Yes

Additional Feedback:

[Author Response · NeurIPS 2020]

1　We thank the reviewers for the detailed and insightful reviews. The reviewers noted that our work 1) makes "significant
2　contribution" [R1] and "significantly improves the communication efficiency" [R3, R7], 2) makes "clear comparison
3　with the state-of-the-art results" [R1], 3) provides "accurate claims" [R1] and "sound theory" [R5], and 4) "the technique
4　is of independent interest" [R7]. We will answer questions below and incorporate feedback into our final revision.
5　**[R1]**: "a much more interesting case is the case of heterogeneous data"
6　• Our proof framework can generalize to heterogeneous data if we allow an overhead term that depends on the inter-
7　client heterogeneity. For example, if we assume $\sup_w \|\nabla F_m(w) - \nabla F(w)\| \leq \zeta$, we can modify the stability analysis
8　by introducing an additive overhead involving $\zeta$ as in the heterogeneous analysis of FEDAVG [Woodworth et al., 2020].
9　That said, it may be more interesting to design heterogeneity-aware algorithms that avoid this overhead, which was so
10　far not well-understood even without acceleration and is beyond the scope of this paper.
11　**[R1]**: "the extra squared root of $K$ (# of local steps) is needed to obtain acceleration in the usual sense"
12　• The $\sqrt{K}$ originates from the acceleration-stability tradeoff (Line 226). While it remains an open question whether the
13　overhead is necessary for lower bounds, we provided two evidences 1) theoretically standard AGD is not initial-value
14　stable (§F), and 2) empirically direct federation of AGD may indeed perform worse than our principled FEDAC (§A.4).
15　**[R1]**: "in lines 97-98 authors do not mention the following works on quantization: ..."
16　• Thanks for the suggestion of the references. We will add these references regarding quantization in the next revision.
17　**[R1]**: "(at Line 39) notice that (Woodworth et al., 2020) analyzed Local-AC-SA for quadratic problems as well."
18　• We have mentioned Woodworth et al.'s work on quadratic at Line 60. We will reiterate at Line 39 in the next revision.
19　**[R1]**: "Why do you test only FEDAC-I in experiment? Have you tried FEDAC-II?"

20　• We have indeed tested FEDAC-II and stated in §A that "FEDAC-II is qualitatively
21　similar to FEDAC-I empirically so we show FEDAC-I only." We will include more
22　experiments on FEDAC-II in the next revision. Particularly under the same settings of
23　§5, FEDAC-I is slightly better as the condition number is large (see figure on the right).
24　**[R3]**: "Can author compare with distributed finite-sum solvers in experiments?"

25　• We compared with DSVRG (Lee et al., 2017), see figure on the right. Under the same
26　settings of §5 (dataset a9a, size 33k) FEDAC outperforms DSVRG. On a smaller (1.6k)
27　dataset a1a, DSVRG is better only if the communication is very frequent. In general,
28　one can obtain moderate accuracy with FEDAC in a short parallel time under limited
29　communication, whereas finite-sum solvers may be preferred if high accuracy is required
30　and the dataset is relatively small. We conjecture that FEDAC can be incorporated with
31　variance reduction techniques to attain better performance in finite-sum (ERM) settings.
32　**[R3]**: "For $Q = 0$, the term about variance is independent from $R$. Is it correct?"
33　• Exactly. When $Q = 0$, the last term will vanish. This gives a smooth interpolation of existing quadratic analysis.
34　**[R3]**: The bounds have undesired terms such as large constants and polylog factor.
35　• Thanks for your suggestions. We will try to reduce these terms in the next revision. For strongly-convex analysis, the
36　polylog factors are the artifacts of constant step-size $\eta$ and do not emerge until the very end of the analysis. For example,
37　Lemma 6 (Convergence of FEDAC-I for general $\eta$) does not involve any polylog factors. As stated in footnote 12 on
38　Page 17, there are standard techniques (e.g., [Lacoste-Julien et al., 2012]) to reduce such polylog factors by decaying $\eta$
39　and averaging. However, adopting such techniques will complicate the overall analysis due to the time-variant $\eta$.
40　**[R5]**: "the insights of bounds are lacked" "compare with more works such as accelerated distributed SGD"
41　• We will clarify the insights more in next revision. Here is the summary: for general-convex case, under A1, our bound
42　for FEDAC-I is $\tilde{\mathcal{O}}(\frac{LD_0^2}{TR} + \frac{\sigma D_0}{\sqrt{MT}} + \frac{L^{1/3}\sigma^{2/3}D_0^{4/3}}{T^{1/3}R^{2/3}})$. The first term corresponds to the deterministic convergence, which is
43　better than the one for FEDAVG, that is, $LD_0^2/T$. The second term corresponds to the stochasticity of the problem which
44　is not improvable. The third term corresponds to the overhead of infrequent communication, which is also better than
45　FEDAVG ($\frac{L^{1/3}\sigma^{2/3}D_0^{4/3}}{T^{1/3}R^{1/3}}$) due to acceleration. The intuition is that FEDAC can achieve the same progress with smaller
46　step-size $\eta$, which lowers this overhead incurred by the discrepancy of clients (see Line 51-55 for related discussions).
47　• Ideally we hope the second term to dominate the bound so one can gain by scaling up $M$. Since the third term of
48　FEDAC has better dependency on $R$, one only need fewer rounds of communication to keep the third term dominated.
49　• The bound for accelerated-distributed-SGD is $\mathcal{O}(\frac{LD_0^2}{R^2} + \frac{\sigma D_0}{\sqrt{MT}})$. In comparison to FEDAC, it has worse deterministic
50　convergence rate (i.e. first term, since $T \geq R$ trivially holds) but does not incur the third-term overhead.
51　**[R7]**: "The authors conjecture FEDAC can be generalized to non-convex problems. Does that mean we can implement
52　accelerated sgd such as (Ghadimi et al., 2013) to Federated setting and gain performance improvement?"
53　• While FEDAC may lead to empirical improvements on non-convex problems, we do not expect it to gain theoretical
54　improvement over FEDAVG. This is because directly applying convex acceleration may not improve non-convex
55　rates even with a single machine (e.g., [Ghadimi et al., 2013] does not have better theoretical bounds than SGD for
56　non-convex problems). However, it is possible to combine our result with recent non-convex acceleration algorithms
57　(e.g., [Carmon et al., 2018]) that use convex acceleration in a more sophisticated way and are provably faster than GD.

[Meta-Review · NeurIPS 2020]

This paper has been well-received by the reviewers and the author response addressed the reviewers' concerns adequately. Thus, the paper is suitable for acceptance at NeurIPS without significant changes.